# Training Neural Networks as Recognizers of Formal Languages

**Alexandra Butoi**[1]     **Ghazal Khalighinejad**[2]     **Anej Svete**[1]
**Josef Valvoda**[3]     **Ryan Cotterell**[1]     **Brian DuSell**[1]
[1]ETH Zürich     [2]Duke University     [3]University of Copenhagen
{alexandra.butoi, anej.svete, ryan.cotterell, brian.dusell}@inf.ethz.ch
ghazal.khalighinejad@duke.edu     jval@di.ku.dk

## Abstract

Characterizing the computational power of neural network architectures in terms of formal language theory remains a crucial line of research, as it describes lower and upper bounds on the reasoning capabilities of modern AI. However, when empirically testing these bounds, existing work often leaves a discrepancy between experiments and the formal claims they are meant to support. The problem is that formal language theory pertains specifically to recognizers: machines that receive a string as input and classify whether it belongs to a language. On the other hand, it is common instead to evaluate language models on proxy tasks, e.g., language modeling or sequence-to-sequence transduction, that are similar in only an informal sense to the underlying theory. We correct this mismatch by training and evaluating neural networks directly as binary classifiers of strings, using a general method that can be applied to a wide variety of languages. As part of this, we extend an algorithm recently proposed by Snæbjarnarson et al. (2025) for efficient length-controlled sampling of strings from regular languages. We provide results on a variety of languages across the Chomsky hierarchy for three neural architectures: a simple RNN, an LSTM, and a causally-masked transformer. We find that the RNN and LSTM often outperform the transformer, and that auxiliary training objectives such as language modeling can help, although no single objective uniformly improves performance across languages and architectures. Our contributions will facilitate theoretically sound empirical testing of language recognition claims in future work. We have released our datasets as a benchmark called FLaRe[1] (Formal Language Recognition), along with our code.[2]

## 1 Introduction

Large language models (LLMs) based on neural networks have been hailed for their impressive reasoning abilities. However, the precise scope of what they can and cannot do is hard to pin down. Fortunately, formal language theory gives us a vocabulary for ascribing hard limits to the kinds of computations neural networks can perform. For example, results from formal language theory allow us to know with certainty that a transformer LM (without extra chain-of-thought steps) cannot determine whether two regular expressions with repetition operators are equivalent; a transformer LM runs in quadratic time, but the aforementioned problem provably requires exponential time (Sipser, 2013).

A long line of research has attempted to precisely characterize the class of problems neural architectures can solve in terms of formal languages (Siegelmann & Sontag, 1995; Gers & Schmidhuber, 2001; Weiss et al., 2018b; Merrill et al., 2020; Strobl et al., 2024b; Yang et al., 2024, *inter alia*). Research in this line comes in two flavors: formal results that mathematically prove language class bounds, often under simplifying assumptions, and empirical results that, in complementary fashion, provide evidence of these bounds in practice. Simplifying assumptions for the transformer architecture have included the absence of layer normalization, the use of hard attention, and the use of special positional encodings; see Strobl et al. (2024b) for a survey. Formal expressivity results

---

[1]https://github.com/rycolab/flare
[2]https://github.com/rycolab/neural-network-recognizers

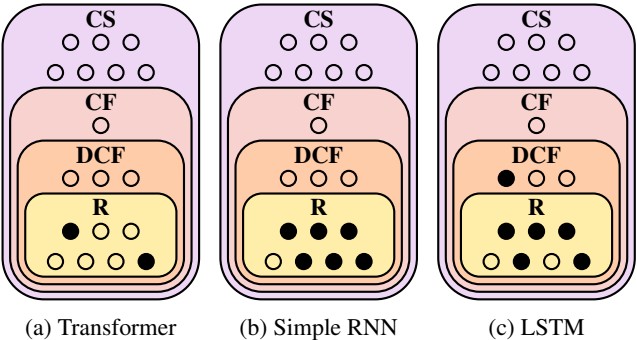

(a) Transformer      (b) Simple RNN      (c) LSTM

Figure 1: Summary of our empirical expressivity results. Dots represent languages, which are listed in Table 1. A filled dot means that the architecture exhibits perfect length generalization (see Table 2 under "Expressivity"). R = regular, DCF = deterministic context-free, CF = context-free, CS = context-sensitive. All architectures are limited to regular languages and the DCF language Majority. The transformer is strictly less expressive than the RNN/LSTM on the languages we tested.

also typically do not comment on whether solutions are practically reachable through training, even though the bias imposed by the training algorithm may render the set of solvable problems much smaller than suggested by formal expressivity results (Hahn & Rofin, 2024). Empirical results are therefore important for validating the practical relevance of formal results.

This paper aims to reconcile a subtle but important disconnect between empirical results and claims about computational power. Formal language theory deals in recognizers: machines that receive a string as input and classify whether it is a member of a language. For instance, the Chomsky hierarchy and the classes P and NP are defined in terms of recognizers. However, some machine learning papers overlook the finer points of these definitions. For instance, the recent study by Delétang et al. (2023) purports to relate the expressivity of neural language models to the Chomsky hierarchy with experiments that use neural networks as string-to-string functions. However, the mismatch between the experimental setup and the actual definition of the Chomsky hierarchy means that the paper would provide evidence about an analogous hierarchy of *functions*, which is not as well understood formally. There are multiple ways to fix this mismatch: one could change the theoretical claims to those of a hierarchy of string-to-string functions (Strobl et al., 2024a) without changing the experiments;[3] or, one could change the experiments to recognition to match the Chomsky hierarchy. In this paper, we explore the latter approach.

Specifically, we propose a method of training neural networks as recognizers that suits a wide variety of languages, and we use it to test simple RNNs, LSTMs, and transformers on 18 formal languages across the Chomsky hierarchy. Our method for generating data only requires two language-specific algorithms: randomly sampling positive examples within a particular length range, and membership testing. We provide efficient instantiations for the class of regular languages based on work by Snæb-jarnarson et al. (2025). We generate adversarial negative examples by modifying positive examples, obviating the need for language-specific negative sampling (cf. Weiss et al., 2018b; Someya et al., 2024; Bhattamishra et al., 2024). Like Delétang et al. (2023), our evaluation focuses on length generalization, but we also include two sets of experiments that carefully distinguish between tests of inductive bias and expressivity. We also compare multiple training objectives in a multi-task learning setup. We find that the transformer generally underperforms the RNN and LSTM, and all architectures are limited to low levels of the Chomsky hierarchy (Figure 1). We have publicly released our datasets as a benchmark called **FLaRe** (**F**ormal **La**nguage **Re**cognition), along with our code.

## 2 BACKGROUND

An **alphabet**, denoted $\Sigma$ throughout this paper, is a non-empty, finite set of elements called **symbols**. A **string** over $\Sigma$ is a finite sequence of symbols in $\Sigma$. We use $\varepsilon$ to denote the empty string. A **language** (or **formal language**) over $\Sigma$ is a (possibly infinite) set of strings over $\Sigma$. Let $\Sigma^*$ denote

---

[3]Hierarchies of language models have also been studied (Icard, 2020; Borenstein et al., 2024).

the language of all possible strings over $\Sigma$. A **recognizer** is any computational device that receives a string as input and produces a decision to **accept** or **reject** it as output, and we say that it **recognizes** the language of strings that it accepts. A **language class** is a (possibly infinite) set of languages. Throughout the rest of this paper, let us assume we are dealing with a specific language $L$ over alphabet $\Sigma$. For any string $w \in \Sigma^*$, we call the proposition $w \in L$ the **label** of $w$.

## 2.1 FINITE AUTOMATA AND REGULAR LANGUAGES

**Definition 1.** *A **partial deterministic finite automaton (partial DFA)** is a tuple $\mathcal{A} = (Q, \Sigma, \delta, q_0, F)$ where (1) $Q$ is a finite set of states, (2) $\Sigma$ is an alphabet, (3) $\delta \colon Q \times \Sigma \to Q \cup \{\emptyset\}$ is the transition function, where $\emptyset$ indicates the absence of a transition, (4) $q_0 \in Q$ is the start state, and (5) $F \subseteq Q$ is the set of accepting states. If $\delta(q, a) = r \neq \emptyset$, we say that $\mathcal{A}$ has a transition from $q$ to $r$ that scans $a$, and we write $q \xrightarrow{a} r \in \delta$. We define $|\delta| \stackrel{\text{def}}{=} |\{(q, a) \in Q \times \Sigma \mid \delta(q, a) \neq \emptyset\}|$.*

Our definition of DFA is *partial* in the sense that we do not require an outgoing transition to exist for all $(q, a) \in Q \times \Sigma$. For simplicity, from now on, we will simply refer to partial DFAs as DFAs. We call a sequence $\pi$ of connecting states and transitions a **path**. If $\pi$'s transitions scan $a_1, \ldots, a_n$, we say that $\pi$ **scans** the string $w = a_1 \cdots a_n$. For any string $w \in \Sigma^*$, if there is a path that starts in $q_0$, scans $w$, and ends in an accepting state, we say that $\mathcal{A}$ **accepts** $w$ and **rejects** it otherwise. We say that $\mathcal{A}$ **recognizes** the language $\{w \in \Sigma^* \mid \mathcal{A} \text{ accepts } w\}$. A language is **regular** if there is a DFA that recognizes it. If all states in a DFA are reachable from the start state and can lead to an accepting state, we call it **trim**.

Because we will present algorithms that use semirings and weighted DFAs, we introduce them here.

**Definition 2.** *A **monoid** is a tuple $(\mathbb{K}, \odot, \boldsymbol{I})$, where $\mathbb{K}$ is a set, $\odot$ is an associative binary operation, and $\boldsymbol{I} \in \mathbb{K}$, called the **identity** element, satisfies $\boldsymbol{I} \odot a = a \odot \boldsymbol{I} = a$ for all $a \in \mathbb{K}$. If $a \odot b = b \odot a$ for all $a, b \in \mathbb{K}$, we say that the monoid is **commutative**.*

**Definition 3.** *A **semiring** is a tuple $(\mathbb{K}, \oplus, \otimes, \boldsymbol{0}, \boldsymbol{1})$ where $(\mathbb{K}, \oplus, \boldsymbol{0})$ is a commutative monoid and $(\mathbb{K}, \otimes, \boldsymbol{1})$ is a monoid. Additionally, $\otimes$ distributes over $\oplus$: $a \otimes (b \oplus c) = (a \otimes b) \oplus (a \otimes c)$ and $(a \oplus b) \otimes c = (a \otimes c) \oplus (b \otimes c)$; furthermore, $\boldsymbol{0}$ is absorbing with respect to $\otimes$: $\boldsymbol{0} \otimes a = a \otimes \boldsymbol{0} = \boldsymbol{0}$.*

**Definition 4.** *A **weighted deterministic finite automaton (WDFA)** over a semiring $(\mathbb{K}, \oplus, \otimes, \boldsymbol{0}, \boldsymbol{1})$ is a tuple $\mathcal{A} = (Q, \Sigma, \delta, q_0, \rho)$ such that (1) $Q$, $\Sigma$, and $q_0$ are defined as in Def. 1, (2) $\delta \colon Q \times \Sigma \to (Q \times \mathbb{K}) \cup \{\emptyset\}$ is the transition function, and (3) $\rho \colon Q \to \mathbb{K}$ is the accept weight function. If $\delta(q, a) = (r, w)$, we say that $\mathcal{A}$ has a transition from $q$ to $r$ that scans $a$ with weight $w$, and we write $q \xrightarrow{a/w} r \in \delta$.*

Rather than accepting or rejecting a string, a WDFA assigns it a weight. The weight of a path is the product of the weights of its transitions and the accept weight of its last state. The weight of a string is the weight of the path that scans it, or $\boldsymbol{0}$ if one does not exist. We can assign probabilities to strings with the **real semiring** $(\mathbb{R}_{\geq 0}, +, \times, 0, 1)$ or, for numerical stability, the **log semiring** $(\mathbb{R} \cup \{-\infty\}, \log(\exp(\cdot) + \exp(\cdot)), +, -\infty, 0)$. A WDFA must constitute a probability distribution in order to allow sampling from it. One way to achieve this is by locally normalizing the WFDA per state.

**Definition 5.** *A WDFA $\mathcal{A} = (Q, \Sigma, \delta, q_0, \rho)$ over the real semiring is called a **probabilistic DFA (PDFA)** if for all $q \in Q$, $\left( \sum_{q \xrightarrow{a/p} r \in \delta} p \right) + \rho(q) = 1$, and $\mathcal{A}$ induces a probability distribution over $\Sigma^*$.[4]*

## 3 METHOD

We now give a method for generating datasets of formal languages and training neural networks as recognizers. We address a number of challenges cited in past work, namely length-constrained sampling, negative sampling, and the paucity of the training signal when training on binary labels.

---

[4]This second condition, known as tightness, is easily checkable in the case of WDFAs (Du et al., 2023).

## 3.1 DATASET GENERATION

Here, we describe how to generate a set of $N$ string–label pairs for language $L$, where the length of every string falls in the range $[n_{\min}, n_{\max}]$. To do this, we will require two language-specific algorithms: (1) an algorithm for sampling a string $w$ from $L$ such that $|w| \in [n_{\min}, n_{\max}]$ (using a distribution to be described in §3.1.1), and (2) an algorithm for determining whether a string $w \in \Sigma^*$ is a member of $L$.[5] To generate $N$ strings, we uniformly sample a label from $\{0, 1\}$ $N$ times, ensuring a balanced dataset, and we generate a positive (§3.1.1) or negative (§3.1.3) string accordingly.

### 3.1.1 POSITIVE SAMPLING

The choice of probability distribution we sample from is important, as it affects the quality of training and evaluation; at a minimum, it should assign non-zero probability to all strings in $L$ with lengths in $[n_{\min}, n_{\max}]$. We detail our choices below.

When $L$ is regular, we assume we have a trim DFA $\mathcal{A} = (Q, \Sigma, \delta, q_0, F)$ that recognizes it. The set of actions that may occur at each state consists of its outgoing transitions and, if it is an accepting state, acceptance. We convert $\mathcal{A}$ to a PDFA $\mathcal{A}'$ by assigning uniform probabilities to these actions at each state (details in Algorithm 1). Let $p_{\mathcal{A}'}$ denote the probability distribution defined by $\mathcal{A}'$, and W a $p_{\mathcal{A}'}$-distributed string-valued random variable. Let $N_{\mathcal{A}'} \stackrel{\text{def}}{=} \{n \in [n_{\min}, n_{\max}] \mid \exists w \in \Sigma^n, p_{\mathcal{A}'}(w) > 0\}$. First, we sample a length $n$ uniformly from $N_{\mathcal{A}'}$. Then, we sample a string from the conditional distribution $p_{\mathcal{A}'}(\text{W} \mid |\text{W}| = n)$. This amounts to sampling from the distribution

$$p(w) = \mathbb{1}[\,|w| \in N_{\mathcal{A}'}\,] \frac{1}{|N_{\mathcal{A}'}|} \, p_{\mathcal{A}'}(\text{W} = w \mid |\text{W}| = |w|). \tag{1}$$

We will show how to sample from $N_{\mathcal{A}'}$ and $p_{\mathcal{A}'}(\text{W} \mid |\text{W}| = n)$ efficiently in §3.1.2. For non-regular languages, we use language-specific distributions detailed in App. E.

### 3.1.2 EFFICIENT LENGTH-CONSTRAINED SAMPLING FOR REGULAR LANGUAGES

How do we compute $N_{\mathcal{A}'}$ and sample from $p_{\mathcal{A}'}(\text{W} \mid |\text{W}| = n)$ efficiently? One approach would be to construct a PDFA for $p_{\mathcal{A}'}(\text{W} \mid |\text{W}| = n)$. We could do this by intersecting $\mathcal{A}'$ with a DFA that recognizes $\Sigma^n$, which would take $O((|Q| + |\delta|)n)$ time. This would result in a WDFA that is not necessarily probabilistic, so as a prerequisite for sampling from it, we would need to re-distribute its weights to locally sum to 1 (Def. 5), using a procedure known as weight pushing (Mohri, 2009) that would run in $O((|Q|n)^3 + |\delta|n)$ time. Supposing $n_{\min} = 0$, it would thus take $O(|Q|^3 n_{\max}^4 + |\delta| n_{\max}^2)$ time to do this for all lengths. This would not be scalable for our experiments, in which we sample strings up to length $n_{\max} = 500$. Fortunately, we can improve this by a factor of $O(n_{\max}^2)$ using an object proposed by Snæbjarnarson et al. (2025) called the **binning semiring**. The key idea is to *share* computation among the different lengths by running a version of the weight pushing algorithm—only once—that computes the normalized weights for all lengths. Instead of weighting the DFA with probabilities, we use *vectors* that bin probabilities by each length in $[0, n_{\max}]$.

**Definition 6.** *Let $\mathcal{W} = (\mathbb{K}, \oplus, \otimes, \mathbf{0}, \mathbf{1})$ be a semiring, and let $D \in \mathbb{Z}_{\geq 0}$. For $\boldsymbol{v} \in \mathbb{K}^{D+1}$, we write $\boldsymbol{v} = (\boldsymbol{v}_0, \boldsymbol{v}_1, \ldots, \boldsymbol{v}_D)$. The $D^{\text{th}}$-order binning semiring with respect to the **base semiring** $\mathcal{W}$ is the semiring $\mathcal{W}^D = (\mathbb{K}^{D+1}, \circledoplus, \circledotimes, \mathbf{\textcircled{0}}, \mathbf{\textcircled{1}})$, where:*

$$(\boldsymbol{u} \circledoplus \boldsymbol{v})_i \stackrel{\text{def}}{=} \boldsymbol{u}_i \oplus \boldsymbol{v}_i \qquad\qquad (\boldsymbol{u}, \boldsymbol{v} \in \mathbb{K}^{D+1}; 0 \leq i \leq D) \tag{2a}$$

$$(\boldsymbol{u} \circledotimes \boldsymbol{v})_i \stackrel{\text{def}}{=} \bigoplus_{j=0}^{i} \boldsymbol{u}_j \otimes \boldsymbol{v}_{i-j} \qquad\qquad (\boldsymbol{u}, \boldsymbol{v} \in \mathbb{K}^{D+1}; 0 \leq i \leq D) \tag{2b}$$

$$\mathbf{\textcircled{0}} \stackrel{\text{def}}{=} (\mathbf{0}, \ldots, \mathbf{0}) \qquad\qquad \mathbf{\textcircled{1}} \stackrel{\text{def}}{=} (\mathbf{1}, \mathbf{0}, \ldots, \mathbf{0}) \tag{2c}$$

The indices of the vectors in the $D^{\text{th}}$-order binning semiring represent the values of an integer counter from 0 to $D$; the meaning of the counter depends on how the semiring is used. We will use

---

[5]This is directly comparable to the classical learning theory of Gold (1967), which assumes the availability of a "text" of positive examples and an "informant" that provides labels.

the counter to keep track of the number of symbols scanned by a DFA, i.e., the length of a sampled string. To do length-constrained sampling from the PDFA $\mathcal{A}'$, we **lift** it to a $n_{\max}^{\text{th}}$-order binning semiring-weighted DFA $\mathcal{A}_D$ as follows. For every transition $q \xrightarrow{a/p} r$ in $\mathcal{A}'$, we set the weight of $q \xrightarrow{a} r$ in $\mathcal{A}_D$ to $(\mathbf{0}, p, \mathbf{0}, \ldots, \mathbf{0})$, indicating that the transition scans exactly one symbol with probability $p$. For every state with accept weight $p$, we set its accept weight in $\mathcal{A}_D$ to $(p, \mathbf{0}, \ldots, \mathbf{0})$, indicating that it accepts (and scans no symbols) with probability $p$. A similar interpretation applies to anything with a weight, including transitions, accepting states, paths, and strings.

Running a semiring generalization of weight pushing on $\mathcal{A}_D$ allows us to compute exactly the quantities we need for efficient sampling from $p_{\mathcal{A}'}(\mathrm{W} \mid |\mathrm{W}| = n)$. The key idea is that for every state and $0 \leq i \leq D$, it computes a probability distribution over actions (transitioning or accepting) conditioned on scanning exactly $i$ symbols in the future. Then $N_{\mathcal{A}'}$ is the set of all $n$ for which we can perform any action at $q_0$ with nonzero probability, conditioned on scanning $n$ symbols in the future. The preprocessing step now takes $O(|Q|^3 n_{\max}^2 + |\delta| n_{\max}^2)$ time, and sampling a string of length $n$ takes $O(n \log |\delta|_{\text{out}})$ time, where $|\delta|_{\text{out}}$ is the maximum out-degree of $\mathcal{A}$. See App. B for details.

### 3.1.3 NEGATIVE SAMPLING AND MEMBERSHIP TESTING

To generate a negative example, we repeatedly propose a random string $w$ until we achieve that $w \notin L$ using a membership testing algorithm for $L$. We propose $w$ in one of two ways, with uniform probability. Half the time, we uniformly sample a length $n$ from $[n_{\min}, n_{\max}]$, then uniformly sample $w$ from $\Sigma^n$. For many languages, this is very likely to produce a negative example, but one that is so superficially dissimilar to positive examples that the classification problem becomes too easy and fails to demonstrate the underlying algorithm. Prior work, therefore, typically uses language-specific rules to generate adversarial negative samples (Weiss et al., 2018b; Bhattamishra et al., 2024; Someya et al., 2024). To keep our methodology more general, the other half of the time, we propose $w$ by sampling a positive example and perturbing it with random edits (cf. Weiss et al., 2018a). More precisely, we (1) sample the number of edits $K$ from a geometric distribution that heavily favors small $K$, and (2) iteratively apply $K$ uniformly-sampled edits (single-symbol insertion, replacement, or deletion). See App. A for details. Typically, this is much more likely to produce negative examples that are difficult to distinguish from positive ones (see also our analysis in Figure 2).

Membership testing is necessary because it is possible that a random string $w$ is inadvertently a member of $L$. DFAs have a straightforward $O(n)$-time membership testing algorithm: start in the initial state, follow the unique $w$-scanning path (if it exists), and accept if it ends in an accepting state. For non-regular languages, we use language-specific membership testing algorithms detailed in App. E. A limitation of this rejection sampling approach is that it is impractical when the complement of $L$ is small, i.e., when the expected probability of successfully sampling a negative string is small. For many languages of interest, including all those we test in this paper, this probability is reasonably large regardless of $n$, in which case it is not an issue.

## 3.2 NEURAL NETWORK ARCHITECTURES

We compare three neural network architectures: (1) a multi-layer **simple RNN** (Elman, 1990) with a $\tanh$ activation function and learned initial hidden states, (2) a multi-layer **LSTM** (Gers & Schmidhuber, 2001) with decoupled input and forget gates and learned initial hidden states, and (3) a causally masked **transformer** encoder (Vaswani et al., 2017) with pre-norm (Wang et al., 2019; Nguyen & Salazar, 2019) and sinusoidal positional encodings. In all cases, the model, $M$, receives a string of symbols $w = w_1 \cdots w_n \in \Sigma^n$ as input and produces a sequence of **hidden vectors** $\boldsymbol{h}_0, \boldsymbol{h}_1, \ldots, \boldsymbol{h}_n \in \mathbb{R}^d$, which are used to compute logits for the loss terms. For the simple RNN and LSTM, the hidden vectors are the hidden states of the last layer. For the transformer, the hidden vectors are the outputs of the last layer, and we always prepend a reserved BOS symbol to the input to compute $\boldsymbol{h}_0$. See App. C for details.

We apply a learned affine transformation, called the **recognition head**, to the last hidden vector to classify whether the string is a member of the language. Let $\sigma(\cdot)$ be the pointwise logistic function. Then, we model the probability of acceptance as

$$p_M(w \in L) \stackrel{\text{def}}{=} \sigma(\boldsymbol{w}_{\mathrm{R}} \cdot \boldsymbol{h}_n + b_{\mathrm{R}}), \tag{3}$$

where $\boldsymbol{w}_\mathrm{R} \in \mathbb{R}^d$ and $b_\mathrm{R} \in \mathbb{R}$ are learnable parameters. We say that the model **accepts** $w$ if $p_M(w \in L) \geq \frac{1}{2}$ and **rejects** it otherwise.

## 3.3 TRAINING OBJECTIVES

In all experiments, we train a neural network to classify whether an input string is in $L$ by minimizing the binary cross-entropy of the recognition head (cf. Weiss et al., 2018b; Bhattamishra et al., 2023; van der Poel et al., 2024; Hahn & Rofin, 2024; Bhattamishra et al., 2024). For any probability $p$ and proposition $\phi$, we define the binary cross-entropy $H_\phi(p)$ as follows:

$$H_\phi(p) \stackrel{\text{def}}{=} \begin{cases} -\log(p) & \text{if } \phi \\ -\log(1-p) & \text{otherwise.} \end{cases} \tag{4}$$

We minimize the following loss function:

$$\mathcal{L}_\mathrm{R}(M, w) \stackrel{\text{def}}{=} H_{w \in L}(p_M(w \in L)). \tag{5}$$

However, using $\mathcal{L}_\mathrm{R}(M, w)$ as the only term in the training objective might not be enough. For one, it might be difficult to learn to orchestrate a large number of internal computational steps given only a single bit of information per example. Moreover, the gradient originates only from the last timestep, which is problematic for the RNN and LSTM, which are susceptible to the exploding and vanishing gradient problems. It is presumably for these reasons, in addition to the need for negative sampling, that most prior work has shied away from a pure language recognition training objective. One way to alleviate these issues is to provide the model with hints about intermediate computational steps. Indeed, a common setup in past work is to have the model predict, for each timestep $t$, the set of symbols that may appear at position $t$ given the prefix $w_{<t} \stackrel{\text{def}}{=} w_1 \cdots w_{t-1}$ (Cleeremans et al., 1989; Gers & Schmidhuber, 2001; Schmidhuber et al., 2002; Rodriguez & Wiles, 1997; Suzgun et al., 2019a;b;c; Bhattamishra et al., 2020a;b;c; Ebrahimi et al., 2020). Other work uses language modeling, eschewing negative sampling entirely (Merrill, 2019; Hewitt et al., 2020; DuSell & Chiang, 2020; 2022; 2023; 2024; Liu et al., 2023; Akyürek et al., 2024; Someya et al., 2024; Borenstein et al., 2024).

To this end, we include experiments that add one or both of the following auxiliary loss terms to the training objective for positive examples: (1) a **language modeling** loss term, $\mathcal{L}_\mathrm{LM}(M, w)$, which requires the model to learn a distribution over the next symbol at each position, and (2) a **next-symbol prediction**[6] loss term, $\mathcal{L}_\mathrm{NS}(M, w)$, which requires the model to predict *whether* each symbol may appear next at each position, given the prefix of $w$ seen so far.

When we include the language modeling loss term, we add a **language modeling head** to the model that computes conditional probabilities $p_M(a \mid w_{<t})$ for each timestep $t$ and symbol $a$. We average cross-entropy over timesteps to get the loss term. We now define this formally. Let $\Sigma_\mathrm{EOS} \stackrel{\text{def}}{=} \Sigma \cup \{\mathrm{EOS}\}$ and $\boldsymbol{E} \in \mathbb{R}^{|\Sigma_\mathrm{EOS}| \times d}$ be a matrix of output embeddings, which is tied to the input embeddings of the model. Then we define the loss term as follows:

$$p_M(a \mid w_{<t}) \stackrel{\text{def}}{=} \mathrm{softmax}(\boldsymbol{E}\boldsymbol{h}_{t-1})_a \qquad (1 \leq t \leq n+1; a \in \Sigma_\mathrm{EOS}) \tag{6a}$$

$$\mathcal{L}_\mathrm{LM}(M, w) \stackrel{\text{def}}{=} \frac{1}{n+1} \sum_{t=1}^{n+1} -\log p_M(w_t \mid w_{<t}) \qquad (w_{n+1} \stackrel{\text{def}}{=} \mathrm{EOS}). \tag{6b}$$

Likewise, when we include the next-symbol prediction loss term, we add a **next-symbol prediction head**. For any string $u \in \Sigma^*$, we define the set of valid next symbols $\mathrm{NEXT}_L(\cdot)$ as follows:

$$\mathrm{NEXT}_L(u) \stackrel{\text{def}}{=} \{a \in \Sigma \mid \exists v \in \Sigma^*, \, uav \in L\} \cup \{\mathrm{EOS} \mid u \in L\}. \tag{7}$$

We use binary cross-entropy to train the model to discern whether each symbol $a \in \Sigma_\mathrm{EOS}$ is in $\mathrm{NEXT}_L(w_{<t})$, by computing the probability $p_M(a \in \mathrm{NEXT}_L(w_{<t}))$. We average cross-entropy over symbol types and timesteps to get the loss term, which we denote formally as

$$p_M(a \in \mathrm{NEXT}_L(w_{<t})) \stackrel{\text{def}}{=} \sigma(\boldsymbol{W}_\mathrm{NS}\boldsymbol{h}_{t-1} + \boldsymbol{b}_\mathrm{NS})_a \qquad (1 \leq t \leq n+1; a \in \Sigma_\mathrm{EOS}) \tag{8}$$

---

[6]This is often called next character prediction in prior work.

$$\mathcal{L}_{\mathrm{NS}}(M, w) \stackrel{\text{def}}{=} \frac{1}{n+1} \sum_{t=1}^{n+1} \frac{1}{|\Sigma_{\mathrm{EOS}}|} \sum_{a \in \Sigma_{\mathrm{EOS}}} H_{a \in \mathrm{NEXT}_L(w_{<t})}(p_M(a \in \mathrm{NEXT}_L(w_{<t}))), \qquad (9)$$

where $\boldsymbol{W}_{\mathrm{NS}} \in \mathbb{R}^{|\Sigma_{\mathrm{EOS}}| \times d}$ and $\boldsymbol{b}_{\mathrm{NS}} \in \mathbb{R}^{|\Sigma_{\mathrm{EOS}}|}$ are learned parameters. We say that at each timestep $t$, the model predicts the set $\{a \in \Sigma_{\mathrm{EOS}} \mid p_M(a \in \mathrm{NEXT}_L(w_{<t})) \geq \frac{1}{2}\}$. Unlike language modeling, next-symbol prediction adds additional information to the training data, as $\mathrm{NEXT}_L(w_{<t})$ can include information about unobserved strings. For trim partial DFAs, we can easily precompute the set of valid next symbols for each state based on the outgoing transitions (Algorithm 6). We use language-specific rules for non-regular languages (App. E). The full loss function with all terms is

$$\mathcal{L}(M, w) \stackrel{\text{def}}{=} \mathcal{L}_{\mathrm{R}}(M, w) + \lambda_{\mathrm{LM}} \mathcal{L}_{\mathrm{LM}}(M, w) + \lambda_{\mathrm{NS}} \mathcal{L}_{\mathrm{NS}}(M, w), \qquad (10)$$

where $\lambda_{\mathrm{LM}}, \lambda_{\mathrm{NS}} \in \mathbb{R}_{\geq 0}$ are coefficients that determine the importance of each auxiliary loss term. Whenever an auxiliary loss term is not included, this is equivalent to setting $\lambda_{\mathrm{LM}}$ or $\lambda_{\mathrm{NS}}$ to 0. Note that we never include them for negative examples. When computing the loss for a whole minibatch of examples, we average the loss $\mathcal{L}(M, w)$ of the individual examples.

### 3.4 COMPARISON TO PRIOR WORK

Many past papers have used neural networks for language recognition, but each uses a slightly different methodology. Our work is an attempt to unify this methodology, to extend it to the broadest possible class of languages, to compare different training objectives, and to spur theoretically sound empirical work in the future. MLRegTest (van der Poel et al., 2024) proposes a recognition setup that is specific to regular languages, whereas our method generalizes to any language that has tractable algorithms for length-constrained positive sampling and membership testing, and whose complement is not small. MLRegTest samples efficiently from regular languages with small complements by constructing a PDFA for the complement directly, but this technique is not available for many higher language classes. As for positive sampling from DFAs, they sample *uniformly* from the set of strings of length $n$, using an intersection-based approach similar to that described in §3.1.2. Their preprocessing step runs in $O(|\Sigma||Q|n_{\max}(n_{\max} - n_{\min}) \log(|Q|n_{\max}))$ time (App. F), whereas ours runs in $O(|Q|^3 n_{\max}^2 + |\delta| n_{\max}^2)$ time. Weiss et al. (2018a) also employ a perturbation-based negative sampling method that edits positive samples up to 9 times; we allow an unlimited number of edits and provide a way of analyzing the ground-truth edit distance distribution in §5, Figure 2.

## 4 EXPERIMENTS

We test the performance of the three architectures in §3.2 on a variety of formal languages (Table 1) that have proven to be of particular interest in prior work, and that come from various levels of the Chomsky hierarchy. These include analogs of the transduction tasks used by Delétang et al. (2023). For each regular language, we generate datasets using a hand-crafted DFA and the algorithms described in §3.1.2. For the other languages, we use language-specific algorithms for length-constrained sampling and membership testing. We describe all languages in more detail in App. E. We call this collection of datasets **FLaRe** (**F**ormal **La**nguage **Re**cognition).

For each language, we sample a single, fixed training set of 10k examples with lengths in $[0, 40]$. We run separate experiments with two different validation sets that are designed to address different scientific questions, each with 1k examples: a **short validation set** with string lengths in $[0, 40]$, and a **long validation set** with string lengths in $[0, 80]$. For any finite training set of strings, there are infinitely many valid ways of extrapolating to longer strings; we refer to the way that a neural network architecture disambiguates these possible solutions as its **inductive bias**. The short validation set reveals an architecture's inductive bias in the absence of any disambiguating information about how it should generalize to longer lengths, as in the experiments of Delétang et al. (2023). The long validation set, on the other hand, does include longer strings, and so the model's performance on strings that are longer than those in the training set modulates the learning rate schedule, early stopping schedule, and model selection. In this way, it is more in line with expressivity work that seeks to understand whether an architecture admits a certain solution at all, regardless of its inductive bias.

We use 5 layers for all models in all experiments. We automatically adjust the hidden vector size $d$ so that the number of parameters is as close as possible to 64k, excluding extra parameters for language

Table 1: Formal languages tested in this paper and included in FLaRe. For each language, we show the language class that it belongs to: regular (**R**), deterministic context-free (**DCF**), context-free (**CF**), or context-sensitive (**CS**). Each language does not belong to the previous language classes. Let $c_u(w)$ be the number of times substring $u$ occurs in $w$, let $w_{i \to a}$ be $w$ with its $i^{\text{th}}$ symbol replaced with $a$, and let $\langle x \rangle$ be the little-endian binary encoding of $x \in \mathbb{Z}_{\geq 0}$. See App. E for details.

| Class | Language | Description | Example String |
|---|---|---|---|
| **R** | Even Pairs | $\{w \in \{0,1\}^* \mid c_{01}(w) + c_{10}(w) \text{ is even}\}$ $= \{aua \mid a \in \{0,1\}, u \in \{0,1\}^*\} \cup \{\varepsilon, 0, 1\}$ | `010110` |
| | Repeat 01 | $\{(01)^n \mid n \geq 0\}$ | `010101` |
| | Parity | $\{w \in \{0,1\}^* \mid c_1(w) \text{ is odd}\}$ | `11011001` |
| | Cycle Navigation | A sequence of left (<), right (>), stay (=) moves on a 5-position cycle, then the final position (0-indexed). | `>>=<>2` |
| | Modular Arithmetic | Expression involving $\{+, -, \times\}$ and $\{0, \ldots, 4\}$, then the result mod 5. No operator precedence. | `1-3×2=1` |
| | Dyck-$(2, 3)$ | Strings of balanced brackets with 2 bracket types and a maximum depth of 3. | `[()([])]()` |
| | First | $\{1w \mid w \in \{0,1\}^*\}$ | `100010` |
| **DCF** | Majority | $\{w \in \{0,1\}^* \mid c_1(w) > c_0(w)\}$ | `101101` |
| | Stack Manipulation | A stack from bottom to top, a sequence of push and pop operations, and the resulting stack from top to bottom. | `011 POP =10` |
| | Marked Reversal | $\{w\#w^R \mid w \in \{0,1\}^*\}$ | `001#100` |
| **CF** | Unmarked Reversal | $\{ww^R \mid w \in \{0,1\}^*\}$ | `001100` |
| **CS** | Marked Copy | $\{w\#w \mid w \in \{0,1\}^*\}$ | `001#001` |
| | Missing Duplicate | $\{(ww)_{i \to \_} \mid w \in \{0,1\}^*, 1 \leq i \leq 2|w|, (ww)_i = 1\}$ | `1_011101` |
| | Odds First | $\{a_1 b_1 \cdots a_n b_n a \# a_1 \cdots a_n a b_1 \cdots b_n \mid$ $n \geq 0; a_i, b_i \in \{0,1\}; a \in \{0,1,\varepsilon\}\}$ | `01010=00011` |
| | Binary Addition | $\{\langle x \rangle 0^i + \langle y \rangle 0^j = \langle x+y \rangle 0^k \mid x, y, i, j, k \in \mathbb{Z}_{\geq 0}\}$ | `110+01=10100` |
| | Binary Multiplication | $\{\langle x \rangle 0^i \times \langle y \rangle 0^j = \langle xy \rangle 0^k \mid x, y, i, j, k \in \mathbb{Z}_{\geq 0}\}$ | `110×0100=011` |
| | Compute Sqrt | $\{\langle x \rangle 0^i = \langle \lfloor \sqrt{x} \rfloor \rangle 0^j \mid x, i, j \in \mathbb{Z}_{\geq 0}\}$ | `01010=1100` |
| | Bucket Sort | Sequence of integers in $\{1, \ldots, 5\}$, then $\#$ and the sorted sequence. | `45134#13445` |

modeling and next-symbol prediction heads. This ensures that all models are of comparable size across architectures and languages. For the simple RNN and LSTM, $d$ is the size of the hidden state vectors. For the transformer, $d$ is the model size $d_{\text{model}}$. Every time we train a model, we randomly sample certain hyperparameters (Bergstra & Bengio, 2012), namely initial learning rate, minibatch size, and (when required) $\lambda_{\text{LM}}$ and $\lambda_{\text{NS}}$. For every combination of architecture, loss function variant, and type of validation set, we train 10 models. We use four loss function variants: recognition with(out) language modeling and with(out) next-symbol prediction. Therefore, for every architecture and validation set, we train 40 models. See App. D for details.

## 5 RESULTS

To test whether a model has learned the underlying recognition algorithm, our evaluation focuses on the ability to generalize to inputs that are longer than those seen in training (cf. Delétang et al., 2023). In Table 2, we show recognition accuracy on a test set with string lengths in $[0, 500]$. It has 5,010 examples, or an average of 10 examples per length. Under "Inductive Bias," we select the loss function with the highest mean accuracy on the test set from among the models trained on the short validation set, and we report this mean accuracy. Although we could select a single model based on performance on the validation set, this would result in noise due to the vagaries of model selection; we find that the mean accuracy that each architecture converges to when aggregated across multiple runs is more informative. Under "Expressivity," we show the maximum test accuracy across all 40 models trained on the long validation set. See unabridged results and additional metrics in App. G.

We find that the RNN and LSTM outperform the transformer in most cases. In the inductive bias experiments, the transformer outperforms the RNN and LSTM only on Even Pairs, First, and Major-

Table 2: Accuracy on a test set with strings in the length range $[0, 500]$. The training data is in the length range $[0, 40]$. "Inductive Bias" uses validation data in the length range $[0, 40]$. We show mean accuracy and standard deviation across 10 runs for the loss function with the highest mean accuracy on the test set. "Expressivity" uses validation data in the length range $[0, 80]$. We show maximum accuracy across all 10 runs of all 4 loss functions. "Tf" = transformer, "RNN" = simple RNN, "LSTM" = LSTM. Best scores among all three architectures are in **bold**.

| Language | Inductive Bias | | | Expressivity | | |
|---|---|---|---|---|---|---|
| | Tf | RNN | LSTM | Tf | RNN | LSTM |
| Even Pairs | $\mathbf{0.99}_{\pm 0.01}$ | $0.60_{\pm 0.20}$ | $0.83_{\pm 0.22}$ | **1.00** | **1.00** | **1.00** |
| Repeat 01 | $0.72_{\pm 0.09}$ | $0.97_{\pm 0.07}$ | $\mathbf{0.97}_{\pm 0.07}$ | 0.86 | **1.00** | **1.00** |
| Parity | $0.56_{\pm 0.03}$ | $0.71_{\pm 0.24}$ | $\mathbf{0.90}_{\pm 0.20}$ | 0.60 | **1.00** | **1.00** |
| Cycle Navigation | $0.84_{\pm 0.05}$ | $\mathbf{0.93}_{\pm 0.01}$ | $0.90_{\pm 0.04}$ | **0.93** | 0.93 | 0.93 |
| Modular Arithmetic | $0.69_{\pm 0.11}$ | $\mathbf{1.00}_{\pm 0.00}$ | $0.98_{\pm 0.03}$ | 0.88 | **1.00** | **1.00** |
| Dyck-$(2, 3)$ | $0.70_{\pm 0.09}$ | $\mathbf{0.95}_{\pm 0.05}$ | $0.91_{\pm 0.10}$ | 0.82 | **1.00** | 0.98 |
| First | $\mathbf{0.98}_{\pm 0.04}$ | $0.80_{\pm 0.24}$ | $0.94_{\pm 0.14}$ | **1.00** | **1.00** | **1.00** |
| Majority | $\mathbf{0.97}_{\pm 0.04}$ | $0.90_{\pm 0.03}$ | $0.95_{\pm 0.04}$ | 1.00 | 0.95 | **1.00** |
| Stack Manipulation | $0.66_{\pm 0.14}$ | $\mathbf{0.84}_{\pm 0.16}$ | $0.75_{\pm 0.17}$ | 0.87 | **0.93** | 0.91 |
| Marked Reversal | $0.64_{\pm 0.12}$ | $0.70_{\pm 0.18}$ | $\mathbf{0.74}_{\pm 0.17}$ | 0.87 | **0.95** | 0.95 |
| Unmarked Reversal | $0.58_{\pm 0.03}$ | $0.72_{\pm 0.08}$ | $\mathbf{0.76}_{\pm 0.01}$ | 0.63 | 0.81 | **0.88** |
| Marked Copy | $0.63_{\pm 0.11}$ | $\mathbf{0.76}_{\pm 0.15}$ | $0.69_{\pm 0.15}$ | 0.86 | **0.95** | **0.95** |
| Missing Duplicate | $0.66_{\pm 0.08}$ | $0.82_{\pm 0.10}$ | $\mathbf{0.85}_{\pm 0.07}$ | 0.86 | **0.95** | 0.94 |
| Odds First | $0.59_{\pm 0.11}$ | $\mathbf{0.79}_{\pm 0.15}$ | $0.67_{\pm 0.14}$ | 0.86 | 0.95 | **0.96** |
| Binary Addition | $0.64_{\pm 0.13}$ | $0.74_{\pm 0.12}$ | $\mathbf{0.74}_{\pm 0.12}$ | 0.88 | 0.92 | **0.92** |
| Binary Multiplication | $0.70_{\pm 0.11}$ | $0.74_{\pm 0.13}$ | $\mathbf{0.78}_{\pm 0.12}$ | **0.92** | 0.92 | 0.92 |
| Compute Sqrt | $0.67_{\pm 0.10}$ | $0.78_{\pm 0.12}$ | $\mathbf{0.84}_{\pm 0.07}$ | 0.86 | **0.89** | **0.89** |
| Bucket Sort | $0.63_{\pm 0.08}$ | $\mathbf{0.84}_{\pm 0.09}$ | $0.69_{\pm 0.13}$ | 0.88 | **0.96** | 0.83 |

ity. In the expressivity experiments, it is never the best (except by a hair on Binary Multiplication), and it only outperforms the LSTM on Bucket Sort. In terms of languages that the architectures can solve perfectly in the expressivity experiments, all architectures are limited to regular languages and Majority (Figure 1). The transformer almost reaches, but does not quite reach, 100% accuracy on Majority. We do see differences based on a language's sensitivity, i.e., the tendency that changing a bit in a string changes its label. Transformers struggle on high-sensitivity languages (Parity) and do well on low-sensitivity languages (Even Pairs, First), in accordance with prior work (Hahn, 2020; Bhattamishra et al., 2023; Hahn & Rofin, 2024). The LSTM outperforms the RNN on some languages that involve long-range dependencies (Even Pairs, First), likely thanks to its memory cell. The LSTM can use its memory cell as a set of counters, which is useful for certain languages (Weiss et al., 2018b); accordingly, we see that it outperforms the RNN on Majority, but surprisingly not on Bucket Sort. Although Yao et al. (2021) gave a construction showing that the transformer can recognize bounded Dyck languages, we see that it struggles on Dyck-$(2, 3)$.

Although it is possible, in principle, for a model's inductive bias to differ substantially from its expressivity, we see a remarkable consistency between inductive bias and expressivity. As expected, all expressivity scores are higher than the corresponding inductive bias score. However, the ranking of the architectures remains the same between inductive bias and expressivity for almost all languages; it only changes slightly for Missing Duplicate and Odds First, and more noticeably for Bucket Sort.

Certain auxiliary loss terms do help in isolated cases: for example, next-symbol prediction is crucial for the RNN to learn Even Pairs (Table 4). On the other hand, they are sometimes detrimental; for example, language modeling results in lower accuracy for the LSTM on Parity (Table 6). However, no term has a consistently positive or negative impact across languages and architectures (Table 3). In fact, recognition alone is the most frequent best loss function, followed by next-symbol prediction; loss functions that include language modeling help the least often. Remarkably, the RNN gets a mean accuracy of 100% in the inductive bias experiments for Modular Arithmetic, using only a recognition training objective.

We see accuracy ceilings in the expressivity results for Cycle Navigation, Binary Addition, Binary Multiplication, and Compute Sqrt. To investigate this, we looked at the examples with the highest cross-entropy. For Cycle Navigation, all architectures struggle on negative examples that have the

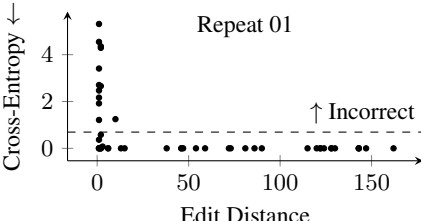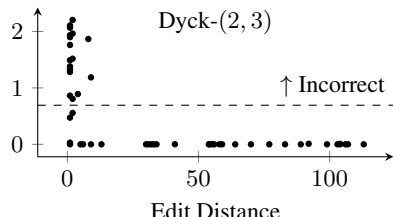

Figure 2: Recognition cross-entropy (lower is better) as a function of edit distance for the transformer model shown under "Expressivity" in Table 2, on a separate dataset of 50 negative examples in the length range $[0, 500]$. The dashed lines show $\log 2$, the threshold for incorrect predictions. Despite being trained on a large proportion of negative examples with low edit distance, the transformer still struggles on examples that resemble positive examples.

right format but the wrong digit at the end;[7] in contrast, the RNN and LSTM got perfect accuracy on a related task in Delétang et al. (2023). On Binary Addition, the transformer and RNN/LSTM fail for different reasons. The RNN/LSTM only misclassify negative examples that have the right format but incorrect arithmetic, but the transformer also misclassifies positive examples and negative examples that have extra +, ×, or = symbols. The RNN and LSTM fail on Compute Sqrt for dissimilar reasons; the RNN sometimes accepts invalid formats. Using larger models does not solve the problem; in most cases, accuracy does not improve with model size (App. H).

How does using recognizers instead of string-to-string functions affect our findings? The most comparable sets of experiments are those of Delétang et al. (2023, Tables B.1 and B.5) and our inductive bias experiments. Whereas they found that transformers struggle on Even Pairs and the RNN/LSTM excel, we see the opposite. We both show that transformers struggle on Parity. Whereas they showed that the RNN/LSTM perfectly solve Cycle Navigation, our results show otherwise, perhaps because in our version, the model must additionally validate the format of the string rather than predict a single digit. We see the same model ranking on Stack Manipulation, but not on Marked Reversal, Odds First, Binary Addition, Binary Multiplication, Compute Sqrt, and Bucket Sort. Notably, the RNN overperforms in our experiments, possibly due to our use of dropout, multiple layers, and a different activation function. Whereas Delétang et al. (2023) found that accuracy often decreases sharply with length, we find that recognition cross-entropy is usually stable (App. I).

In order to examine how the similarity of negative examples to positive examples affects classification difficulty, for Repeat 01 and Dyck-$(2, 3)$, we plot the transformer's recognition cross-entropy vs. the minimum **edit distance** between a string and any string in the language in Figure 2 (we do not plot the RNN and LSTM because they get almost 100% accuracy). Note that the number of edits $K$ performed in §3.1 is an upper bound for, but not necessarily equal to, the true edit distance. We give formal definitions and algorithms for computing edit distance in App. J. We do see that for both languages, the lower the edit distance is, the more the transformer struggles, particularly near 1 and 2 edits. This confirms that strings with few perturbations are indeed adversarial (cf. van der Poel et al., 2024), even when the training distribution is highly skewed toward few edits.

## 6 CONCLUSION

We have proposed a general method for training neural networks as recognizers of formal languages, filling a crucial gap between formal results and experiments meant to support them. We also gave a new, efficient algorithm for length-constrained sampling of strings from regular languages. We provided results for RNNs, LSTMs, and transformers on a wide range of formal languages commonly used in prior work, showing that RNNs and LSTMs often outperform transformers. An interesting question to address in future work is why transformers perform so much better on natural language than on formal languages. We tested auxiliary training objectives that have been previously used as a proxy for recognition and found that these do not improve model performance reliably. We have publicly released our datasets as a benchmark called FLaRe (Formal Language Recognition).

---

[7]Using larger models, more adversarial examples, and more training data did not alleviate this issue.

ACKNOWLEDGMENTS

We thank Leonardo Nevali, Vésteinn Snæbjarnarson, and Paul Soulos for helpful discussions, and we thank Dakotah Lambert for answering questions about MLRegTest's length-constrained sampling algorithm. Anej Svete is supported by the ETH AI Center Doctoral Fellowship. Josef Valvoda is funded by the Nordic Programme for Interdisciplinary Research Grant 105178 and the Danish National Research Foundation Grant no. DNRF169.

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

## A    DETAILS OF PERTURBATION SAMPLING

We perform perturbation sampling as follows. Given a positive string $w \in L$ with $|w| \in [n_{\min}, n_{\max}]$, we first sample a number of edits $K$ from a geometric distribution with a success probability of $p = \frac{1}{2}$. Note that this is highly skewed toward small $K$, ensuring that strings with few edits (i.e., similar to positive examples) are well-represented in the dataset. Then, $K$ times, we randomly apply an edit as follows. Let $w'$ be the current edited version of $w$. We uniformly sample a type of edit: insertion, replacement, or deletion. We disallow insertion if it would increase $|w'|$ beyond $n_{\max}$, and deletion if it would decrease it below $n_{\min}$. We disallow replacement if $|\Sigma| = 1$. For each insertion, we uniformly sample an insertion position from $\{1, \ldots, |w'| + 1\}$ and a symbol to insert from $\Sigma$. For each replacement, we uniformly sample a replacement position $i$ from $\{1, \ldots, |w'|\}$ and a new symbol from $\Sigma \setminus \{w'_i\}$. For each deletion, we uniformly sample a deletion position from $\{1, \ldots, |w'|\}$.

## B    DETAILS OF LENGTH-CONSTRAINED SAMPLING FOR REGULAR LANGUAGES

Here, we discuss our length-constrained sampling algorithm for regular languages, and the binning semiring, in more detail.

### B.1    FURTHER EXPOSITION OF THE BINNING SEMIRING

Notice that if $\boldsymbol{u}$ is a vector whose only non-zero element is $\boldsymbol{u}_i$, and $\boldsymbol{v}$ is a vector whose only non-zero element is $\boldsymbol{v}_j$, then $\boldsymbol{u} \otimes\!\!\!\!\bigcirc \boldsymbol{v}$ is a vector with the value $\boldsymbol{u}_i \otimes \boldsymbol{v}_j$ at index $i + j$ and $\boldsymbol{0}$ elsewhere. More generally, Eq. (2b) convolves the two vectors, in effect marginalizing over all ways of reaching a count of $i$ for each $0 \leq i \leq D$.

In the case of $\mathcal{A}_D$ (§3.1.2), in which all transition and accept weights are vectors with at most one non-zero element, the weight of any path is also a vector with at most one non-zero element, whose index is equal to the number of symbols the path scans, and whose value is the product of the original transition probabilities. We combine the weights of multiple paths by adding them elementwise (Eq. (2a)).

When the base semiring is the log semiring, as in $\mathcal{A}_D$, the $\oplus$ operation can be computed in $O(D)$ time, and the $\otimes$ operation (Eq. (2b)) in $O(D^2)$ time. When the base semiring is the real semiring, the $\otimes$ operation can be computed in only $O(D \log D)$ using the fast Fourier transform, but since this results in underflow when multiplying many probabilities, we do not use it in this paper.

## B.2  CLOSED SEMIRINGS

A key component of our method is the semiring generalization of weight pushing (Mohri, 2009). Weight pushing sums over an infinite number of paths in a WDFA, which requires the ability to perform infinite summations in its semiring.

**Definition 7.** *Let* $(\mathbb{K}, \oplus, \otimes, \mathbf{0}, \mathbf{1})$ *be a semiring. Let* $a^{\otimes i} = \bigotimes_{j=1}^{i} a$, *and* $a^* = \bigoplus_{i=0}^{\infty} a^{\otimes i}$. *If* $a^*$ *is defined and in* $\mathbb{K}$ *for all* $a \in \mathbb{K}$, *we say the semiring is **closed**.*

In the real semiring, $a^* = \frac{1}{1-a}$. In the log semiring, $a^* = \log \frac{1}{1-\exp a} = -\log(1 - \exp(a))$.

If a semiring $\mathcal{W} = (\mathbb{K}, \oplus, \otimes, \mathbf{0}, \mathbf{1})$ is closed, so is any $D^{\text{th}}$-order binning semiring with respect to $\mathcal{W}$, i.e., $\mathcal{W}^D = (\mathbb{K}^{D+1}, \boldsymbol{\oplus}, \boldsymbol{\otimes}, \mathbf{0}, \mathbf{1})$. We write $\boldsymbol{v}^{\circledast} \stackrel{\text{def}}{=} \boldsymbol{\bigoplus}_{j=0}^{\infty} \boldsymbol{v}^{\otimes j}$. It has the following closed-form solution.

$$(\boldsymbol{v}^{\circledast})_i = \boldsymbol{v}_0^* \otimes \left( \mathbf{1}_i \oplus \bigoplus_{j=1}^{i} \boldsymbol{v}_j \otimes (\boldsymbol{v}^{\circledast})_{i-j} \right) \qquad (0 \leq i \leq D) \qquad (11)$$

A derivation is given by Snæbjarnarson et al. (2025). The elements $(\boldsymbol{v}^{\circledast})_i$ can be computed in order of increasing $i$. Assuming $\oplus$, $\otimes$, and $^*$ run in $O(1)$ time, the total time complexity is $O(D^2)$, since it involves $O(D)$ iterations, each of which takes $O(D)$ time.

## B.3  ALGORITHMS

Here, we describe our algorithms for sampling strings from regular languages in more detail. Given a DFA $\mathcal{A}$ and length range $[n_{\min}, n_{\max}]$, we run the following steps once and for all for each language:

1. convert $\mathcal{A}$ to a PDFA $\mathcal{A}'$, then convert $\mathcal{A}'$ to a WDFA $\mathcal{A}_D$ over the $n_{\max}{}^{\text{th}}$-order binning semiring with respect to the log semiring in $O((|Q| + |\delta|)n_{\max})$ time (Algorithm 1);

2. push the weights of $\mathcal{A}_D$ to get a new WDFA $\mathcal{A}'_D$ over the $n_{\max}{}^{\text{th}}$-order binning semiring with respect to the real semiring in $O((|Q|^3 + |\delta|)n_{\max}^2)$ time (Algorithm 2), using the backward algorithm (Algorithm 3) and Lehmann's algorithm (Algorithm 4);

3. compute the subset of valid lengths $N_{\mathcal{A}'} \subseteq [n_{\min}, n_{\max}]$ in $O(n_{\max} - n_{\min})$ time (Algorithm 5);

4. precompute the next-symbol sets for each state in $O(|Q| + |\delta|)$ time (Algorithm 6).

The total runtime of these preprocessing steps is $O(|Q|^3 n_{\max}^2 + |\delta| n_{\max}^2)$, being dominated by the weight pushing step.

Afterwards, we can sample strings as many times as desired as follows:

1. uniformly sample a length $n$ from $N_{\mathcal{A}'}$ in $O(1)$ time;

2. sample a string of length $n$ with next-symbol sets in $O(n \log |\delta|_{\text{out}})$ time, where $|\delta|_{\text{out}}$ is the maximum out-degree of the automaton (Algorithm 7).

**Algorithm 1** Convert a partial DFA $\mathcal{A}$ to a WDFA $\mathcal{A}_D$ over the $n_{\max}{}^{\text{th}}$-order binning semiring with respect to the log semiring. This implicitly creates an intermediate PDFA $\mathcal{A}'$ over the real semiring with uniform probabilities. Time complexity: $O((|Q| + |\delta|)n_{\max})$.

---

1. **def** LiftWeights($\mathcal{A} = (Q, \Sigma, \delta, q_0, F), n_{\max}$):
2.     let $\mathcal{A}_D = (Q, \Sigma, \delta', q_0, \rho)$ be a new WDFA over the $n_{\max}{}^{\text{th}}$-order binning semiring with respect to the log semiring
3.     **for** $q \in Q$ :
4.       $k \leftarrow 0$
5.       **for** $q \xrightarrow{a} r \in \delta$ :
6.         $k \leftarrow k + 1$
7.       **if** $q \in F$ :
8.         $k \leftarrow k + 1$
9.       $p \leftarrow -\log k$         $\triangleright$*Set the probability to $\frac{1}{k}$ (in log space)*
10.       **for** $q \xrightarrow{a} r \in \delta$ :
11.         add $q \xrightarrow{a/(-\infty, p, -\infty, \ldots, -\infty)} r$ to $\delta'$
12.       **if** $q \in F$ :
13.         $\rho(q) \leftarrow (p, -\infty, \ldots, -\infty)$
14.       **else**
15.         $\rho(q) \leftarrow (-\infty, \ldots, -\infty)$
16.     **return** $\mathcal{A}_D$

---

**Algorithm 2** Weight pushing on $\mathcal{A}_D$, where $\mathcal{A}_D$ is a WDFA over the $D^{\text{th}}$-order binning semiring with respect to the log semiring. Given $\mathcal{A}_D$, produce a WDFA $\mathcal{A}'_D$ over the $D^{\text{th}}$-order binning semiring with respect to the real semiring that is suitable for length-constrained sampling (Algorithm 7). Also return the allsum weight $z$, which can be used to compute the set of valid lengths (Algorithm 5). Time complexity: $O((|Q|^3 + |\delta|)D^2)$.

---

1. **def** PushWeights($\mathcal{A}_D = (Q, \Sigma, \delta, q_0, \rho)$):
2.     $\boldsymbol{\beta} \leftarrow$ Backward($\mathcal{A}_D$)
3.     let $\mathcal{A}'_D = (Q, \Sigma, \delta', q_0, \rho')$ be a new WDFA over the $D^{\text{th}}$-order binning semiring with respect to the real semiring
4.     **for** $q \in Q$ :
5.       let $T$ be an empty mapping from $\Sigma$ to $(\mathbb{R} \cup \{-\infty\})^{D+1}$
6.       **for** $q \xrightarrow{a/\boldsymbol{v}} r \in \delta$ :
7.         $T[a] \leftarrow \boldsymbol{v} \otimes \boldsymbol{\beta}[r]$
8.       $T \leftarrow \underset{a}{\text{softmax}}\ T[a, :]$    $\triangleright$*Convert log probabilities to normalized probabilities. This may safely return NaN for columns with all $-\infty$.*
9.       **for** $q \xrightarrow{a/\boldsymbol{v}} r \in \delta$ :
10.         add $q \xrightarrow{a/T[a]} r$ to $\delta'$
11.     $z \leftarrow \boldsymbol{\beta}[q_0]$
12.     **return** $(\mathcal{A}'_D, z)$

---

---

**Algorithm 3** Backward algorithm on a WDFA $\mathcal{A}$ over the closed semiring $(\mathbb{K}, \oplus, \otimes, \mathbf{0}, \mathbf{1})$. Time complexity: $O(T_\oplus |Q|^3 + T_\otimes |Q|^3 + T_* |Q| + T_\oplus |\delta|)$, where $T_\oplus$, $T_\otimes$, and $T_*$ are the runtimes of $\oplus$, $\otimes$, and $^*$, respectively.

1. **def** BACKWARD($\mathcal{A} = (Q, \Sigma, \delta, q_0, \rho)$):
2.     let $A$ be a matrix indexed by $Q \times Q$ full of $\mathbf{0}$
3.     **for** $q \xrightarrow{a/w} r \in \delta$ :
4.         $A[q, r] \leftarrow A[q, r] \oplus w$
5.     $A \leftarrow$ LEHMANN($A$)
6.     let $\boldsymbol{\beta}$ be a table indexed by $Q$
7.     **for** $q \in Q$ :
8.         $\boldsymbol{\beta}[q] \leftarrow \bigoplus_{r \in Q} A[q, r] \otimes \rho(r)$
9.     **return** $\boldsymbol{\beta}$

---

**Algorithm 4** Lehmann's algorithm for inverting matrix $A \in \mathbb{K}^{N \times N}$ in the closed semiring $(\mathbb{K}, \oplus, \otimes, \mathbf{0}, \mathbf{1})$. Time complexity: $O(T_\oplus N^3 + T_\otimes N^3 + T_* N)$, where $T_\oplus$, $T_\otimes$, and $T_*$ are the runtimes of $\oplus$, $\otimes$, and $^*$, respectively.

1. **def** LEHMANN($A$):
2.     **for** $k = 1, \ldots, N$ :
3.         let $A'$ be a $N \times N$ matrix full of $\mathbf{0}$
4.         $a \leftarrow A[k, k]^*$
5.         **for** $i = 1, \ldots, N$ :
6.             **for** $j = 1, \ldots, N$ :
7.                 $A'[i, j] \leftarrow A[i, j] \oplus (A[i, k] \otimes a \otimes A[k, j])$
8.         $A \leftarrow A'$
9.     **for** $k = 1, \ldots, N$ :
10.         $A[k, k] \leftarrow A[k, k] \oplus \mathbf{1}$
11.     **return** $A$

---

**Algorithm 5** Given an allsum weight $\boldsymbol{z} \in (\mathbb{R} \cup \{-\infty\})^{D+1}$, a minimum length $n_{\min}$, and a maximum length $n_{\max}$, where $D \geq n_{\max}$, compute the set of valid lengths $N_{\mathcal{A}'}$. The allsum weight $\boldsymbol{z}$ must be the second output of Algorithm 2. Time complexity: $O(n_{\max} - n_{\min})$.

1. **def** COMPUTEVALIDLENGTHS($\boldsymbol{z}, n_{\min}, n_{\max}$):
2.     **return** $\{n \in \{n_{\min}, \ldots, n_{\max}\} \mid \boldsymbol{z}_n > -\infty\}$

---

**Algorithm 6** Given a trim partial DFA $\mathcal{A}$, precompute the next-symbol set for each state. Time complexity: $O(|Q| + |\delta|)$.

1. **def** COMPUTENEXT($\mathcal{A} = (Q, \Sigma, \delta, q_0, F)$):
2.     let NEXT be a table indexed by $Q$
3.     **for** $q \in Q$ :
4.         NEXT$[q] \leftarrow \emptyset$
5.         **for** $q \xrightarrow{a} r \in \delta$ :
6.             add $a$ to NEXT$[q]$
7.         **if** $q \in F$ :
8.             add EOS to NEXT$[q]$
9.     **return** NEXT

---

---

**Algorithm 7** Sample a string of length $n$ from a WDFA $\mathcal{A}'_D$ over the $D^{\text{th}}$-order binning semiring with respect to the real semiring, where $D \geq n$. Also output a sequence of $n+1$ next-symbol sets. The DFA $\mathcal{A}'_D$ must be the first output of Algorithm 2, and NEXT must be the output of Algorithm 6. Time complexity: $O(n \log |\delta|_{\text{out}})$, where $|\delta|_{\text{out}}$ is the maximum out-degree of $\mathcal{A}'_D$.

---

1. **def** SAMPLE($\mathcal{A}'_D = (Q, \Sigma, \delta, q_0, \rho), n, \text{NEXT}$):
2.     $q \leftarrow q_0$
3.     $w \leftarrow \varepsilon$
4.     $s \leftarrow \{\text{NEXT}[q]\}$
5.     **for** $i = n, \ldots, 1$ :
6.         sample $(a, r) \sim p$, where $p((a,r)) = \boldsymbol{v}_i$ for $q \xrightarrow{a/\boldsymbol{v}} r \in \delta$    ▷*Log. time due to binary search.*
7.         $q \leftarrow r$
8.         $w \leftarrow wa$
9.         append NEXT$[q]$ to $s$
10.     **return** $(w, s)$

---

### B.4 EXPLANATION AND DETAILS

The log semiring is used in Algorithm 1 instead of the real semiring in order to avoid underflow.

Running a semiring generalization of weight pushing on $\mathcal{A}_D$ allows us to compute exactly the quantities we need for efficient sampling from $p_{\mathcal{A}'}(\mathrm{W} \mid |\mathrm{W}| = n)$. At every state, and for every $0 \leq i \leq D$, it computes a probability distribution over (1) outgoing transitions and (2) whether to accept, conditioned on scanning exactly $i$ symbols in the future, according to the probabilities of $\mathcal{A}'$. It does this by computing (1) for every transition, the sum of the weights of all paths in $\mathcal{A}_D$ that start with that transition, and (2) for every state, the sum of the weights of all paths in $\mathcal{A}_D$ that start and end at that state. Once we have these weights, which are vectors, if we locally normalize them elementwise at each state, we get the aforementioned probability distributions. Now, a $O(n)$-time sampling algorithm for sampling a string of exactly length $n$ becomes straightforward (Algorithm 7). We start in $q_0$ and initialize a counter $i$ to $n$. Repeatedly, we sample transitions from the normalized distributions for index $i$ from the current state and decrement $i$ for every symbol scanned. We stop when we sample an accept action (this happens implicitly in Algorithm 7 after $i = 1$, as the automaton is guaranteed by construction to accept at that point). The set of valid string lengths $N_{\mathcal{A}'}$ is the set of all $n$ for which we can take any transition or accept at $q_0$, conditioned on scanning $n$ symbols in the future, with nonzero probability (Algorithm 5).

We now discuss details of the weight pushing algorithm. Let us define a path in a WDFA as follows (cf. Def. 12).

**Definition 8.** *For any WDFA $\mathcal{A} = (Q, \Sigma, \delta, q_0, \rho)$, a **path** is a sequence of states and transitions*

$$\pi = r_0 \xrightarrow{a_1/w_1} r_1 \cdots r_{m-1} \xrightarrow{a_m/w_m} r_m \tag{12}$$

*such that for all $i = 0, \ldots, m-1$, $r_i \xrightarrow{a_{i+1}/w_{i+1}} r_{i+1} \in \delta$. We say that $\pi$ **scans** the string $a_1 \cdots a_m$ and that the **inner path weight** of $\pi$ is*

$$\mathbf{w}_\mathrm{I}(\pi) \stackrel{\text{def}}{=} \bigotimes_{i=1}^{m} w_i. \tag{13}$$

*The **path weight** of $\pi$ is*

$$\mathbf{w}(\pi) \stackrel{\text{def}}{=} \mathbf{w}_\mathrm{I}(\pi) \otimes \rho(r_m). \tag{14}$$

Lehmann's algorithm (Lehmann, 1977) computes the total inner weight of all paths between all pairs of states. When a WDFA $\mathcal{A}$ contains cycles, this set of paths can be infinite, so a closed semiring with a defined $^*$ operation is required. We give Lehmann's algorithm a table $A$ indexed by $Q \times Q$ such that

$$A[q, r] = \bigoplus_{q \xrightarrow{a/w} r \in \delta} w \qquad\qquad (q, r \in Q). \tag{15}$$

Let $\Pi(\mathcal{A}, q \rightsquigarrow r)$ denote the (infinite) set of all paths starting at $q$ and ending at $r$ in $\mathcal{A}$. Lehmann's algorithm computes a table $A'$ indexed by $Q \times Q$ where

$$A'[q, r] = \bigoplus_{\pi \in \Pi(\mathcal{A}, q \rightsquigarrow r)} \mathbf{w}_{\mathrm{I}}(\pi) \qquad (q, r \in Q). \tag{16}$$

This allows us to compute the backward weight of each state $q$, or the total weight of accepting if starting in $q$.

**Definition 9.** *For a WDFA $\mathcal{A}$, the **backward weight** $\boldsymbol{\beta}[q]$ is the sum of the weights of all paths from $q$ to any other state.*

$$\boldsymbol{\beta}[q] \stackrel{\text{def}}{=} \bigoplus_{\substack{r \in Q, \\ \pi \in \Pi(\mathcal{A}, q \rightsquigarrow r)}} \mathbf{w}(\pi). \tag{17}$$

The quantity $\boldsymbol{z} = \boldsymbol{\beta}[q_0]$ is the total weight assigned by $\mathcal{A}$ to strings in $\Sigma^*$. We call it the **allsum**.

We use the backward weights for weight pushing. Weight pushing redistributes the weights of $\mathcal{A}_D$ so that the weight of each transition in $\mathcal{A}_D'$ is the normalized (infinite) sum of the weights of all paths in $\mathcal{A}_D$ that start with that transition (Algorithm 2, line 7). Note that unlike the standard weight pushing algorithm (Mohri, 2009), we do not "normalize" the weights by left-multiplying them by $\boldsymbol{\beta}[q]^{-1}$, where, in general, $a^{-1} \in \mathbb{K}$ is the solution to the equation $a^{-1} \otimes a = \mathbf{1}$. The reason for this is not because computing $a^{-1}$ is hard; it is possible to do so by solving a system of linear equations. The issue is that this would result in weights that add up elementwise to $\textcircled{1} = (\mathbf{1}, \mathbf{0}, \ldots, \mathbf{0})$, which would not be useful for our purposes. Instead, we normalize them by dividing them elementwise, which we do implicitly when we apply $\mathrm{softmax}$ (Algorithm 2, line 8).

In our implementation, we precompute the elementwise *cumulative sum* of the transition weights. In our Python implementation of Algorithm 7, line 6, we pass the cumulative sum to the `python.choices` function as the argument `cum_weights` in order to avoid recomputing it every time. Note that `python.choices` runs in logarithmic time in the length of `cum_weights` due to its use of binary search.

An important implementation detail is that all of the operations in Def. 6 and weight pushing are amenable to vectorization, and we take advantage of this by accelerating it with PyTorch (Paszke et al., 2019). This significantly improves the runtime of the preprocessing step.

## C   DETAILS OF NEURAL NETWORK ARCHITECTURES

In this section, we describe each of the neural network architectures referenced in §3.2 in more detail. Our implementations for all three architectures are based on those provided by PyTorch (Paszke et al., 2019). Each architecture consists of a configurable number of layers $L$. Each architecture uses an input embedding matrix $\boldsymbol{E}'$ to map each symbol $w_t$ of the input string to an embedding $\boldsymbol{x}_t = \boldsymbol{E}'_{w_t}$. The size of the embeddings is always $d$, the size of the hidden vectors. In the following, $\mathrm{DROPOUT}(\cdot)$ indicates the application of dropout.

### C.1   SIMPLE RNN

Let $\boldsymbol{h}_t^{(\ell)}$ denote the hidden state of the $\ell^{\text{th}}$ layer at timestep $t$. We apply dropout to the input embeddings, the hidden states between layers, and the hidden states output from the last layer (Zaremba et al., 2015). Our simple RNN architecture is defined as follows.

$$\boldsymbol{h}_t^{(0)} \stackrel{\text{def}}{=} \boldsymbol{x}_t = \boldsymbol{E}'_{w_t} \qquad\qquad (1 \leq t \leq n) \tag{18a}$$

$$\boldsymbol{h}_0^{(\ell)} \stackrel{\text{def}}{=} \tanh(\boldsymbol{w}_0^{(\ell)}) \qquad\qquad (1 \leq \ell \leq L) \tag{18b}$$

$$\boldsymbol{\hbar}_t^{(\ell)} \stackrel{\text{def}}{=} \mathrm{DROPOUT}(\boldsymbol{h}_t^{(\ell)}) \qquad\qquad (0 \leq \ell \leq L; 0 \leq t \leq n) \tag{18c}$$

$$\boldsymbol{h}_t^{(\ell)} \stackrel{\text{def}}{=} \tanh\left(\boldsymbol{W}_{\mathrm{h}}^{(\ell)} \begin{bmatrix} \boldsymbol{\hbar}_t^{(\ell-1)} \\ \boldsymbol{h}_{t-1}^{(\ell)} \end{bmatrix} + \boldsymbol{b}_{\mathrm{h}}^{(\ell)}\right) \qquad\qquad (1 \leq \ell \leq L; 1 \leq t \leq n) \tag{18d}$$

$$\boldsymbol{h}_t \stackrel{\text{def}}{=} \boldsymbol{\hbar}_t^{(L)} \qquad\qquad (0 \leq t \leq n) \tag{18e}$$

Here, $\boldsymbol{w}_0^{(\ell)} \in \mathbb{R}^d$ is a learned parameter, making the initial hidden state $\boldsymbol{h}_0^{(\ell)}$ of each layer learned. Note that PyTorch's RNN implementation includes redundant bias parameters $b_{ih}$ and $b_{hh}$; we have modified it to use a single bias parameter $\boldsymbol{b}_{\mathrm{h}}^{(\ell)}$ per layer instead.

## C.2   LSTM

As with the simple RNN, we apply dropout following Zaremba et al. (2015). Our LSTM architecture is defined as follows. Let $\odot$ denote elementwise multiplication.

$$\boldsymbol{h}_t^{(0)} \overset{\text{def}}{=} \boldsymbol{x}_t = \boldsymbol{E}'_{w_t} \qquad (1 \le t \le n) \tag{19a}$$

$$\boldsymbol{h}_0^{(\ell)} \overset{\text{def}}{=} \tanh(\boldsymbol{w}_0^{(\ell)}) \qquad (1 \le \ell \le L) \tag{19b}$$

$$\boldsymbol{\cancel{h}}_t^{(\ell)} \overset{\text{def}}{=} \text{DROPOUT}(\boldsymbol{h}_t^{(\ell)}) \qquad (0 \le \ell \le L; 0 \le t \le n) \tag{19c}$$

$$\boldsymbol{i}_t^{(\ell)} \overset{\text{def}}{=} \sigma\left(\boldsymbol{W}_{\mathrm{i}}^{(\ell)} \begin{bmatrix} \boldsymbol{\cancel{h}}_t^{(\ell-1)} \\ \boldsymbol{h}_{t-1}^{(\ell)} \end{bmatrix} + \boldsymbol{b}_{\mathrm{i}}^{(\ell)}\right) \qquad (1 \le \ell \le L; 1 \le t \le n) \tag{19d}$$

$$\boldsymbol{f}_t^{(\ell)} \overset{\text{def}}{=} \sigma\left(\boldsymbol{W}_{\mathrm{f}}^{(\ell)} \begin{bmatrix} \boldsymbol{\cancel{h}}_t^{(\ell-1)} \\ \boldsymbol{h}_{t-1}^{(\ell)} \end{bmatrix} + \boldsymbol{b}_{\mathrm{f}}^{(\ell)}\right) \qquad (1 \le \ell \le L; 1 \le t \le n) \tag{19e}$$

$$\boldsymbol{g}_t^{(\ell)} \overset{\text{def}}{=} \tanh\left(\boldsymbol{W}_{\mathrm{g}}^{(\ell)} \begin{bmatrix} \boldsymbol{\cancel{h}}_t^{(\ell-1)} \\ \boldsymbol{h}_{t-1}^{(\ell)} \end{bmatrix} + \boldsymbol{b}_{\mathrm{g}}^{(\ell)}\right) \qquad (1 \le \ell \le L; 1 \le t \le n) \tag{19f}$$

$$\boldsymbol{o}_t^{(\ell)} \overset{\text{def}}{=} \sigma\left(\boldsymbol{W}_{\mathrm{o}}^{(\ell)} \begin{bmatrix} \boldsymbol{\cancel{h}}_t^{(\ell-1)} \\ \boldsymbol{h}_{t-1}^{(\ell)} \end{bmatrix} + \boldsymbol{b}_{\mathrm{o}}^{(\ell)}\right) \qquad (1 \le \ell \le L; 1 \le t \le n) \tag{19g}$$

$$\boldsymbol{c}_t^{(\ell)} \overset{\text{def}}{=} \boldsymbol{f}_t^{(\ell)} \odot \boldsymbol{c}_{t-1}^{(\ell)} + \boldsymbol{i}_t^{(\ell)} \odot \boldsymbol{g}_t^{(\ell)} \qquad (1 \le \ell \le L; 1 \le t \le n) \tag{19h}$$

$$\boldsymbol{h}_t^{(\ell)} \overset{\text{def}}{=} \boldsymbol{o}_t^{(\ell)} \odot \tanh(\boldsymbol{c}_t^{(\ell)}) \qquad (1 \le \ell \le L; 1 \le t \le n) \tag{19i}$$

$$\boldsymbol{c}_0^{(\ell)} \overset{\text{def}}{=} \boldsymbol{0} \qquad (1 \le \ell \le L) \tag{19j}$$

$$\boldsymbol{h}_t \overset{\text{def}}{=} \boldsymbol{\cancel{h}}_t^{(L)} \qquad (0 \le t \le n) \tag{19k}$$

Here, $\boldsymbol{w}_0^{(\ell)} \in \mathbb{R}^d$ is a learned parameter, making the initial hidden state $\boldsymbol{h}_0^{(\ell)}$ of each layer learned. Note that PyTorch's LSTM implementation includes pairs of redundant bias parameters: $b_{ii}$ and $b_{hi}$, $b_{if}$ and $b_{hf}$, $b_{ig}$ and $b_{hg}$, and $b_{io}$ and $b_{ho}$. We have modified it so that each pair is replaced with a single bias parameter per layer.

## C.3   TRANSFORMER

We use PyTorch's transformer implementation. Following Vaswani et al. (2017), we map input symbols to vectors of size $d$ with a scaled embedding layer and add sinusoidal positional encodings. Note that Delétang et al. (2023) found that the type of positional encoding does not seem to have a large impact on the transformer's performance, whereas Ruoss et al. (2023) found that transformers with *randomized* positional encodings perform better on the same set of tasks. We use 8 attention heads in each layer, and we set the number of hidden units in each feedforward layer to $4d$. We use pre-norm instead of post-norm and apply layer norm to the output of the last layer. We use the same dropout rate throughout the transformer. We apply it in the same places as Vaswani et al. (2017), and, as implemented by PyTorch, we also apply it to the hidden units of feedforward sublayers and to the attention probabilities of scaled dot-product attention operations. We always use BOS as the first input symbol to the transformer, which has been shown to improve performance on formal languages (Ebrahimi et al., 2020).

## D   DETAILS OF EXPERIMENTS

Here, we provide additional details about the models and training procedures used in §4.

Wherever dropout is applicable, we use a dropout rate of 0.1. For the transformer, when adjusting $d$ to accommodate the parameter budget, we round it to the nearest multiple of 8, as PyTorch requires it to be a multiple of the number of attention heads. We use Xavier uniform initialization (Glorot & Bengio, 2010) to initialize the fully-connected layers in the recognition and next-symbol prediction heads. For layer norm, we initialize weights to 1 and biases to 0. We initialize all other parameters by sampling uniformly from $[-0.1, 0.1]$.

For each epoch, we randomly shuffle the training set and group strings of similar lengths into the same minibatch, enforcing an upper limit of $B$ symbols per batch, including padding, BOS, and EOS symbols. We train each model by minimizing the loss function defined in §3.3 using Adam (Kingma & Ba, 2015). We clip gradients with a threshold of 5 using $L^2$ norm rescaling. We take a checkpoint every 10k examples (i.e., at the end of each epoch), at which point we evaluate the model on the validation set and update the learning rate and early stopping schedules. We multiply the learning rate by 0.5 after 5 checkpoints of no decrease in recognition cross-entropy on the validation set, and we stop early after 10 checkpoints of no decrease. We select the checkpoint with the lowest recognition cross-entropy on the validation set when reporting results. We train for a maximum of 1k epochs.

Every time we train a model, we randomly sample a number of hyperparameters. We randomly sample the batch size $B$ from a uniform distribution over $[128, 4096]$. We randomly sample the initial learning rate from a log-uniform distribution over $[0.0001, 0.01]$. We randomly sample the loss term coefficients $\lambda_{\text{LM}}$ and $\lambda_{\text{NS}}$, when they are needed, from a log-uniform distribution over $[0.01, 10]$.

Our experiments are small enough that we are able to run them in CPU mode, without GPU acceleration.

# E    DETAILS OF LANGUAGES

Here, we give more detailed descriptions of the languages listed in Table 1. For each language, we indicate whether it is regular (**R**), deterministic context-free (**DCF**), context-free (**CF**), or context-sensitive (**CS**); this also indicates that the language does not belong to previous classes.

**Even Pairs (R).**    Binary strings where the total number of 01 and 10 substrings is even. Equivalently, this is the language of binary strings that have the same first and last symbol, or strings with fewer than two symbols. This is a low-sensitivity language, since only changing the first or last bit changes membership. This language corresponds to the Even Pairs task of Delétang et al. (2023). This language is given as an example in Sipser (2013, Chapter 1.4); see also Example 1.11.

| Positive Examples | Negative Examples |
|---|---|
| $\varepsilon$ | 01 |
| 0 | 10100 |
| 11 | 100110 |
| 010100 | |
| 11101101 | |

**Repeat 01 (R).**    The string 01 repeated any number of times. As a star-free language, it is a low-sensitivity language under the uniform distribution over $\{0, 1\}^n$ for a given string length $n$, since most strings are not in the language. However, under our distribution in which exactly half of the strings are in the language, Repeat 01 can be considered a high-sensitivity language.

| Positive Examples | Negative Examples |
|---|---|
| $\varepsilon$ | 0 |
| 01 | 10101 |
| 0101 | 011001 |

**Parity (R).**    Binary strings with an odd number of 1s. This is a high-sensitivity language, since changing any bit changes membership. It appears commonly in the theoretical literature on the representational capacity of transformers, since its high sensitivity makes it difficult for transformers to represent the language (Hahn, 2020; Chiang & Cholak, 2022; Bhattamishra et al., 2023; 2020a;

Hahn & Rofin, 2024). However, it can easily be learned by RNNs and scratchpad-augmented transformers (Liu et al., 2023; Hahn & Rofin, 2024). This language corresponds to the Parity Check task of Delétang et al. (2023).

| Positive Examples | Negative Examples |
|---|---|
| 1 | $\varepsilon$ |
| 01011 | 101110 |

**Cycle Navigation (R).** Suppose an agent is on a 5-state cycle, numbered from 0 to 4, starting at state 0. Strings in this language consist of a sequence of moves—move right (>), move left (<), or stay (=)—followed by the integer corresponding to the state reached after executing the sequence of moves. This language corresponds to the Cycle Navigation task of Delétang et al. (2023).

| Positive Examples | Negative Examples |
|---|---|
| 0 | 3 |
| >=>><2 | >=>><4 |
| <=<>=<3 | <=<>=< |
| >=>==<1 | 4=31< |

**Modular Arithmetic (R).** An expression involving the digits $\{0, \ldots, 4\}$ and the operators $\{+, -, \times\}$, then the result of evaluating that expression in modulo 5 arithmetic. All operators are left-associative infix operators with equal precedence. Note that this is different from the Modular Arithmetic (Simple) task of Delétang et al. (2023), which gives higher precedence to ×, which would result in a more complex DFA.

| Positive Examples | Negative Examples |
|---|---|
| 3=3 | $\varepsilon$ |
| 2+4+0−3=3 | 1=4 |
| 1−3×2=1 | 2+4+0−3=2 |
| | 1−3×2=0 |
| | −1=4 |
| | =×3+−0+ |

**Dyck-$(2, 3)$ (R).** In general, the language Dyck-$(k, m)$ contains strings of balanced brackets of $k$ types with a maximum nesting depth of $m$. We specifically test $k = 2$ and $m = 3$. Bounded Dyck languages have been studied for RNNs (Hewitt et al., 2020; Bhattamishra et al., 2020b) and transformers (Ebrahimi et al., 2020; Bhattamishra et al., 2020a;b; Yao et al., 2021; Wen et al., 2023) both in terms of (empirical) representational capacity as well as interpretability. These languages are star-free (Strobl et al., 2024b), and the language Dyck-$(k, m)$ has a dot-depth of $m$. In this sense, bounded Dyck languages span the (infinite) hierarchy of star-free languages, which have been closely linked to transformers (Yang et al., 2024). Bhattamishra et al. (2020a) argue that transformers struggle to learn languages beyond dot-depth 1.

| Positive Examples | Negative Examples |
|---|---|
| $\varepsilon$ | ](]))[(] |
| ([]) | ([] |
| [()] | [(]) |
| ([()]())[()] | ([(())]())[()] |
| | )][( |

**First (R).** Binary strings that start with 1. This is a low-sensitivity language, since only changing the first bit changes membership. More concretely, it is a special case of a 1-Parity language, i.e., the Parity language restricted to a single position (Hahn & Rofin, 2024). Bhattamishra et al. (2023) refer to such functions as 1-Sparse functions. Transformers have been shown to learn such sparse functions well (Edelman et al., 2022; Bhattamishra et al., 2023; Hahn & Rofin, 2024).

| Positive Examples | Negative Examples |
|---|---|
| 1 | $\varepsilon$ |
| 101110 | 0 |
| | 0111010 |

**Majority (DCF).**    Binary strings with more 1s than 0s. It is between low- and high-sensitivity, since string membership flips only when half of the bits in the string are 1. This language has been studied by Pérez et al. (2021); Merrill et al. (2022); Bhattamishra et al. (2023); Strobl (2023). Although it lies higher on the Chomsky hierarchy than the high-sensitivity languages Parity and Even Pairs, its lower sensitivity makes it easier for transformers to learn (Bhattamishra et al., 2023; Hahn & Rofin, 2024).

| Positive Examples | Negative Examples |
|---|---|
| 1 | $\varepsilon$ |
| 110 | 001 |
| 011011010 | 1100 |

We generate a positive example by first sampling a length $n$ uniformly from $[\max(n_{\min}, 1), n_{\max}]$, then a number of 1s $c_1$ uniformly from $[\lfloor n/2 \rfloor + 1, n]$. We compute the number of 0s as $c_0 = n - c_1$. We return a random permutation of the string $0^{c_0} 1^{c_1}$.

To test whether a string is in the language, we simply return whether the number of 1s is greater than the number of 0s.

To compute $\text{NEXT}_L(w_{<t})$, we always include $\{0, 1\}$, and we add EOS if $w_{<t} \in L$.

**Stack Manipulation (DCF).**    Each string starts with a binary string representing the contents of a stack, written bottom to top. Then, there is a sequence of operations to be performed on the stack, where popping is indicated with $\boxed{\text{POP}}$, pushing 0 is indicated with the string $\boxed{\text{PUSH}}$ 0, and pushing 1 is indicated with the string $\boxed{\text{PUSH}}$ 1. Popping from an empty stack is not allowed. Finally, there is a =, and the contents of the resulting stack are written top to bottom (if the stack were not reversed, this language would not be context-free). This language corresponds to the Stack Manipulation task of Delétang et al. (2023); note that their version treats popping from an empty stack as a no-op.

| Positive Examples | Negative Examples |
|---|---|
| = | $\varepsilon$ |
| 01011 $\boxed{\text{POP}}$ $\boxed{\text{PUSH}}$ 0 $\boxed{\text{PUSH}}$ 1=101010 | 01011 $\boxed{\text{POP}}$ $\boxed{\text{PUSH}}$ 0 $\boxed{\text{PUSH}}$ 1=010101 |
| 11 $\boxed{\text{POP}}$ $\boxed{\text{PUSH}}$ 0=01 | 11= $\boxed{\text{POP}}$ $\boxed{\text{PUSH}}$ =01 |
| 01 $\boxed{\text{POP}}$ $\boxed{\text{POP}}$ $\boxed{\text{PUSH}}$ 0 $\boxed{\text{PUSH}}$ 1=10 | 01 $\boxed{\text{POP}}$ $\boxed{\text{POP}}$ $\boxed{\text{POP}}$ $\boxed{\text{PUSH}}$ 0 $\boxed{\text{PUSH}}$ 1=10 |

To generate a positive example, we first sample the number of original stack symbols $n_{\text{stack}}$ and then the number of push operations $n_{\text{push}}$. Let $n_{\text{pop}}$ be the number of pop operations. Note that the length of the resulting stack is $n_{\text{stack}} + n_{\text{push}} - n_{\text{pop}}$, and the total length of the string is $n = n_{\text{stack}} + 2n_{\text{push}} + n_{\text{pop}} + 1 + n_{\text{push}} - n_{\text{pop}} = 2n_{\text{stack}} + 3n_{\text{push}} + 1$. The minimum value of $n_{\text{push}}$ is 0. So, following simple algebra, we sample $n_{\text{stack}}$ uniformly from $\left[\max(0, \lceil \frac{n_{\min}-1}{2} \rceil), \lfloor \frac{n_{\max}-1}{2} \rfloor\right]$, and then we sample $n_{\text{push}}$ uniformly from $\left[\max(0, \lceil \frac{n_{\min}-2n_{\text{stack}}-1}{3} \rceil), \lfloor \frac{n_{\max}-2n_{\text{stack}}-1}{3} \rfloor\right]$. We sample an initial stack uniformly from $\{0, 1\}^{n_{\text{stack}}}$. We then sample a sequence of operations while counting the number of pushes generated so far and simulating the stack actions. At each step, we sample an action uniformly from $\{\boxed{\text{PUSH}}, \boxed{\text{POP}}\}$. We disallow $\boxed{\text{POP}}$ if the stack is empty. If we sample $\boxed{\text{PUSH}}$, we uniformly sample a pushed symbol from $\{0, 1\}$. We stop when we sample a $\boxed{\text{PUSH}}$ and $n_{\text{push}}$ pushes have already been generated (this allows $\boxed{\text{POP}}$ to occur after the last push). We then add = and the resulting stack.

To test whether a string is in the language, we scan it from left to right while checking that it has the right format and simulating the stack actions. We push all 0s and 1s at the beginning to a stack. We scan $\boxed{\text{PUSH}}$ and $\boxed{\text{POP}}$ commands and perform them on the stack, until we scan =. We reject if $\boxed{\text{PUSH}}$ is not followed by 0 or 1, or if we attempt to pop from an empty stack. We then check that the rest of the string is equal to the resulting stack.

To compute $\text{NEXT}_L(w_{<t})$, we scan $w$ in order of increasing $t$ while simulating the stack actions, as above. If $w_{<t}$ ends within the initial stack part, we set it to $\{0, 1, \boxed{\text{POP}}, \boxed{\text{PUSH}}, =\}$. If $w_{<t}$ ends with $\boxed{\text{POP}}$, or a 0 or 1 after a $\boxed{\text{PUSH}}$, we include =, and we include $\boxed{\text{PUSH}}$ if fewer than $n_{\text{push}}$ pushes have been seen, and we include $\boxed{\text{POP}}$ if the stack is not empty. If $w_{<t}$ ends with $\boxed{\text{PUSH}}$, we set it to $\{0, 1\}$. There is only one correct string for the final stack; if $w_{<t}$ ends within the final stack part, we set it to $\{0\}$, $\{1\}$, or $\{\text{EOS}\}$ depending on what the correct string is.

**Marked Reversal (DCF).** Strings of the form $u\#u^R$, where $u$ is a binary string. This is a classic example of a deterministic context-free language (Hopcroft et al., 2006). This corresponds to the Reverse String task of Delétang et al. (2023), which explicitly marks the point when a model should stop reading $w$ and start generating $w^R$. DuSell & Chiang (2020; 2022; 2023; 2024) showed that LSTM and transformer language models struggle on a weighted version of this language compared to stack-augmented neural networks.

| Positive Examples | Negative Examples |
|---|---|
| # | $\varepsilon$ |
| 011#110 | 011#101101 |
| 0#0 | 011#11 |
| 01001#10010 | 0#11#110# |
| | 011110 |

Let $m = |u|$. To generate a positive example, we first sample $m$ uniformly from $[\lceil \max(0, \frac{n_{\min}-1}{2}) \rceil, \lfloor \frac{n_{\max}-1}{2} \rfloor]$, then we sample $u$ uniformly from $\{0, 1\}^m$. We then return the string $u\#u^R$. Notice that the string is guaranteed to be in the desired length range.

To test whether a string is in the language, we check whether there is a single #, and whether the substring after the marker is the reverse of the substring before the marker.

To compute $\text{NEXT}_L(w_{<t})$, we scan $w$ in order of increasing $t$. If $w_{<t}$ ends within the first half (before the # symbol), then the set of next valid symbols is set to $\{0, 1, \#\}$. The rest of the string is deterministic based on $w$, and we use one of $\{0\}$, $\{1\}$, or $\{\text{EOS}\}$ as needed.

**Unmarked Reversal (CF).** Strings of the form $uu^R$, where $u$ is a binary string. This is a classic example of a nondeterministic context-free language (Hopcroft et al., 2006, p. 254). DuSell & Chiang (2020; 2022; 2024) showed that LSTM and transformer language models struggle on this language compared to stack-augmented neural networks.

| Positive Examples | Negative Examples |
|---|---|
| $\varepsilon$ | 1 |
| 011110 | 01110 |
| 00 | 011100 |
| 0100110010 | 11110 |

Let $m = |u|$. To generate a positive example, we first sample $m$ uniformly from $[\lceil \frac{n_{\min}}{2} \rceil, \lfloor \frac{n_{\max}}{2} \rfloor]$, then we sample $u$ uniformly from $\{0, 1\}^m$. We then return the string $uu^R$.

To test whether a string is in the language, we check whether the length of the string is even and whether the second half is the reverse of the first half.

To compute $\text{NEXT}_L(w_{<t})$, we always include $\{0, 1\}$, and we include EOS if $w_{<t} \in L$.

**Marked Copy (CS).** Strings of the form $u\#u$, where $u$ is a binary string. This is a classic example of a mildly context-sensitive language (Joshi, 1985). This language is somewhat similar to the Duplicate String task of Delétang et al. (2023), which requires a model to read $u$ and output $uu$, which is more like the language $\{u\#uu \mid u \in \{0, 1\}^*\}$. Jelassi et al. (2024) showed both theoretically and empirically that transformers are better at copying than modern recurrent architectures, since the latter are constrained by their hidden state bottleneck. This language is also analogous to the String Equality task in Bhattamishra et al. (2024), who also found that transformers outperform both modern and classic recurrent architectures. DuSell & Chiang (2023) showed that LSTM language models struggle on a weighted version of this language compared to certain stack-augmented LSTMs.

| Positive Examples | Negative Examples |
|---|---|
| # | $\varepsilon$ |
| 011#011 | 011#01 |
| 0#0 | 011011 |
| 01001#01001 | 0##11#01#1 |

We generate positive examples, test membership, and compute $\text{NEXT}_L(w_{<t})$ similarly to Marked Reversal.

**Missing Duplicate (CS).** This language contains strings of the form $uu$, where $u$ is a binary string, but where one of the symbols in $uu$ has been replaced with _, and where the replaced symbol was a 1. This language corresponds to the Missing Duplicate task of Delétang et al. (2023), which does not explicitly mark the boundary between the two $u$s.

| Positive Examples | Negative Examples |
|---|---|
| _1 | $\varepsilon$ |
| 001000_0 | 00100_10 |
| 11_01001110100 | 11101001110100 |
| | _01_1_00 |

To generate a positive example, we sample $m = |u|$ uniformly from $[\max(1, \lceil \frac{n_{\min}}{2} \rceil), \lfloor \frac{n_{\max}}{2} \rfloor]$. To ensure that $u$ contains at least one 1, we first sample a string $u'$ uniformly from $\{0, 1\}^m$, then we uniformly at random replace one of its symbols with 1 to get $u$. We uniformly at random pick one of the 1s in $uu$ and replace it with _ to get the final string $w$.

To test whether a string is in the language, we check if its length is even and if it contains exactly one _. We replace the _ with 1 and check if the first half is the same as the second half.

To compute $\text{NEXT}_L(w_{<t})$, if $w_{<t}$ does not contain _, we set it to $\{0, 1, \_\}$. If $w_{<t}$ does contain _, we include $\{0, 1\}$, and we add EOS if $w_{<t} \in L$.

**Odds First (CS).** A binary string $u$, then #, then a string $v = u_{\text{odd}} u_{\text{even}}$, where $u_{\text{odd}}$ is all the symbols in $u$ at odd positions, and $u_{\text{even}}$ is all the symbols in $u$ at even positions. In other words, strings in this language are of the form $u' a \# u'_{\text{odd}} a u_{\text{even}}$, where $u'$ is the perfect shuffle of $u'_{\text{odd}}, u_{\text{even}} \in \{0, 1\}^*$, and $a \in \{0, 1, \varepsilon\}$. This corresponds to the Odds First task of Delétang et al. (2023).

| Positive Examples | Negative Examples |
|---|---|
| # | $\varepsilon$ |
| 1#1 | 010101#000110 |
| 010101#000111 | 010101000111 |
| 0101010#0000111 | 0#1## |
| 10011011#10110101 | |

To generate a positive example, we first sample a string $u$ as in Marked Reversal. We then return $u \# u_{\text{odd}} u_{\text{even}}$.

To test whether a string is in the language, we first check whether it contains exactly one #. We let the string to the left of # be $u$, and we check if the string to the right is equal to $u_{\text{odd}} u_{\text{even}}$.

To compute $\text{NEXT}_L(w_{<t})$, if $w_{<t}$ ends before #, we set it to $\{0, 1, \#\}$. The rest of the string is deterministic based on the value of $u$, and we use either $\{0\}$, $\{1\}$, or $\{\text{EOS}\}$ to match $u_{\text{odd}} u_{\text{even}}$.

**Binary Addition (CS).** Strings of the form $u_x$+$u_y$=$u_z$, where $u_x, u_y, u_z$ are little-endian binary encodings (possibly with trailing 0s) of integers $x, y, z \in \mathbb{Z}_{\geq 0}$, and $x + y = z$. The number 0 is encoded as 0, but not $\varepsilon$. This language corresponds to the Binary Addition task of Delétang et al. (2023).

| Positive Examples | Negative Examples |
|---|---|
| 0+0=0 | $\varepsilon$ |
| 001+1=101 | += |
| 001000+100=1010000 | 001+1=011 |
| 101+01011=11111 | 100+1=101 |
| 1+11=001 | 0011101 |
| | =0+10=1+ |

We generate a positive example as follows. Note that, in general, binary encodings must have at least one bit, and a binary string of length $m$ can only encode integers in $[0, 2^m - 1]$. Let $n_x = |u_x|, n_y = |u_y|, n_z = |u_z|$. We first sample $n_x, n_y, n_z$, then we sample $x, y, z$ that satisfy $x \leq 2^{n_x} - 1, y \leq 2^{n_y} - 1, z = x + y \leq 2^{n_z} - 1$. We sample a total string length $n$ uniformly from $[\max(5, n_{\min}), n_{\max}]$. Let $n_x = n'_x + 1, n_y = n'_y + 1, n_z = n'_z + 1$. We sample $n'_x, n'_y, n'_z$ using a Dirichlet distribution with parameters $(1, 1, 1)$ so that $n_x, n_y, n_z$ are equally distributed and always

sum to $n-2$. If $n_y > n_x$, we swap them, so that $n_x \leq n_y$; this will make the distribution over $x$ less restrictive on $y$, because it reduces cases where $x$ is so large that few $y$ can be chosen that satisfy the constraint on $z$. We sample $x$ uniformly from $[0, \min(2^{n_x} - 1, 2^{n_z} - 1)]$, and $y$ uniformly from $[0, \min(2^{n_y} - 1, 2^{n_z} - 1 - x)]$. We encode $x, y, z = x + y$ as $u_x, u_y, u_z$, padding them with 0s as needed to reach lengths of exactly $n_x, n_y, n_z$. In order to avoid bias in the distribution of $x$ vs. $y$, with probability $\frac{1}{2}$, we swap $u_x$ and $u_y$. We return $u_x$+$u_y$=$u_z$.

To test whether a string is in the language, we simply check that it has the expected format, parse $x, y, z$, and check that $x + y = z$.

To compute $\text{NEXT}_L(w_{<t})$, we scan $w$ in order of increasing $t$. Before $u_x$, we set it to $\{0, 1\}$. After any symbol in $u_x$, we set it to $\{0, 1, +\}$. Similarly, before $u_y$, we set it to $\{0, 1\}$, and after any symbol in $u_y$, we set it to $\{0, 1, =\}$. After =, we must deterministically generate $\langle z \rangle$, so we set it to $\{0\}$ or $\{1\}$ as needed. After $\langle z \rangle$, and after any trailing 0s, we set it to $\{0, \text{EOS}\}$.

**Binary Multiplication (CS).** Strings of the form $u_x$×$u_y$=$u_z$, where, like Binary Addition, $u_x, u_y, u_z$ are binary encodings of integers $x, y, z \in \mathbb{Z}_{\geq 0}$, and $xy = z$. This language corresponds to the Binary Multiplication task of Delétang et al. (2023).

| Positive Examples | Negative Examples |
|---|---|
| 0×0=0 | $\varepsilon$ |
| 001×11=0011 | ×= |
| 001000×1100=0011000 | 001×11=1011 |
| 1001×0111=0111111 | 100×1010=0101000 |
| | 0011101 |
| | =0×10=1× |

We generate a positive example similarly to Binary Addition. We first sample $n_x, n_y, n_z$, then we sample $x, y, z$ that satisfy $x \leq 2^{n_x} - 1, y \leq 2^{n_y} - 1, z = xy \leq 2^{n_z} - 1$. We sample $n, n_x, n_y, n_z$ in the same way as Binary Addition, except the Dirichlet distribution has parameters $(1, 1, 2)$. This means $n_z$ tends to be twice as big as $n_x$ or $n_y$; we do this because the number of bits required for $xy$ is approximately the sum of the bits required for $x$ and $y$. Note that guaranteeing $n_x \leq n_y$ is particularly important here for a good distribution of $y$. We sample $x$ uniformly from $[0, 2^{n_x} - 1]$. If $x > 0$, we sample $y$ uniformly from $[0, \min(2^{n_y} - 1, \lfloor \frac{2^{n_z}-1}{x} \rfloor)]$. Otherwise, we sample $y$ uniformly from $[0, 2^{n_y} - 1]$. The rest is like Binary Addition, except we return $u_x$×$u_y$=$u_z$.

We test whether a string is in the language and compute $\text{NEXT}_L(w_{<t})$ like Binary Addition, except we use $xy = z$ instead of $x + y = z$, and × instead of +.

**Compute Sqrt (CS).** Strings of the form $u_x$=$u_z$, where, similarly to Binary Addition and Binary Multiplication, $u_x, u_z$ are binary encodings of integers $x, z \in \mathbb{Z}_{\geq 0}$, and $\lfloor \sqrt{x} \rfloor = z$. This language corresponds to the Compute Sqrt task of Delétang et al. (2023).

| Positive Examples | Negative Examples |
|---|---|
| 0=0 | $\varepsilon$ |
| 011=11 | = |
| 00101=001 | 011=01 |
| 00101000=00100 | 0=11=1 |

We generate a positive example similarly to Binary Addition and Binary Multiplication. Let $n_x = |u_x|, n_z = |u_z|$. We first sample $n_x, n_z$, then we sample $x, z$ that satisfy $x \leq 2^{n_x} - 1, z = \lfloor \sqrt{x} \rfloor \leq 2^{n_z} - 1$. We sample a total string length $n$ uniformly from $[\max(3, n_{\min}), n_{\max}]$. Let $n_x = n'_x + 1, n_z = n'_z + 1$. We sample $n'_x, n'_z$ using a Dirichlet distribution with parameters $(2, 1)$, so that $n_x$ and $n_z$ sum to $n - 2$, and $n_x$ tends to be twice as big as $n_z$. We do this because the number of bits required for $\lfloor \sqrt{x} \rfloor$ is about half of that required for $x$. We sample $x$ uniformly from $[0, \min(2^{n_x} - 1, 2^{2n_z} - 1)]$. We encode $x, z = \lfloor \sqrt{x} \rfloor$ as $u_x, u_z$, padding them with 0s as needed to reach lengths of exactly $n_x, n_z$. We return $u_x$=$u_z$.

To test whether a string is in the language, we simply check that it has the expected format, parse $x, z$, and check that $\lfloor \sqrt{x} \rfloor = z$.

To compute $\text{NEXT}_L(w_{<t})$, we scan $w$ in order of increasing $t$. Before $u_x$, we set it to $\{0, 1\}$. After any symbol in $u_x$, we set it to $\{0, 1, =\}$. After =, we must deterministically generate $\langle z \rangle$, so we set it to $\{0\}$ or $\{1\}$ as needed. After $\langle z \rangle$, and after any trailing 0s, we set it to $\{0, \text{EOS}\}$.

**Bucket Sort (CS).** A string $u \in \{1, \ldots, 5\}^*$, then #, then the digits of $u$ in sorted order. Note that it is only necessary to keep track of the counts of each type of digit to recognize this language. This language corresponds to the Bucket Sort task of Delétang et al. (2023).

| Positive Examples | Negative Examples |
|---|---|
| # | $\varepsilon$ |
| 4512345#1234455 | 4512345#1434255 |
| 31204124#01122344 | 31204124#0112 |
| 41#14 | 1#2##12 |

Let $m = |u|$. To generate a positive example, we first sample $m$ as in Marked Reversal, then we sample $u$ uniformly from $\{1, \ldots, 5\}^m$. We then compute the sorted string $u'$ and return the string $u\#u'$.

To test whether a string is in the language, we check whether there is a single #, and whether the substring after the marker is the bucket sort of the substring before the marker.

To compute $\text{NEXT}_L(w_{<t})$, we scan $w$ in order of increasing $t$. If $w_{<t}$ ends within the first half (before the # symbol), then the set of next valid symbols is set to $\{1, \ldots, 5, \#\}$. The rest of the string is deterministic based on the value of $u$, and we use one of $\{1\}, \ldots, \{5\}, \{\text{EOS}\}$ as needed.

# F  LENGTH-CONSTRAINED SAMPLING FOR REGULAR LANGUAGES IN MLREGTEST

MLRegTest (van der Poel et al., 2024) assumes the availability of a complete DFA $\mathcal{A} = (Q, \Sigma, \delta, q_0, F)$ that recognizes $L$. In order to sample strings within a given length range $[n_{\min}, n_{\max}]$, it performs the following steps for each length $n$ in $[n_{\min}, n_{\max}]$.

1. Construct a DFA that recognizes $\Sigma^n$ and intersect it with $\mathcal{A}$ in $O(|\Sigma||Q|n)$ time, resulting in an acyclic DFA with $O(|Q|n)$ states.

2. Minimize the DFA in $O(|\Sigma||Q|n \log(|Q|n))$ time, resulting in $O(|Q|n)$ states.

3. Topologically sort the states, then run a dynamic programming algorithm that counts the number of accepting paths leading out of each state, all in $O(|\Sigma||Q|n)$ time. This is equivalent to running the backward algorithm in the counting semiring $(\mathbb{Z}_{\geq 0}, +, \times, 0, 1)$.

4. Use the path counts to make the DFA a PDFA encoding the uniform distribution over $L \cap \Sigma^n$.

The total runtime of this approach is $O(|\Sigma||Q|n_{\max}(n_{\max} - n_{\min}) \log(|Q|n_{\max}))$, being dominated by the minimization step. If minimization is removed, it is only $O(|\Sigma||Q|n_{\max}(n_{\max} - n_{\min}))$.

# G  FULL RESULTS

We show the best loss functions for each architecture, language, and validation set in Table 3.

We show unabridged versions of the results from §5 for all languages in Tables 4 to 21. Every row is aggregated across 10 runs. The scores shown in each row are of the model with the lowest recognition cross-entropy on the validation set (this value is shown under "Val. CE"; lower is better). All columns are accuracy scores except for "Val. CE." We show the best score in each column in **bold**.

Here, we refer to the test set used in §5, which has lengths in $[0, 500]$, as the **long test set**. We also report accuracy on a **short test set** of 1k held-out examples with lengths in $[0, 40]$. The short test set only includes examples that do not occur in the training set, short validation set, or long validation set; it tests how well a model generalizes to unseen strings within the same length distribution. For languages where the training and validation data already includes all possible strings, we leave this

Table 3: The best loss functions, corresponding to the accuracy scores reported in Table 2. "R" = recognition; "LM" = language modeling; "NS" = next-symbol prediction. No single loss function consistently results in the best performance; the most frequent winner is just R.

| Language | Inductive Bias | | | Expressivity | | |
|---|---|---|---|---|---|---|
| | Tf | RNN | LSTM | Tf | RNN | LSTM |
| Even Pairs | R | R+LM+NS | R | R | R+NS | R |
| Repeat 01 | R | R+NS | R | R | R | R |
| Parity | R+NS | R+NS | R+NS | R+NS | R+LM | R |
| Cycle Navigation | R+LM+NS | R | R | R | R | R |
| Modular Arithmetic | R | R | R | R+NS | R | R |
| Dyck-$(2,3)$ | R+LM | R+LM+NS | R | R+NS | R+NS | R+NS |
| First | R+NS | R+LM | R+LM | R | R | R |
| Majority | R+LM | R+NS | R+NS | R+LM | R+LM+NS | R+LM+NS |
| Stack Manipulation | R | R+NS | R | R+LM+NS | R | R+LM |
| Marked Reversal | R+NS | R+NS | R | R+LM | R+LM | R+LM |
| Unmarked Reversal | R | R+NS | R+NS | R | R+NS | R+NS |
| Marked Copy | R+NS | R+LM | R | R+NS | R | R |
| Missing Duplicate | R+LM+NS | R | R | R+LM+NS | R+LM+NS | R+LM |
| Odds First | R | R | R | R+LM+NS | R | R+LM+NS |
| Binary Addition | R | R+NS | R | R+LM | R+NS | R+NS |
| Binary Multiplication | R+NS | R | R+NS | R+NS | R+LM | R+NS |
| Compute Sqrt | R | R | R | R | R | R |
| Bucket Sort | R | R+LM+NS | R | R+NS | R+NS | R+LM+NS |

column blank. We also report accuracy on the training and validation sets to show how well the model fits the training data.

In order to see the effects of model selection, we also report the mean and maximum accuracy scores on the long test set across all runs, as it is often the case that the model that generalizes best to longer strings is not the one with the lowest recognition cross-entropy. Although this kind of test-set-based model selection is impossible in the wild, for our purposes it is useful for revealing when an architecture is capable of chancing upon a solution that generalizes, even if it cannot be reliably found with model selection.

In all rows, "+LM" means a language modeling loss term is added, "+NS" means a next-symbol prediction loss term is added, "S" means a short validation set is used, and "L" means a long validation set is used. "Train" is accuracy on the training set, "Val. CE" is recognition cross-entropy on the validation set (which is used as the model selection criterion), "Val." is accuracy on the validation set, "S. Test" is accuracy on the short test set, "L. Test" is accuracy on the long test set, "L. Test (Mean)" is "L. Test" averaged across runs with standard deviations, and "L. Test (Max)" is the maximum.

Table 4: Full results on the **Even Pairs** language.

| Model | Train | Val. CE ↓ | Val. | S. Test | L. Test | L. Test (Mean) | L. Test (Max) |
|---|---|---|---|---|---|---|---|
| Tf (S) | **1.000** | 0.000 | **1.000** | **1.000** | **1.000** | 0.994 ± 0.01 | **1.000** |
| Tf (L) | **1.000** | **0.000** | **1.000** | **1.000** | **1.000** | **0.999** ± 0.00 | **1.000** |
| Tf (+LM, S) | **1.000** | 0.000 | **1.000** | **1.000** | 0.997 | 0.978 ± 0.04 | **1.000** |
| Tf (+LM, L) | **1.000** | 0.000 | **1.000** | **1.000** | 0.923 | 0.956 ± 0.06 | **1.000** |
| Tf (+NS, S) | **1.000** | 0.000 | **1.000** | **1.000** | 0.998 | 0.994 ± 0.01 | **1.000** |
| Tf (+NS, L) | **1.000** | 0.000 | **1.000** | **1.000** | 1.000 | 0.996 ± 0.01 | **1.000** |
| Tf (+LM+NS, S) | **1.000** | 0.000 | **1.000** | **1.000** | 0.992 | 0.959 ± 0.06 | **1.000** |
| Tf (+LM+NS, L) | **1.000** | 0.000 | **1.000** | **1.000** | **1.000** | 0.992 ± 0.01 | **1.000** |
| RNN (S) | 0.549 | 0.673 | 0.565 | 0.513 | 0.512 | 0.504 ± 0.01 | 0.517 |
| RNN (L) | 0.542 | 0.680 | 0.527 | 0.480 | 0.509 | 0.508 ± 0.01 | 0.518 |
| RNN (+LM, S) | 0.541 | 0.674 | 0.559 | 0.520 | 0.505 | 0.506 ± 0.01 | 0.517 |
| RNN (+LM, L) | 0.576 | 0.658 | 0.607 | 0.575 | 0.640 | 0.523 ± 0.04 | 0.640 |
| RNN (+NS, S) | **1.000** | 0.000 | **1.000** | **1.000** | **1.000** | 0.556 ± 0.15 | **1.000** |
| RNN (+NS, L) | **1.000** | 0.000 | **1.000** | **1.000** | **1.000** | 0.621 ± 0.19 | **1.000** |
| RNN (+LM+NS, S) | **1.000** | 0.000 | **1.000** | **1.000** | **1.000** | 0.601 ± 0.20 | **1.000** |
| RNN (+LM+NS, L) | **1.000** | 0.000 | **1.000** | **1.000** | **1.000** | 0.614 ± 0.20 | **1.000** |
| LSTM (S) | **1.000** | 0.000 | **1.000** | **1.000** | **1.000** | 0.831 ± 0.22 | **1.000** |
| LSTM (L) | **1.000** | 0.000 | **1.000** | **1.000** | 0.761 | 0.900 ± 0.15 | **1.000** |
| LSTM (+LM, S) | **1.000** | 0.000 | **1.000** | **1.000** | **1.000** | 0.554 ± 0.15 | **1.000** |
| LSTM (+LM, L) | 0.538 | 0.680 | 0.520 | 0.518 | 0.509 | 0.503 ± 0.01 | 0.515 |
| LSTM (+NS, S) | **1.000** | 0.000 | **1.000** | **1.000** | **1.000** | 0.611 ± 0.20 | **1.000** |
| LSTM (+NS, L) | **1.000** | 0.000 | **1.000** | **1.000** | **1.000** | 0.697 ± 0.22 | **1.000** |
| LSTM (+LM+NS, S) | 0.539 | 0.671 | 0.551 | 0.508 | 0.511 | 0.504 ± 0.01 | 0.516 |
| LSTM (+LM+NS, L) | 0.537 | 0.681 | 0.513 | 0.518 | 0.497 | 0.503 ± 0.01 | 0.517 |

Table 5: Full results on the **Repeat 01** language.

| Model | Train | Val. CE ↓ | Val. | S. Test | L. Test | L. Test (Mean) | L. Test (Max) |
|---|---|---|---|---|---|---|---|
| Tf (S) | 0.999 | 0.000 | **1.000** | | 0.707 | 0.717 ± 0.09 | 0.844 |
| Tf (L) | 0.994 | 0.134 | 0.969 | | 0.593 | 0.693 ± 0.10 | 0.857 |
| Tf (+LM, S) | 0.998 | 0.001 | **1.000** | | 0.833 | 0.675 ± 0.12 | 0.847 |
| Tf (+LM, L) | 0.997 | 0.087 | 0.979 | | 0.617 | 0.687 ± 0.08 | 0.838 |
| Tf (+NS, S) | 1.000 | 0.000 | **1.000** | | 0.546 | 0.710 ± 0.10 | 0.842 |
| Tf (+NS, L) | 0.971 | 0.126 | 0.962 | | 0.845 | 0.734 ± 0.10 | 0.845 |
| Tf (+LM+NS, S) | 0.999 | 0.003 | **1.000** | | 0.681 | 0.659 ± 0.09 | 0.842 |
| Tf (+LM+NS, L) | 0.997 | 0.105 | 0.970 | | 0.592 | 0.706 ± 0.10 | 0.850 |
| RNN (S) | **1.000** | 0.000 | **1.000** | | **1.000** | 0.935 ± 0.10 | **1.000** |
| RNN (L) | **1.000** | 0.000 | **1.000** | | **1.000** | 0.880 ± 0.10 | **1.000** |
| RNN (+LM, S) | **1.000** | 0.000 | **1.000** | | **1.000** | 0.956 ± 0.09 | **1.000** |
| RNN (+LM, L) | **1.000** | 0.000 | **1.000** | | **1.000** | 0.948 ± 0.08 | **1.000** |
| RNN (+NS, S) | **1.000** | 0.000 | **1.000** | | **1.000** | 0.969 ± 0.07 | **1.000** |
| RNN (+NS, L) | **1.000** | 0.000 | **1.000** | | **1.000** | 0.978 ± 0.07 | **1.000** |
| RNN (+LM+NS, S) | **1.000** | 0.000 | **1.000** | | **1.000** | 0.911 ± 0.11 | **1.000** |
| RNN (+LM+NS, L) | **1.000** | 0.000 | **1.000** | | **1.000** | 0.914 ± 0.10 | **1.000** |
| LSTM (S) | **1.000** | **0.000** | **1.000** | | **1.000** | 0.972 ± 0.07 | **1.000** |
| LSTM (L) | **1.000** | 0.000 | **1.000** | | **1.000** | 0.905 ± 0.13 | **1.000** |
| LSTM (+LM, S) | **1.000** | 0.000 | **1.000** | | 0.864 | 0.928 ± 0.14 | **1.000** |
| LSTM (+LM, L) | **1.000** | 0.000 | **1.000** | | 0.939 | 0.975 ± 0.02 | **1.000** |
| LSTM (+NS, S) | **1.000** | 0.000 | **1.000** | | **1.000** | 0.921 ± 0.10 | **1.000** |
| LSTM (+NS, L) | **1.000** | 0.000 | **1.000** | | **1.000** | **0.982** ± 0.05 | **1.000** |
| LSTM (+LM+NS, S) | **1.000** | 0.000 | **1.000** | | **1.000** | 0.953 ± 0.13 | **1.000** |
| LSTM (+LM+NS, L) | **1.000** | 0.000 | **1.000** | | 0.964 | 0.881 ± 0.20 | **1.000** |

Table 6: Full results on the **Parity** language.

| Model | Train | Val. CE ↓ | Val. | S. Test | L. Test | L. Test (Mean) | L. Test (Max) |
|---|---|---|---|---|---|---|---|
| Tf (S) | 0.713 | 0.503 | 0.688 | 0.601 | 0.538 | $0.528 \pm 0.01$ | 0.544 |
| Tf (L) | 0.664 | 0.620 | 0.595 | 0.569 | 0.521 | $0.529 \pm 0.01$ | 0.543 |
| Tf (+LM, S) | 0.764 | 0.452 | 0.722 | 0.680 | 0.558 | $0.530 \pm 0.01$ | 0.558 |
| Tf (+LM, L) | 0.756 | 0.552 | 0.658 | 0.686 | 0.561 | $0.532 \pm 0.01$ | 0.561 |
| Tf (+NS, S) | 0.969 | 0.060 | 0.973 | 0.953 | 0.559 | $0.557 \pm 0.03$ | 0.630 |
| Tf (+NS, L) | 0.903 | 0.424 | 0.766 | 0.882 | 0.550 | $0.547 \pm 0.02$ | 0.599 |
| Tf (+LM+NS, S) | 0.909 | 0.198 | 0.895 | 0.874 | 0.579 | $0.552 \pm 0.03$ | 0.604 |
| Tf (+LM+NS, L) | 0.865 | 0.444 | 0.751 | 0.820 | 0.587 | $0.553 \pm 0.03$ | 0.591 |
| RNN (S) | 0.546 | 0.677 | 0.531 | 0.534 | 0.521 | $0.507 \pm 0.01$ | 0.525 |
| RNN (L) | 0.554 | 0.687 | 0.540 | 0.540 | 0.522 | $0.512 \pm 0.01$ | 0.541 |
| RNN (+LM, S) | **1.000** | 0.000 | **1.000** | **1.000** | **1.000** | $0.605 \pm 0.20$ | **1.000** |
| RNN (+LM, L) | **1.000** | 0.000 | **1.000** | **1.000** | **1.000** | $0.572 \pm 0.15$ | **1.000** |
| RNN (+NS, S) | **1.000** | 0.000 | **1.000** | **1.000** | **1.000** | $0.712 \pm 0.24$ | **1.000** |
| RNN (+NS, L) | **1.000** | 0.000 | **1.000** | **1.000** | **1.000** | $0.671 \pm 0.19$ | **1.000** |
| RNN (+LM+NS, S) | **1.000** | 0.000 | **1.000** | **1.000** | **1.000** | $0.706 \pm 0.24$ | **1.000** |
| RNN (+LM+NS, L) | **1.000** | 0.000 | **1.000** | **1.000** | **1.000** | $0.682 \pm 0.21$ | **1.000** |
| LSTM (S) | **1.000** | 0.000 | **1.000** | **1.000** | **1.000** | $0.664 \pm 0.22$ | **1.000** |
| LSTM (L) | **1.000** | 0.000 | **1.000** | **1.000** | **1.000** | $0.705 \pm 0.24$ | **1.000** |
| LSTM (+LM, S) | **1.000** | 0.000 | **1.000** | **1.000** | **1.000** | $0.599 \pm 0.20$ | **1.000** |
| LSTM (+LM, L) | 0.544 | 0.686 | 0.542 | 0.542 | 0.497 | $0.494 \pm 0.00$ | 0.497 |
| LSTM (+NS, S) | **1.000** | **0.000** | **1.000** | **1.000** | **1.000** | $\mathbf{0.902} \pm 0.20$ | **1.000** |
| LSTM (+NS, L) | **1.000** | 0.000 | **1.000** | **1.000** | **1.000** | $0.752 \pm 0.25$ | **1.000** |
| LSTM (+LM+NS, S) | **1.000** | 0.000 | **1.000** | **1.000** | **1.000** | $0.613 \pm 0.20$ | **1.000** |
| LSTM (+LM+NS, L) | **1.000** | 0.000 | **1.000** | **1.000** | **1.000** | $0.547 \pm 0.15$ | **1.000** |

Table 7: Full results on the **Cycle Navigation** language.

| Model | Train | Val. CE ↓ | Val. | S. Test | L. Test | L. Test (Mean) | L. Test (Max) |
|---|---|---|---|---|---|---|---|
| Tf (S) | 0.931 | 0.250 | 0.911 | 0.946 | 0.811 | $0.836 \pm 0.05$ | **0.934** |
| Tf (L) | 0.923 | 0.192 | 0.938 | 0.946 | 0.901 | $0.844 \pm 0.05$ | **0.934** |
| Tf (+LM, S) | 0.931 | 0.249 | 0.911 | 0.946 | 0.805 | $0.804 \pm 0.10$ | 0.933 |
| Tf (+LM, L) | 0.926 | 0.191 | 0.938 | 0.946 | 0.923 | $0.866 \pm 0.05$ | 0.933 |
| Tf (+NS, S) | **0.984** | **0.074** | **0.973** | **0.987** | 0.767 | $0.812 \pm 0.04$ | 0.884 |
| Tf (+NS, L) | 0.950 | 0.169 | 0.943 | 0.957 | 0.776 | $0.819 \pm 0.04$ | 0.927 |
| Tf (+LM+NS, S) | 0.931 | 0.251 | 0.911 | 0.946 | 0.810 | $0.838 \pm 0.05$ | 0.932 |
| Tf (+LM+NS, L) | 0.924 | 0.192 | 0.938 | 0.946 | 0.932 | $0.838 \pm 0.05$ | 0.932 |
| RNN (S) | 0.930 | 0.255 | 0.910 | 0.946 | **0.934** | $\mathbf{0.930} \pm 0.01$ | **0.934** |
| RNN (L) | 0.928 | 0.192 | 0.939 | 0.946 | **0.934** | $0.917 \pm 0.05$ | **0.934** |
| RNN (+LM, S) | 0.929 | 0.254 | 0.910 | 0.946 | **0.934** | $0.915 \pm 0.05$ | **0.934** |
| RNN (+LM, L) | 0.931 | 0.189 | 0.941 | 0.946 | **0.934** | $0.904 \pm 0.09$ | **0.934** |
| RNN (+NS, S) | 0.930 | 0.256 | 0.910 | 0.946 | **0.934** | $0.900 \pm 0.07$ | **0.934** |
| RNN (+NS, L) | 0.929 | 0.190 | 0.940 | 0.946 | **0.934** | $0.877 \pm 0.13$ | **0.934** |
| RNN (+LM+NS, S) | 0.928 | 0.264 | 0.907 | 0.946 | **0.934** | $0.888 \pm 0.11$ | **0.934** |
| RNN (+LM+NS, L) | 0.929 | 0.192 | 0.940 | 0.946 | **0.934** | $0.839 \pm 0.17$ | **0.934** |
| LSTM (S) | 0.929 | 0.257 | 0.910 | 0.946 | **0.934** | $0.900 \pm 0.04$ | **0.934** |
| LSTM (L) | 0.929 | 0.190 | 0.940 | 0.946 | 0.927 | $0.914 \pm 0.03$ | **0.934** |
| LSTM (+LM, S) | 0.930 | 0.252 | 0.910 | 0.946 | **0.934** | $0.878 \pm 0.13$ | **0.934** |
| LSTM (+LM, L) | 0.929 | 0.191 | 0.940 | 0.946 | **0.934** | $0.828 \pm 0.17$ | **0.934** |
| LSTM (+NS, S) | 0.931 | 0.255 | 0.911 | 0.946 | **0.934** | $0.822 \pm 0.16$ | **0.934** |
| LSTM (+NS, L) | 0.929 | 0.193 | 0.940 | 0.946 | 0.923 | $0.880 \pm 0.13$ | 0.933 |
| LSTM (+LM+NS, S) | 0.935 | 0.237 | 0.912 | 0.947 | 0.884 | $0.874 \pm 0.13$ | **0.934** |
| LSTM (+LM+NS, L) | 0.929 | 0.189 | 0.940 | 0.946 | 0.933 | $0.798 \pm 0.20$ | **0.934** |

Table 8: Full results on the **Modular Arithmetic** language.

| Model | Train | Val. CE ↓ | Val. | S. Test | L. Test | L. Test (Mean) | L. Test (Max) |
|---|---|---|---|---|---|---|---|
| Tf (S) | 0.977 | 0.093 | 0.976 | 0.979 | 0.643 | 0.686 ± 0.11 | 0.812 |
| Tf (L) | 0.978 | 0.117 | 0.963 | 0.981 | 0.829 | 0.698 ± 0.09 | 0.830 |
| Tf (+LM, S) | 0.983 | 0.085 | 0.980 | 0.980 | 0.531 | 0.659 ± 0.11 | 0.796 |
| Tf (+LM, L) | 0.929 | 0.270 | 0.889 | 0.919 | 0.559 | 0.676 ± 0.08 | 0.790 |
| Tf (+NS, S) | 0.984 | 0.085 | 0.981 | 0.981 | 0.740 | 0.654 ± 0.12 | 0.869 |
| Tf (+NS, L) | 0.975 | 0.100 | 0.969 | 0.976 | 0.884 | 0.706 ± 0.09 | 0.884 |
| Tf (+LM+NS, S) | 0.979 | 0.089 | 0.976 | 0.981 | 0.826 | 0.671 ± 0.12 | 0.852 |
| Tf (+LM+NS, L) | 0.972 | 0.122 | 0.965 | 0.973 | 0.582 | 0.612 ± 0.08 | 0.793 |
| RNN (S) | 0.986 | 0.080 | 0.982 | 0.978 | 0.997 | **0.996** ± 0.00 | **0.997** |
| RNN (L) | **0.988** | 0.062 | **0.987** | **0.984** | **0.997** | 0.989 ± 0.02 | **0.997** |
| RNN (+LM, S) | 0.987 | 0.081 | 0.982 | 0.980 | **0.997** | 0.882 ± 0.10 | **0.997** |
| RNN (+LM, L) | **0.988** | 0.062 | **0.987** | **0.984** | **0.997** | 0.964 ± 0.07 | **0.997** |
| RNN (+NS, S) | 0.987 | 0.079 | 0.982 | 0.980 | 0.997 | 0.950 ± 0.08 | **0.997** |
| RNN (+NS, L) | **0.988** | 0.062 | **0.987** | **0.984** | 0.996 | 0.965 ± 0.07 | **0.997** |
| RNN (+LM+NS, S) | 0.986 | 0.079 | 0.982 | 0.978 | 0.997 | 0.966 ± 0.07 | **0.997** |
| RNN (+LM+NS, L) | **0.988** | 0.062 | **0.987** | **0.984** | **0.997** | 0.955 ± 0.07 | **0.997** |
| LSTM (S) | 0.986 | 0.078 | 0.982 | 0.978 | 0.997 | 0.982 ± 0.03 | **0.997** |
| LSTM (L) | 0.985 | 0.063 | **0.987** | 0.978 | 0.997 | 0.955 ± 0.08 | 0.997 |
| LSTM (+LM, S) | 0.986 | 0.081 | 0.982 | 0.978 | 0.997 | 0.981 ± 0.03 | **0.997** |
| LSTM (+LM, L) | **0.988** | 0.061 | **0.987** | **0.984** | **0.997** | 0.995 ± 0.00 | **0.997** |
| LSTM (+NS, S) | 0.986 | 0.078 | 0.982 | 0.978 | 0.997 | 0.952 ± 0.09 | 0.997 |
| LSTM (+NS, L) | **0.988** | **0.061** | **0.987** | **0.984** | 0.997 | 0.957 ± 0.07 | 0.997 |
| LSTM (+LM+NS, S) | 0.986 | 0.081 | 0.982 | 0.978 | 0.964 | 0.918 ± 0.14 | **0.997** |
| LSTM (+LM+NS, L) | **0.988** | 0.062 | **0.987** | **0.984** | 0.996 | 0.950 ± 0.08 | 0.997 |

Table 9: Full results on the **Dyck-**$(2, 3)$ language.

| Model | Train | Val. CE ↓ | Val. | S. Test | L. Test | L. Test (Mean) | L. Test (Max) |
|---|---|---|---|---|---|---|---|
| Tf (S) | 0.968 | 0.119 | 0.969 | 0.968 | 0.585 | 0.630 ± 0.05 | 0.711 |
| Tf (L) | 0.965 | 0.146 | 0.953 | 0.967 | 0.804 | 0.690 ± 0.08 | 0.804 |
| Tf (+LM, S) | 0.998 | 0.010 | 0.998 | 0.994 | 0.604 | 0.702 ± 0.09 | 0.811 |
| Tf (+LM, L) | 0.979 | 0.157 | 0.950 | 0.975 | 0.599 | 0.652 ± 0.06 | 0.778 |
| Tf (+NS, S) | 0.998 | 0.002 | **1.000** | 0.995 | 0.596 | 0.622 ± 0.07 | 0.765 |
| Tf (+NS, L) | 0.964 | 0.163 | 0.941 | 0.969 | 0.756 | 0.687 ± 0.09 | 0.819 |
| Tf (+LM+NS, S) | 0.996 | 0.008 | 0.996 | 0.991 | 0.693 | 0.627 ± 0.05 | 0.734 |
| Tf (+LM+NS, L) | 0.994 | 0.123 | 0.959 | 0.989 | 0.788 | 0.693 ± 0.08 | 0.791 |
| RNN (S) | 0.972 | 0.093 | 0.976 | 0.977 | 0.980 | 0.907 ± 0.08 | 0.982 |
| RNN (L) | 0.972 | 0.092 | 0.979 | 0.976 | 0.972 | 0.929 ± 0.10 | 0.982 |
| RNN (+LM, S) | 0.999 | 0.005 | **1.000** | 0.998 | 0.991 | 0.941 ± 0.07 | 0.991 |
| RNN (+LM, L) | 0.999 | 0.014 | 0.998 | 0.998 | 0.826 | 0.945 ± 0.05 | 0.982 |
| RNN (+NS, S) | 0.999 | 0.005 | 0.999 | 0.999 | 0.957 | 0.952 ± 0.07 | 0.998 |
| RNN (+NS, L) | 0.998 | 0.014 | 0.998 | 0.997 | 0.988 | 0.923 ± 0.07 | 0.996 |
| RNN (+LM+NS, S) | **1.000** | **0.000** | **1.000** | **1.000** | **0.999** | **0.953** ± 0.05 | **0.999** |
| RNN (+LM+NS, L) | 1.000 | 0.001 | **1.000** | **1.000** | 0.991 | 0.922 ± 0.09 | 0.996 |
| LSTM (S) | 0.971 | 0.097 | 0.975 | 0.978 | 0.982 | 0.907 ± 0.10 | 0.982 |
| LSTM (L) | 0.971 | 0.094 | 0.979 | 0.978 | 0.977 | 0.883 ± 0.10 | 0.977 |
| LSTM (+LM, S) | 0.971 | 0.100 | 0.975 | 0.978 | 0.982 | 0.835 ± 0.14 | 0.982 |
| LSTM (+LM, L) | 0.972 | 0.089 | 0.979 | 0.976 | 0.701 | 0.868 ± 0.10 | 0.980 |
| LSTM (+NS, S) | 0.972 | 0.097 | 0.975 | 0.977 | 0.982 | 0.882 ± 0.16 | 0.982 |
| LSTM (+NS, L) | 0.972 | 0.091 | 0.979 | 0.975 | 0.976 | 0.942 ± 0.07 | 0.983 |
| LSTM (+LM+NS, S) | 0.971 | 0.098 | 0.974 | 0.977 | 0.962 | 0.838 ± 0.19 | 0.983 |
| LSTM (+LM+NS, L) | 0.972 | 0.089 | 0.979 | 0.977 | 0.975 | 0.922 ± 0.10 | 0.982 |

Table 10: Full results on the **First** language.

| Model | Train | Val. CE ↓ | Val. | S. Test | L. Test | L. Test (Mean) | L. Test (Max) |
|---|---|---|---|---|---|---|---|
| Tf (S) | **1.000** | 0.000 | **1.000** | **1.000** | **1.000** | $0.926 \pm 0.12$ | **1.000** |
| Tf (L) | **1.000** | 0.000 | **1.000** | **1.000** | 0.987 | $0.936 \pm 0.10$ | **1.000** |
| Tf (+LM, S) | **1.000** | 0.000 | **1.000** | **1.000** | **1.000** | $0.934 \pm 0.08$ | **1.000** |
| Tf (+LM, L) | **1.000** | 0.000 | **1.000** | **1.000** | **1.000** | $0.974 \pm 0.04$ | **1.000** |
| Tf (+NS, S) | **1.000** | 0.000 | **1.000** | **1.000** | 0.952 | $0.982 \pm 0.04$ | **1.000** |
| Tf (+NS, L) | **1.000** | 0.000 | **1.000** | **1.000** | **1.000** | $0.969 \pm 0.08$ | **1.000** |
| Tf (+LM+NS, S) | **1.000** | 0.000 | **1.000** | **1.000** | 0.999 | $0.935 \pm 0.14$ | **1.000** |
| Tf (+LM+NS, L) | **1.000** | 0.000 | **1.000** | **1.000** | **1.000** | $0.928 \pm 0.12$ | **1.000** |
| RNN (S) | **1.000** | 0.000 | **1.000** | **1.000** | **1.000** | $0.629 \pm 0.20$ | **1.000** |
| RNN (L) | **1.000** | 0.000 | **1.000** | **1.000** | **1.000** | $0.701 \pm 0.24$ | **1.000** |
| RNN (+LM, S) | **1.000** | 0.000 | **1.000** | **1.000** | **1.000** | $0.800 \pm 0.24$ | **1.000** |
| RNN (+LM, L) | **1.000** | 0.000 | **1.000** | **1.000** | **1.000** | $0.800 \pm 0.24$ | **1.000** |
| RNN (+NS, S) | **1.000** | 0.000 | **1.000** | **1.000** | **1.000** | $0.703 \pm 0.24$ | **1.000** |
| RNN (+NS, L) | **1.000** | 0.000 | **1.000** | **1.000** | **1.000** | $0.799 \pm 0.25$ | **1.000** |
| RNN (+LM+NS, S) | **1.000** | 0.000 | **1.000** | **1.000** | **1.000** | $0.750 \pm 0.25$ | **1.000** |
| RNN (+LM+NS, L) | **1.000** | 0.000 | **1.000** | **1.000** | **1.000** | $0.704 \pm 0.24$ | **1.000** |
| LSTM (S) | **1.000** | 0.000 | **1.000** | **1.000** | 1.000 | $0.932 \pm 0.15$ | **1.000** |
| LSTM (L) | **1.000** | **0.000** | **1.000** | **1.000** | **1.000** | $\mathbf{1.000} \pm 0.00$ | **1.000** |
| LSTM (+LM, S) | **1.000** | 0.000 | **1.000** | **1.000** | **1.000** | $0.938 \pm 0.14$ | **1.000** |
| LSTM (+LM, L) | **1.000** | 0.000 | **1.000** | **1.000** | **1.000** | $0.896 \pm 0.15$ | **1.000** |
| LSTM (+NS, S) | **1.000** | 0.000 | **1.000** | **1.000** | **1.000** | $0.850 \pm 0.23$ | **1.000** |
| LSTM (+NS, L) | **1.000** | 0.000 | **1.000** | **1.000** | **1.000** | $0.944 \pm 0.09$ | **1.000** |
| LSTM (+LM+NS, S) | **1.000** | 0.000 | **1.000** | **1.000** | **1.000** | $0.850 \pm 0.23$ | **1.000** |
| LSTM (+LM+NS, L) | **1.000** | 0.000 | **1.000** | **1.000** | 0.779 | $0.912 \pm 0.15$ | **1.000** |

Table 11: Full results on the **Majority** language.

| Model | Train | Val. CE ↓ | Val. | S. Test | L. Test | L. Test (Mean) | L. Test (Max) |
|---|---|---|---|---|---|---|---|
| Tf (S) | **1.000** | 0.001 | **1.000** | **1.000** | 0.970 | $0.946 \pm 0.07$ | 0.990 |
| Tf (L) | **1.000** | 0.004 | **1.000** | **1.000** | 0.985 | $0.943 \pm 0.07$ | 0.995 |
| Tf (+LM, S) | **1.000** | 0.001 | **1.000** | **1.000** | 0.992 | $0.969 \pm 0.04$ | 0.992 |
| Tf (+LM, L) | **1.000** | 0.005 | **1.000** | **1.000** | 0.991 | $0.959 \pm 0.05$ | 0.996 |
| Tf (+NS, S) | **1.000** | 0.000 | **1.000** | **1.000** | 0.992 | $0.961 \pm 0.05$ | 0.992 |
| Tf (+NS, L) | **1.000** | 0.005 | 0.998 | **1.000** | 0.985 | $0.966 \pm 0.04$ | 0.993 |
| Tf (+LM+NS, S) | **1.000** | 0.001 | **1.000** | **1.000** | 0.989 | $0.969 \pm 0.02$ | 0.989 |
| Tf (+LM+NS, L) | **1.000** | 0.007 | 0.998 | **1.000** | 0.991 | $\mathbf{0.975} \pm 0.04$ | 0.993 |
| RNN (S) | **1.000** | 0.003 | **1.000** | **1.000** | 0.913 | $0.868 \pm 0.12$ | 0.928 |
| RNN (L) | **1.000** | 0.025 | 0.997 | **1.000** | 0.926 | $0.834 \pm 0.14$ | 0.931 |
| RNN (+LM, S) | **1.000** | 0.000 | **1.000** | **1.000** | 0.918 | $0.875 \pm 0.12$ | 0.924 |
| RNN (+LM, L) | **1.000** | 0.021 | 0.996 | **1.000** | 0.916 | $0.880 \pm 0.13$ | 0.945 |
| RNN (+NS, S) | 0.999 | 0.005 | 0.999 | 0.998 | 0.899 | $0.896 \pm 0.03$ | 0.927 |
| RNN (+NS, L) | 0.999 | 0.030 | 0.994 | 0.998 | 0.934 | $0.916 \pm 0.01$ | 0.934 |
| RNN (+LM+NS, S) | **1.000** | 0.006 | 0.999 | **1.000** | 0.919 | $0.887 \pm 0.07$ | 0.927 |
| RNN (+LM+NS, L) | **1.000** | 0.021 | 0.997 | **1.000** | 0.927 | $0.877 \pm 0.13$ | 0.949 |
| LSTM (S) | **1.000** | 0.000 | **1.000** | **1.000** | 0.979 | $0.939 \pm 0.04$ | 0.997 |
| LSTM (L) | **1.000** | 0.000 | **1.000** | **1.000** | 0.973 | $0.943 \pm 0.02$ | 0.989 |
| LSTM (+LM, S) | **1.000** | 0.000 | **1.000** | **1.000** | 0.980 | $0.942 \pm 0.04$ | 0.998 |
| LSTM (+LM, L) | **1.000** | 0.000 | **1.000** | **1.000** | 0.990 | $0.946 \pm 0.04$ | 0.995 |
| LSTM (+NS, S) | **1.000** | 0.000 | **1.000** | **1.000** | 0.999 | $0.952 \pm 0.04$ | 0.999 |
| LSTM (+NS, L) | **1.000** | 0.000 | **1.000** | **1.000** | 0.989 | $0.862 \pm 0.18$ | 0.997 |
| LSTM (+LM+NS, S) | **1.000** | **0.000** | **1.000** | **1.000** | **1.000** | $0.913 \pm 0.14$ | **1.000** |
| LSTM (+LM+NS, L) | **1.000** | 0.000 | **1.000** | **1.000** | 0.991 | $0.963 \pm 0.04$ | 1.000 |

Table 12: Full results on the **Stack Manipulation** language.

| Model | Train | Val. CE ↓ | Val. | S. Test | L. Test | L. Test (Mean) | L. Test (Max) |
|---|---|---|---|---|---|---|---|
| Tf (S) | 0.929 | 0.190 | 0.917 | 0.912 | 0.610 | $0.664 \pm 0.14$ | 0.869 |
| Tf (L) | 0.920 | 0.294 | 0.885 | 0.897 | 0.573 | $0.640 \pm 0.13$ | 0.868 |
| Tf (+LM, S) | 0.981 | 0.072 | 0.983 | 0.980 | 0.515 | $0.595 \pm 0.13$ | 0.854 |
| Tf (+LM, L) | 0.935 | 0.238 | 0.909 | 0.922 | 0.869 | $0.701 \pm 0.14$ | 0.869 |
| Tf (+NS, S) | 0.976 | 0.101 | 0.973 | 0.976 | 0.857 | $0.591 \pm 0.14$ | 0.868 |
| Tf (+NS, L) | 0.928 | 0.255 | 0.906 | 0.914 | 0.545 | $0.684 \pm 0.14$ | 0.869 |
| Tf (+LM+NS, S) | 0.981 | 0.059 | 0.985 | 0.979 | 0.515 | $0.571 \pm 0.11$ | 0.868 |
| Tf (+LM+NS, L) | 0.950 | 0.246 | 0.914 | 0.939 | 0.870 | $0.608 \pm 0.13$ | 0.870 |
| RNN (S) | 0.926 | 0.229 | 0.923 | 0.919 | 0.928 | $0.834 \pm 0.12$ | 0.929 |
| RNN (L) | 0.943 | 0.187 | 0.940 | 0.944 | 0.933 | $0.783 \pm 0.17$ | 0.933 |
| RNN (+LM, S) | 0.985 | 0.070 | 0.986 | 0.983 | 0.520 | $0.691 \pm 0.18$ | 0.930 |
| RNN (+LM, L) | 0.925 | 0.211 | 0.927 | 0.919 | 0.930 | $0.817 \pm 0.12$ | 0.930 |
| RNN (+NS, S) | 0.986 | 0.073 | 0.986 | 0.989 | 0.520 | $0.838 \pm 0.16$ | 0.930 |
| RNN (+NS, L) | 0.930 | 0.208 | 0.931 | 0.919 | 0.927 | $\mathbf{0.854} \pm 0.12$ | 0.930 |
| RNN (+LM+NS, S) | 0.988 | 0.050 | **0.991** | 0.990 | 0.523 | $0.762 \pm 0.20$ | 0.930 |
| RNN (+LM+NS, L) | 0.935 | 0.206 | 0.934 | 0.937 | 0.932 | $0.852 \pm 0.11$ | 0.932 |
| LSTM (S) | 0.989 | 0.047 | **0.991** | 0.993 | **0.987** | $0.746 \pm 0.17$ | **0.987** |
| LSTM (L) | 0.956 | 0.189 | 0.936 | 0.951 | 0.876 | $0.784 \pm 0.07$ | 0.885 |
| LSTM (+LM, S) | 0.987 | 0.045 | **0.991** | 0.992 | 0.551 | $0.568 \pm 0.09$ | 0.818 |
| LSTM (+LM, L) | 0.989 | 0.069 | 0.986 | 0.990 | 0.540 | $0.674 \pm 0.14$ | 0.906 |
| LSTM (+NS, S) | 0.987 | 0.051 | 0.988 | 0.992 | 0.664 | $0.669 \pm 0.14$ | 0.922 |
| LSTM (+NS, L) | 0.989 | 0.063 | 0.986 | **0.993** | 0.750 | $0.674 \pm 0.10$ | 0.799 |
| LSTM (+LM+NS, S) | **0.989** | **0.043** | **0.991** | **0.993** | 0.602 | $0.528 \pm 0.03$ | 0.602 |
| LSTM (+LM+NS, L) | 0.988 | 0.066 | 0.985 | **0.993** | 0.541 | $0.620 \pm 0.13$ | 0.896 |

Table 13: Full results on the **Marked Reversal** language.

| Model | Train | Val. CE ↓ | Val. | S. Test | L. Test | L. Test (Mean) | L. Test (Max) |
|---|---|---|---|---|---|---|---|
| Tf (S) | 0.969 | 0.116 | 0.969 | 0.967 | 0.552 | $0.604 \pm 0.11$ | 0.847 |
| Tf (L) | 0.876 | 0.316 | 0.879 | 0.877 | 0.616 | $0.693 \pm 0.08$ | 0.758 |
| Tf (+LM, S) | 0.969 | 0.117 | 0.969 | 0.967 | 0.534 | $0.577 \pm 0.08$ | 0.740 |
| Tf (+LM, L) | 0.960 | 0.197 | 0.943 | 0.953 | 0.870 | $0.689 \pm 0.13$ | 0.870 |
| Tf (+NS, S) | 0.995 | 0.032 | 0.992 | 0.991 | 0.541 | $0.641 \pm 0.12$ | 0.827 |
| Tf (+NS, L) | 0.969 | 0.176 | 0.944 | 0.967 | 0.868 | $0.647 \pm 0.12$ | 0.868 |
| Tf (+LM+NS, S) | 0.993 | 0.027 | 0.995 | **0.994** | 0.539 | $0.580 \pm 0.10$ | 0.859 |
| Tf (+LM+NS, L) | 0.966 | 0.170 | 0.941 | 0.962 | 0.853 | $0.665 \pm 0.09$ | 0.853 |
| RNN (S) | 0.968 | 0.119 | 0.969 | 0.965 | 0.603 | $0.693 \pm 0.17$ | 0.947 |
| RNN (L) | 0.960 | 0.131 | 0.965 | 0.961 | 0.594 | $0.761 \pm 0.15$ | 0.947 |
| RNN (+LM, S) | 0.969 | 0.119 | 0.969 | 0.967 | 0.535 | $0.662 \pm 0.14$ | 0.911 |
| RNN (+LM, L) | 0.959 | 0.130 | 0.965 | 0.960 | 0.948 | $\mathbf{0.793} \pm 0.11$ | 0.948 |
| RNN (+NS, S) | 0.967 | 0.124 | 0.969 | 0.965 | 0.544 | $0.699 \pm 0.18$ | 0.948 |
| RNN (+NS, L) | 0.968 | 0.102 | 0.975 | 0.966 | 0.643 | $0.612 \pm 0.12$ | 0.947 |
| RNN (+LM+NS, S) | 0.968 | 0.119 | 0.969 | 0.967 | 0.538 | $0.651 \pm 0.17$ | 0.947 |
| RNN (+LM+NS, L) | 0.955 | 0.136 | 0.963 | 0.958 | 0.947 | $0.660 \pm 0.14$ | 0.947 |
| LSTM (S) | 0.969 | 0.116 | 0.969 | 0.967 | **0.954** | $0.744 \pm 0.17$ | **0.963** |
| LSTM (L) | 0.952 | 0.163 | 0.950 | 0.946 | 0.574 | $0.755 \pm 0.13$ | 0.859 |
| LSTM (+LM, S) | **0.999** | **0.001** | **1.000** | 0.992 | 0.584 | $0.552 \pm 0.05$ | 0.679 |
| LSTM (+LM, L) | 0.969 | 0.103 | 0.975 | 0.967 | 0.596 | $0.681 \pm 0.16$ | 0.947 |
| LSTM (+NS, S) | 0.969 | 0.117 | 0.969 | 0.967 | 0.561 | $0.653 \pm 0.12$ | 0.845 |
| LSTM (+NS, L) | 0.969 | 0.107 | 0.975 | 0.967 | 0.578 | $0.623 \pm 0.10$ | 0.852 |
| LSTM (+LM+NS, S) | 0.969 | 0.116 | 0.969 | 0.967 | 0.617 | $0.624 \pm 0.11$ | 0.946 |
| LSTM (+LM+NS, L) | 0.981 | 0.073 | 0.981 | 0.980 | 0.593 | $0.585 \pm 0.05$ | 0.674 |

Table 14: Full results on the **Unmarked Reversal** language.

| Model | Train | Val. CE ↓ | Val. | S. Test | L. Test | L. Test (Mean) | L. Test (Max) |
|---|---|---|---|---|---|---|---|
| Tf (S) | 0.970 | 0.099 | 0.971 | 0.969 | 0.552 | 0.584 ± 0.03 | 0.632 |
| Tf (L) | 0.818 | 0.454 | 0.774 | 0.839 | 0.583 | 0.600 ± 0.04 | 0.631 |
| Tf (+LM, S) | 0.955 | 0.147 | 0.952 | 0.958 | 0.563 | 0.552 ± 0.04 | 0.617 |
| Tf (+LM, L) | 0.695 | 0.558 | 0.674 | 0.677 | 0.559 | 0.544 ± 0.04 | 0.623 |
| Tf (+NS, S) | 0.963 | 0.112 | 0.965 | 0.949 | 0.541 | 0.557 ± 0.02 | 0.612 |
| Tf (+NS, L) | 0.868 | 0.456 | 0.779 | 0.846 | 0.547 | 0.578 ± 0.04 | 0.630 |
| Tf (+LM+NS, S) | 0.959 | 0.127 | 0.957 | 0.940 | 0.540 | 0.539 ± 0.03 | 0.578 |
| Tf (+LM+NS, L) | 0.752 | 0.529 | 0.726 | 0.748 | 0.607 | 0.589 ± 0.04 | 0.630 |
| RNN (S) | 0.975 | 0.094 | 0.974 | 0.977 | 0.670 | 0.708 ± 0.09 | 0.763 |
| RNN (L) | 0.970 | 0.139 | 0.965 | 0.962 | 0.726 | 0.736 ± 0.05 | 0.763 |
| RNN (+LM, S) | 0.973 | **0.020** | 0.982 | **0.998** | 0.622 | 0.715 ± 0.05 | 0.763 |
| RNN (+LM, L) | 0.981 | 0.097 | 0.982 | 0.984 | 0.748 | 0.721 ± 0.05 | 0.763 |
| RNN (+NS, S) | 0.976 | 0.097 | 0.971 | 0.968 | 0.682 | 0.716 ± 0.08 | 0.766 |
| RNN (+NS, L) | 0.974 | 0.158 | 0.957 | 0.963 | 0.665 | 0.744 ± 0.05 | 0.813 |
| RNN (+LM+NS, S) | 0.961 | 0.061 | 0.973 | 0.984 | 0.636 | 0.712 ± 0.07 | 0.763 |
| RNN (+LM+NS, L) | 0.971 | 0.150 | 0.954 | 0.961 | 0.765 | 0.744 ± 0.04 | 0.765 |
| LSTM (S) | 0.963 | 0.150 | 0.954 | 0.944 | 0.610 | 0.685 ± 0.09 | 0.763 |
| LSTM (L) | **0.989** | 0.056 | **0.988** | 0.987 | 0.746 | 0.717 ± 0.08 | 0.763 |
| LSTM (+LM, S) | 0.786 | 0.440 | 0.774 | 0.751 | 0.763 | 0.711 ± 0.10 | 0.763 |
| LSTM (+LM, L) | 0.786 | 0.444 | 0.782 | 0.751 | 0.763 | 0.711 ± 0.10 | 0.763 |
| LSTM (+NS, S) | 0.966 | 0.145 | 0.953 | 0.934 | 0.743 | **0.761** ± 0.01 | 0.763 |
| LSTM (+NS, L) | 0.987 | 0.068 | 0.982 | 0.985 | **0.838** | 0.731 ± 0.12 | **0.884** |
| LSTM (+LM+NS, S) | 0.786 | 0.441 | 0.774 | 0.751 | 0.763 | 0.711 ± 0.10 | 0.763 |
| LSTM (+LM+NS, L) | 0.786 | 0.444 | 0.782 | 0.751 | 0.763 | 0.711 ± 0.10 | 0.763 |

Table 15: Full results on the **Marked Copy** language.

| Model | Train | Val. CE ↓ | Val. | S. Test | L. Test | L. Test (Mean) | L. Test (Max) |
|---|---|---|---|---|---|---|---|
| Tf (S) | 0.970 | 0.113 | 0.970 | 0.977 | 0.548 | 0.565 ± 0.04 | 0.639 |
| Tf (L) | 0.859 | 0.327 | 0.866 | 0.872 | 0.795 | 0.698 ± 0.07 | 0.795 |
| Tf (+LM, S) | 0.970 | 0.114 | 0.970 | 0.977 | 0.542 | 0.614 ± 0.10 | 0.846 |
| Tf (+LM, L) | 0.949 | 0.206 | 0.937 | 0.955 | 0.625 | 0.646 ± 0.08 | 0.785 |
| Tf (+NS, S) | 0.991 | 0.035 | **0.992** | 0.991 | 0.588 | 0.627 ± 0.11 | 0.872 |
| Tf (+NS, L) | 0.950 | 0.213 | 0.922 | 0.950 | 0.843 | 0.693 ± 0.14 | 0.856 |
| Tf (+LM+NS, S) | 0.990 | **0.025** | 0.991 | 0.992 | 0.550 | 0.573 ± 0.08 | 0.827 |
| Tf (+LM+NS, L) | 0.963 | 0.226 | 0.944 | 0.970 | 0.573 | 0.612 ± 0.08 | 0.775 |
| RNN (S) | 0.970 | 0.118 | 0.970 | 0.977 | 0.602 | 0.723 ± 0.15 | 0.920 |
| RNN (L) | 0.958 | 0.134 | 0.964 | 0.965 | 0.946 | 0.710 ± 0.14 | 0.946 |
| RNN (+LM, S) | 0.970 | 0.115 | 0.970 | 0.977 | 0.545 | 0.756 ± 0.15 | 0.945 |
| RNN (+LM, L) | 0.959 | 0.135 | 0.965 | 0.969 | 0.595 | 0.730 ± 0.15 | 0.945 |
| RNN (+NS, S) | 0.970 | 0.115 | 0.970 | 0.977 | 0.677 | 0.755 ± 0.14 | 0.908 |
| RNN (+NS, L) | 0.960 | 0.133 | 0.965 | 0.967 | **0.946** | **0.757** ± 0.13 | 0.946 |
| RNN (+LM+NS, S) | 0.970 | 0.117 | 0.970 | 0.976 | 0.598 | 0.730 ± 0.15 | 0.946 |
| RNN (+LM+NS, L) | 0.961 | 0.127 | 0.966 | 0.968 | 0.591 | 0.728 ± 0.14 | 0.945 |
| LSTM (S) | 0.970 | 0.113 | 0.970 | 0.977 | 0.564 | 0.691 ± 0.15 | 0.946 |
| LSTM (L) | 0.970 | 0.099 | 0.976 | 0.977 | 0.649 | 0.717 ± 0.12 | 0.946 |
| LSTM (+LM, S) | 0.970 | 0.113 | 0.970 | 0.977 | 0.568 | 0.582 ± 0.07 | 0.781 |
| LSTM (+LM, L) | 0.970 | 0.098 | 0.976 | 0.977 | 0.596 | 0.656 ± 0.09 | 0.807 |
| LSTM (+NS, S) | **0.995** | 0.041 | 0.990 | **0.994** | 0.568 | 0.609 ± 0.12 | **0.962** |
| LSTM (+NS, L) | 0.970 | 0.098 | 0.976 | 0.977 | 0.721 | 0.664 ± 0.11 | 0.923 |
| LSTM (+LM+NS, S) | 0.970 | 0.113 | 0.970 | 0.977 | 0.631 | 0.604 ± 0.09 | 0.849 |
| LSTM (+LM+NS, L) | 0.970 | 0.098 | 0.976 | 0.977 | 0.657 | 0.600 ± 0.07 | 0.760 |

Table 16: Full results on the **Missing Duplicate** language.

| Model | Train | Val. CE ↓ | Val. | S. Test | L. Test | L. Test (Mean) | L. Test (Max) |
|---|---|---|---|---|---|---|---|
| Tf (S) | 0.938 | 0.200 | 0.933 | 0.949 | 0.614 | 0.650 ± 0.09 | 0.781 |
| Tf (L) | 0.856 | 0.316 | 0.876 | 0.874 | 0.857 | 0.709 ± 0.10 | 0.857 |
| Tf (+LM, S) | 0.880 | 0.311 | 0.871 | 0.890 | 0.638 | 0.608 ± 0.05 | 0.687 |
| Tf (+LM, L) | 0.856 | 0.315 | 0.876 | 0.874 | 0.782 | 0.699 ± 0.09 | 0.796 |
| Tf (+NS, S) | 0.953 | 0.169 | 0.943 | 0.952 | 0.720 | 0.637 ± 0.10 | 0.857 |
| Tf (+NS, L) | 0.860 | 0.317 | 0.875 | 0.874 | 0.777 | 0.651 ± 0.07 | 0.777 |
| Tf (+LM+NS, S) | 0.955 | 0.166 | 0.948 | 0.956 | 0.696 | 0.661 ± 0.08 | 0.760 |
| Tf (+LM+NS, L) | 0.864 | 0.315 | 0.880 | 0.874 | 0.858 | 0.728 ± 0.11 | 0.858 |
| RNN (S) | 0.946 | 0.176 | 0.944 | 0.949 | 0.919 | 0.824 ± 0.10 | 0.944 |
| RNN (L) | 0.951 | 0.163 | 0.953 | 0.957 | 0.783 | 0.815 ± 0.07 | 0.937 |
| RNN (+LM, S) | 0.951 | 0.178 | 0.946 | 0.957 | 0.945 | 0.789 ± 0.10 | 0.945 |
| RNN (+LM, L) | 0.951 | 0.163 | 0.953 | 0.957 | 0.945 | 0.860 ± 0.07 | 0.945 |
| RNN (+NS, S) | 0.959 | 0.162 | 0.952 | 0.959 | **0.946** | 0.804 ± 0.14 | **0.946** |
| RNN (+NS, L) | **0.982** | **0.116** | **0.967** | **0.970** | 0.942 | **0.879** ± 0.10 | 0.945 |
| RNN (+LM+NS, S) | 0.950 | 0.179 | 0.946 | 0.957 | 0.945 | 0.790 ± 0.13 | 0.945 |
| RNN (+LM+NS, L) | 0.951 | 0.162 | 0.954 | 0.957 | 0.696 | 0.813 ± 0.15 | 0.946 |
| LSTM (S) | 0.951 | 0.178 | 0.946 | 0.957 | 0.926 | 0.852 ± 0.07 | 0.945 |
| LSTM (L) | 0.951 | 0.162 | 0.953 | 0.957 | 0.922 | 0.794 ± 0.07 | 0.922 |
| LSTM (+LM, S) | 0.951 | 0.179 | 0.946 | 0.957 | 0.945 | 0.795 ± 0.15 | 0.945 |
| LSTM (+LM, L) | 0.951 | 0.161 | 0.953 | 0.957 | 0.771 | 0.769 ± 0.12 | 0.945 |
| LSTM (+NS, S) | 0.958 | 0.163 | 0.951 | 0.957 | 0.946 | 0.821 ± 0.15 | 0.946 |
| LSTM (+NS, L) | 0.951 | 0.162 | 0.953 | 0.957 | 0.913 | 0.784 ± 0.13 | 0.945 |
| LSTM (+LM+NS, S) | 0.954 | 0.174 | 0.947 | 0.957 | 0.940 | 0.770 ± 0.18 | 0.945 |
| LSTM (+LM+NS, L) | 0.951 | 0.162 | 0.953 | 0.957 | 0.904 | 0.749 ± 0.15 | 0.904 |

Table 17: Full results on the **Odds First** language.

| Model | Train | Val. CE ↓ | Val. | S. Test | L. Test | L. Test (Mean) | L. Test (Max) |
|---|---|---|---|---|---|---|---|
| Tf (S) | 0.966 | 0.122 | 0.968 | 0.966 | 0.537 | 0.592 ± 0.11 | 0.851 |
| Tf (L) | 0.835 | 0.345 | 0.853 | 0.849 | 0.850 | 0.730 ± 0.09 | 0.850 |
| Tf (+LM, S) | 0.969 | 0.116 | 0.969 | 0.968 | 0.536 | 0.559 ± 0.07 | 0.754 |
| Tf (+LM, L) | 0.958 | 0.207 | 0.924 | 0.959 | 0.561 | 0.613 ± 0.09 | 0.763 |
| Tf (+NS, S) | 0.993 | 0.048 | 0.991 | 0.993 | 0.545 | 0.537 ± 0.00 | 0.545 |
| Tf (+NS, L) | 0.941 | 0.236 | 0.919 | 0.944 | 0.861 | 0.724 ± 0.11 | 0.861 |
| Tf (+LM+NS, S) | 0.995 | 0.043 | 0.991 | **0.998** | 0.538 | 0.579 ± 0.08 | 0.799 |
| Tf (+LM+NS, L) | 0.955 | 0.182 | 0.937 | 0.952 | 0.864 | 0.693 ± 0.11 | 0.864 |
| RNN (S) | 0.966 | 0.119 | 0.968 | 0.966 | 0.544 | 0.793 ± 0.15 | 0.948 |
| RNN (L) | 0.957 | 0.137 | 0.962 | 0.959 | 0.948 | **0.795** ± 0.12 | 0.948 |
| RNN (+LM, S) | 0.968 | 0.119 | 0.969 | 0.968 | 0.542 | 0.708 ± 0.14 | 0.948 |
| RNN (+LM, L) | 0.959 | 0.137 | 0.963 | 0.959 | 0.948 | 0.766 ± 0.15 | 0.948 |
| RNN (+NS, S) | 0.959 | 0.138 | 0.963 | 0.959 | 0.948 | 0.789 ± 0.18 | 0.948 |
| RNN (+NS, L) | 0.967 | 0.116 | 0.972 | 0.967 | 0.594 | 0.712 ± 0.15 | 0.948 |
| RNN (+LM+NS, S) | 0.969 | 0.117 | 0.969 | 0.968 | 0.536 | 0.575 ± 0.08 | 0.777 |
| RNN (+LM+NS, L) | 0.957 | 0.138 | 0.961 | 0.959 | 0.669 | 0.758 ± 0.14 | 0.947 |
| LSTM (S) | 0.969 | 0.116 | 0.969 | 0.968 | 0.572 | 0.668 ± 0.14 | 0.921 |
| LSTM (L) | 0.969 | 0.102 | 0.974 | 0.968 | 0.838 | 0.718 ± 0.15 | 0.914 |
| LSTM (+LM, S) | 0.969 | 0.115 | 0.969 | 0.968 | 0.617 | 0.612 ± 0.10 | 0.857 |
| LSTM (+LM, L) | 0.969 | 0.103 | 0.974 | 0.968 | 0.701 | 0.687 ± 0.10 | 0.853 |
| LSTM (+NS, S) | **0.997** | **0.014** | **0.997** | 0.997 | 0.575 | 0.595 ± 0.13 | 0.936 |
| LSTM (+NS, L) | 0.969 | 0.104 | 0.974 | 0.968 | 0.611 | 0.673 ± 0.12 | 0.906 |
| LSTM (+LM+NS, S) | 0.997 | 0.027 | 0.995 | 0.993 | 0.565 | 0.597 ± 0.09 | 0.848 |
| LSTM (+LM+NS, L) | 0.969 | 0.103 | 0.974 | 0.968 | **0.964** | 0.696 ± 0.13 | **0.964** |

Table 18: Full results on the **Binary Addition** language.

| Model | Train | Val. CE ↓ | Val. | S. Test | L. Test | L. Test (Mean) | L. Test (Max) |
|---|---|---|---|---|---|---|---|
| Tf (S) | 0.925 | 0.235 | 0.919 | 0.922 | 0.673 | $0.643 \pm 0.13$ | 0.913 |
| Tf (L) | 0.915 | 0.262 | 0.902 | 0.915 | 0.645 | $0.654 \pm 0.09$ | 0.778 |
| Tf (+LM, S) | 0.974 | 0.107 | 0.963 | 0.962 | 0.562 | $0.602 \pm 0.10$ | 0.913 |
| Tf (+LM, L) | 0.933 | 0.228 | 0.917 | 0.925 | 0.744 | $0.713 \pm 0.09$ | 0.876 |
| Tf (+NS, S) | 0.976 | 0.084 | 0.977 | 0.968 | 0.632 | $0.629 \pm 0.13$ | 0.914 |
| Tf (+NS, L) | 0.924 | 0.248 | 0.910 | 0.924 | 0.831 | $0.692 \pm 0.10$ | 0.831 |
| Tf (+LM+NS, S) | 0.975 | 0.094 | 0.968 | 0.966 | 0.577 | $0.615 \pm 0.11$ | 0.896 |
| Tf (+LM+NS, L) | 0.967 | 0.154 | 0.944 | 0.960 | 0.850 | $0.684 \pm 0.11$ | 0.854 |
| RNN (S) | 0.924 | 0.227 | 0.922 | 0.921 | 0.549 | $0.719 \pm 0.10$ | 0.879 |
| RNN (L) | 0.927 | 0.236 | 0.915 | 0.924 | 0.598 | $\mathbf{0.813} \pm 0.08$ | 0.878 |
| RNN (+LM, S) | 0.943 | 0.194 | 0.931 | 0.940 | 0.566 | $0.718 \pm 0.15$ | 0.893 |
| RNN (+LM, L) | 0.929 | 0.235 | 0.916 | 0.925 | 0.822 | $0.703 \pm 0.14$ | 0.914 |
| RNN (+NS, S) | 0.988 | 0.085 | 0.979 | 0.981 | 0.604 | $0.743 \pm 0.12$ | 0.915 |
| RNN (+NS, L) | 0.930 | 0.226 | 0.914 | 0.928 | 0.731 | $0.790 \pm 0.12$ | 0.915 |
| RNN (+LM+NS, S) | 0.978 | 0.094 | 0.974 | 0.968 | 0.604 | $0.675 \pm 0.10$ | 0.898 |
| RNN (+LM+NS, L) | 0.939 | 0.214 | 0.923 | 0.934 | 0.903 | $0.762 \pm 0.12$ | 0.914 |
| LSTM (S) | 0.929 | 0.220 | 0.925 | 0.925 | 0.595 | $0.745 \pm 0.12$ | **0.916** |
| LSTM (L) | 0.929 | 0.233 | 0.916 | 0.925 | 0.651 | $0.783 \pm 0.09$ | 0.915 |
| LSTM (+LM, S) | 0.989 | 0.081 | 0.979 | 0.969 | 0.685 | $0.702 \pm 0.12$ | 0.914 |
| LSTM (+LM, L) | 0.951 | 0.210 | 0.923 | 0.952 | 0.692 | $0.780 \pm 0.09$ | 0.915 |
| LSTM (+NS, S) | 0.988 | 0.083 | **0.981** | 0.976 | 0.648 | $0.709 \pm 0.13$ | 0.902 |
| LSTM (+NS, L) | 0.929 | 0.233 | 0.916 | 0.925 | **0.916** | $0.750 \pm 0.14$ | **0.916** |
| LSTM (+LM+NS, S) | **0.992** | **0.074** | **0.981** | **0.982** | 0.581 | $0.715 \pm 0.12$ | 0.866 |
| LSTM (+LM+NS, L) | 0.986 | 0.115 | 0.966 | 0.978 | 0.820 | $0.786 \pm 0.11$ | **0.916** |

Table 19: Full results on the **Binary Multiplication** language.

| Model | Train | Val. CE ↓ | Val. | S. Test | L. Test | L. Test (Mean) | L. Test (Max) |
|---|---|---|---|---|---|---|---|
| Tf (S) | 0.959 | 0.118 | 0.958 | 0.952 | 0.563 | $0.616 \pm 0.12$ | 0.889 |
| Tf (L) | 0.933 | 0.216 | 0.916 | 0.928 | 0.720 | $0.705 \pm 0.09$ | 0.821 |
| Tf (+LM, S) | 0.973 | 0.082 | 0.974 | 0.968 | 0.627 | $0.573 \pm 0.03$ | 0.627 |
| Tf (+LM, L) | 0.932 | 0.255 | 0.899 | 0.931 | 0.582 | $0.688 \pm 0.10$ | 0.807 |
| Tf (+NS, S) | 0.975 | 0.071 | 0.977 | 0.958 | 0.760 | $0.700 \pm 0.11$ | 0.922 |
| Tf (+NS, L) | 0.952 | 0.165 | 0.941 | 0.949 | 0.723 | $0.703 \pm 0.11$ | **0.923** |
| Tf (+LM+NS, S) | 0.983 | 0.054 | 0.979 | 0.970 | 0.617 | $0.642 \pm 0.12$ | 0.884 |
| Tf (+LM+NS, L) | 0.972 | 0.169 | 0.933 | 0.961 | 0.615 | $0.637 \pm 0.07$ | 0.787 |
| RNN (S) | 0.937 | 0.168 | 0.944 | 0.930 | 0.606 | $0.745 \pm 0.13$ | 0.919 |
| RNN (L) | 0.936 | 0.194 | 0.934 | 0.931 | 0.889 | $\mathbf{0.801} \pm 0.07$ | 0.893 |
| RNN (+LM, S) | 0.980 | 0.063 | 0.980 | 0.969 | 0.604 | $0.672 \pm 0.13$ | 0.897 |
| RNN (+LM, L) | 0.936 | 0.192 | 0.934 | 0.931 | 0.597 | $0.736 \pm 0.12$ | 0.919 |
| RNN (+NS, S) | 0.963 | 0.122 | 0.958 | 0.955 | 0.559 | $0.684 \pm 0.14$ | 0.919 |
| RNN (+NS, L) | 0.935 | 0.195 | 0.934 | 0.931 | 0.645 | $0.750 \pm 0.14$ | 0.919 |
| RNN (+LM+NS, S) | 0.982 | 0.060 | 0.978 | 0.973 | 0.610 | $0.714 \pm 0.15$ | 0.919 |
| RNN (+LM+NS, L) | 0.956 | 0.190 | 0.937 | 0.945 | **0.910** | $0.784 \pm 0.10$ | 0.916 |
| LSTM (S) | 0.946 | 0.145 | 0.951 | 0.939 | 0.616 | $0.734 \pm 0.10$ | 0.902 |
| LSTM (L) | 0.937 | 0.189 | 0.933 | 0.933 | 0.774 | $0.712 \pm 0.06$ | 0.792 |
| LSTM (+LM, S) | 0.995 | **0.019** | **0.994** | 0.983 | 0.610 | $0.664 \pm 0.10$ | 0.898 |
| LSTM (+LM, L) | 0.961 | 0.155 | 0.943 | 0.952 | 0.876 | $0.759 \pm 0.12$ | 0.897 |
| LSTM (+NS, S) | 0.992 | 0.046 | 0.987 | 0.980 | 0.652 | $0.779 \pm 0.12$ | 0.919 |
| LSTM (+NS, L) | 0.961 | 0.165 | 0.942 | 0.966 | 0.890 | $0.784 \pm 0.11$ | 0.915 |
| LSTM (+LM+NS, S) | **0.997** | 0.031 | **0.994** | **0.986** | 0.639 | $0.638 \pm 0.07$ | 0.756 |
| LSTM (+LM+NS, L) | 0.971 | 0.123 | 0.953 | 0.966 | 0.771 | $0.688 \pm 0.10$ | 0.915 |

Table 20: Full results on the **Compute Sqrt** language.

| Model | Train | Val. CE ↓ | Val. | S. Test | L. Test | L. Test (Mean) | L. Test (Max) |
|---|---|---|---|---|---|---|---|
| Tf (S) | 0.956 | 0.130 | 0.954 | 0.948 | 0.718 | 0.673 ± 0.10 | 0.828 |
| Tf (L) | 0.937 | 0.254 | 0.898 | 0.926 | 0.629 | 0.705 ± 0.10 | 0.860 |
| Tf (+LM, S) | 0.971 | 0.100 | 0.964 | 0.960 | 0.629 | 0.623 ± 0.10 | 0.885 |
| Tf (+LM, L) | 0.911 | 0.264 | 0.887 | 0.906 | 0.629 | 0.648 ± 0.09 | 0.782 |
| Tf (+NS, S) | 0.973 | 0.101 | 0.967 | 0.967 | 0.578 | 0.631 ± 0.11 | 0.852 |
| Tf (+NS, L) | 0.953 | 0.205 | 0.919 | 0.947 | 0.645 | 0.639 ± 0.06 | 0.739 |
| Tf (+LM+NS, S) | 0.970 | 0.094 | 0.968 | 0.963 | 0.592 | 0.648 ± 0.11 | 0.856 |
| Tf (+LM+NS, L) | 0.960 | 0.177 | 0.932 | 0.952 | 0.811 | 0.695 ± 0.11 | 0.846 |
| RNN (S) | 0.892 | 0.280 | 0.897 | 0.884 | **0.887** | 0.784 ± 0.12 | 0.887 |
| RNN (L) | 0.892 | 0.306 | 0.878 | 0.884 | 0.773 | 0.699 ± 0.13 | 0.887 |
| RNN (+LM, S) | 0.976 | 0.110 | 0.965 | 0.975 | 0.647 | 0.681 ± 0.15 | 0.882 |
| RNN (+LM, L) | 0.911 | 0.281 | 0.893 | 0.898 | 0.806 | 0.776 ± 0.09 | 0.878 |
| RNN (+NS, S) | 0.982 | 0.099 | 0.976 | 0.982 | 0.636 | 0.720 ± 0.15 | 0.887 |
| RNN (+NS, L) | 0.894 | 0.301 | 0.874 | 0.884 | 0.709 | 0.767 ± 0.08 | 0.887 |
| RNN (+LM+NS, S) | 0.987 | 0.080 | 0.979 | 0.983 | 0.644 | 0.638 ± 0.10 | 0.887 |
| RNN (+LM+NS, L) | 0.917 | 0.283 | 0.888 | 0.910 | 0.816 | 0.715 ± 0.14 | **0.887** |
| LSTM (S) | 0.892 | 0.278 | 0.897 | 0.884 | 0.853 | **0.836** ± 0.07 | 0.887 |
| LSTM (L) | 0.892 | 0.307 | 0.878 | 0.883 | **0.887** | 0.799 ± 0.07 | 0.887 |
| LSTM (+LM, S) | 0.995 | 0.043 | 0.989 | 0.989 | 0.658 | 0.653 ± 0.15 | 0.887 |
| LSTM (+LM, L) | 0.985 | 0.123 | 0.959 | 0.982 | 0.610 | 0.697 ± 0.13 | 0.887 |
| LSTM (+NS, S) | 0.986 | 0.076 | 0.976 | 0.978 | 0.576 | 0.795 ± 0.10 | 0.887 |
| LSTM (+NS, L) | 0.915 | 0.293 | 0.869 | 0.914 | 0.729 | 0.830 ± 0.07 | 0.887 |
| LSTM (+LM+NS, S) | **0.995** | **0.036** | **0.991** | **0.994** | 0.659 | 0.614 ± 0.08 | 0.831 |
| LSTM (+LM+NS, L) | 0.984 | 0.140 | 0.944 | 0.981 | 0.613 | 0.679 ± 0.11 | 0.887 |

Table 21: Full results on the **Bucket Sort** language.

| Model | Train | Val. CE ↓ | Val. | S. Test | L. Test | L. Test (Mean) | L. Test (Max) |
|---|---|---|---|---|---|---|---|
| Tf (S) | 0.974 | 0.107 | 0.973 | 0.963 | 0.569 | 0.629 ± 0.08 | 0.764 |
| Tf (L) | 0.821 | 0.417 | 0.806 | 0.824 | 0.670 | 0.727 ± 0.03 | 0.756 |
| Tf (+LM, S) | 0.986 | 0.029 | 0.989 | 0.977 | 0.556 | 0.590 ± 0.10 | 0.858 |
| Tf (+LM, L) | 0.988 | 0.094 | 0.973 | 0.975 | 0.837 | 0.706 ± 0.11 | 0.880 |
| Tf (+NS, S) | 0.976 | 0.077 | 0.978 | 0.977 | 0.891 | 0.616 ± 0.12 | 0.891 |
| Tf (+NS, L) | 0.945 | 0.200 | 0.920 | 0.936 | 0.878 | 0.709 ± 0.13 | 0.884 |
| Tf (+LM+NS, S) | 0.997 | 0.008 | 0.999 | 0.992 | 0.553 | 0.603 ± 0.10 | 0.874 |
| Tf (+LM+NS, L) | 0.960 | 0.168 | 0.947 | 0.948 | 0.579 | 0.572 ± 0.02 | 0.629 |
| RNN (S) | 0.967 | 0.132 | 0.966 | 0.953 | 0.911 | 0.766 ± 0.09 | 0.911 |
| RNN (L) | 0.945 | 0.181 | 0.942 | 0.929 | 0.757 | 0.769 ± 0.14 | 0.897 |
| RNN (+LM, S) | 0.990 | 0.063 | 0.984 | 0.977 | 0.575 | 0.647 ± 0.12 | 0.880 |
| RNN (+LM, L) | 0.960 | 0.128 | 0.965 | 0.953 | 0.630 | 0.706 ± 0.10 | 0.908 |
| RNN (+NS, S) | 0.965 | 0.138 | 0.964 | 0.957 | 0.924 | 0.778 ± 0.13 | 0.924 |
| RNN (+NS, L) | 0.962 | 0.126 | 0.966 | 0.954 | **0.957** | 0.771 ± 0.15 | 0.957 |
| RNN (+LM+NS, S) | 0.962 | 0.142 | 0.962 | 0.950 | 0.930 | **0.836** ± 0.09 | 0.951 |
| RNN (+LM+NS, L) | 0.960 | 0.129 | 0.965 | 0.953 | 0.613 | 0.690 ± 0.09 | 0.869 |
| LSTM (S) | 0.969 | 0.095 | 0.966 | 0.966 | 0.546 | 0.688 ± 0.13 | **0.967** |
| LSTM (L) | 0.971 | 0.109 | 0.970 | 0.956 | 0.813 | 0.700 ± 0.09 | 0.813 |
| LSTM (+LM, S) | 0.998 | 0.013 | 0.997 | 0.991 | 0.592 | 0.645 ± 0.10 | 0.774 |
| LSTM (+LM, L) | **0.999** | **0.003** | **1.000** | **0.998** | 0.605 | 0.621 ± 0.10 | 0.769 |
| LSTM (+NS, S) | 0.996 | 0.015 | 0.998 | 0.992 | 0.576 | 0.572 ± 0.04 | 0.664 |
| LSTM (+NS, L) | 0.979 | 0.087 | 0.980 | 0.968 | 0.592 | 0.588 ± 0.04 | 0.671 |
| LSTM (+LM+NS, S) | 0.992 | 0.039 | 0.991 | 0.984 | 0.569 | 0.554 ± 0.03 | 0.597 |
| LSTM (+LM+NS, L) | 0.989 | 0.075 | 0.982 | 0.981 | 0.594 | 0.640 ± 0.09 | 0.835 |

Table 22: Results for different model sizes.

| Language | Tf | | | RNN | | | LSTM | | |
|---|---|---|---|---|---|---|---|---|---|
| | 32k | 64k | 128k | 32k | 64k | 128k | 32k | 64k | 128k |
| Repeat 01 | 0.85 | 0.86 | 0.85 | 1.00 | 1.00 | 1.00 | 1.00 | 1.00 | 1.00 |
| Parity | 0.54 | 0.54 | 0.54 | 0.53 | 0.54 | 0.52 | 1.00 | 1.00 | 1.00 |
| Cycle Navigation | 0.93 | 0.93 | 0.93 | 0.93 | 0.93 | 0.93 | 0.93 | 0.93 | 0.93 |
| Modular Arithmetic | 0.86 | 0.83 | **0.92** | 1.00 | 1.00 | 1.00 | 1.00 | 1.00 | 1.00 |
| Dyck-$(2, 3)$ | 0.73 | 0.80 | 0.79 | 0.98 | 0.98 | 0.98 | 0.98 | 0.98 | 0.98 |
| Majority | 1.00 | 0.99 | 1.00 | 0.93 | 0.93 | 0.94 | 1.00 | 0.99 | 1.00 |
| Stack Manipulation | 0.82 | 0.87 | 0.87 | 0.93 | 0.93 | 0.93 | 0.84 | 0.89 | 0.92 |
| Marked Reversal | 0.75 | 0.76 | 0.79 | 0.95 | 0.95 | 0.94 | 0.91 | 0.86 | 0.96 |
| Unmarked Reversal | 0.63 | 0.63 | 0.63 | 0.76 | 0.76 | 0.76 | 0.76 | 0.76 | 0.78 |
| Marked Copy | 0.76 | 0.80 | 0.85 | 0.95 | 0.95 | 0.95 | 0.94 | 0.95 | 0.92 |
| Missing Duplicate | 0.83 | 0.86 | 0.77 | 0.95 | 0.94 | 0.95 | 0.95 | 0.92 | 0.95 |
| Odds First | 0.77 | 0.85 | 0.85 | 0.95 | 0.95 | 0.92 | 0.95 | 0.91 | 0.95 |
| Binary Addition | 0.79 | 0.78 | 0.83 | 0.92 | 0.88 | 0.88 | 0.92 | 0.92 | 0.92 |
| Binary Multiplication | 0.80 | 0.82 | 0.92 | 0.92 | 0.89 | 0.78 | 0.89 | 0.79 | 0.92 |
| Compute Sqrt | 0.82 | 0.86 | 0.80 | 0.89 | 0.89 | 0.89 | 0.89 | 0.89 | 0.89 |
| Bucket Sort | 0.76 | 0.76 | 0.87 | 0.92 | 0.90 | 0.91 | 0.80 | 0.81 | **0.93** |

## H  RESULTS WITH SMALLER AND LARGER MODELS

Is the reason that some models do not get perfect accuracy in §5 simply because they are not big enough? In this section, we find that the answer is generally no. To test this, we show results with larger and smaller models in Table 22. The experiments in §5 use models with approximately 64k parameters. Here, we train models with approximately 32k, 64k, and 128k parameters, using the same rounding technique described in §4. We omit the languages Even Pairs and First, as all architectures already get perfect accuracy on them in Table 2. We run experiments according to the expressivity setup described in §4, but using only a recognition head, without auxiliary loss terms. We report the best test accuracy of 10 runs. There are only two cases where a larger model results in an improvement over Table 2 of more than 0.01, which we show in **bold**. We see a strict positive correlation between accuracy and model size in only a few cases: transformer on Marked Reversal, Marked Copy, and Binary Multiplication; and LSTM on Stack Manipulation and Bucket Sort. Interestingly, we see a negative correlation in the case of RNN on Binary Multiplication. Otherwise, we do not see clear trends, suggesting that the limitations are due to model architecture rather than size.

## I  PERFORMANCE VS. INPUT LENGTH

We show recognition cross-entropy (lower is better) vs. input length for the models shown under "Expressivity" in Table 2. At every length $n$ on the x-axis that is a multiple of 10, we show the average cross-entropy of the model on all strings in the long test set with lengths in the range $[n - 10, n + 10]$ (this smooths the curves for the sake of readability). The shaded regions indicate one standard deviation. The vertical dashed lines indicate the maximum lengths in the training and validation sets. The horizontal dashed lines mark $\log 2$, which is the threshold for incorrect predictions. In general, we find that cross-entropy usually does not increase significantly on longer strings. Notable exceptions include Repeat 01, Parity, Modular Arithmetic, and Dyck-$(2, 3)$ for the transformer and Majority for the RNN. This is in contrast to Delétang et al. (2023), who found that models often fail catastrophically on longer input strings in their string-to-string transduction setup.

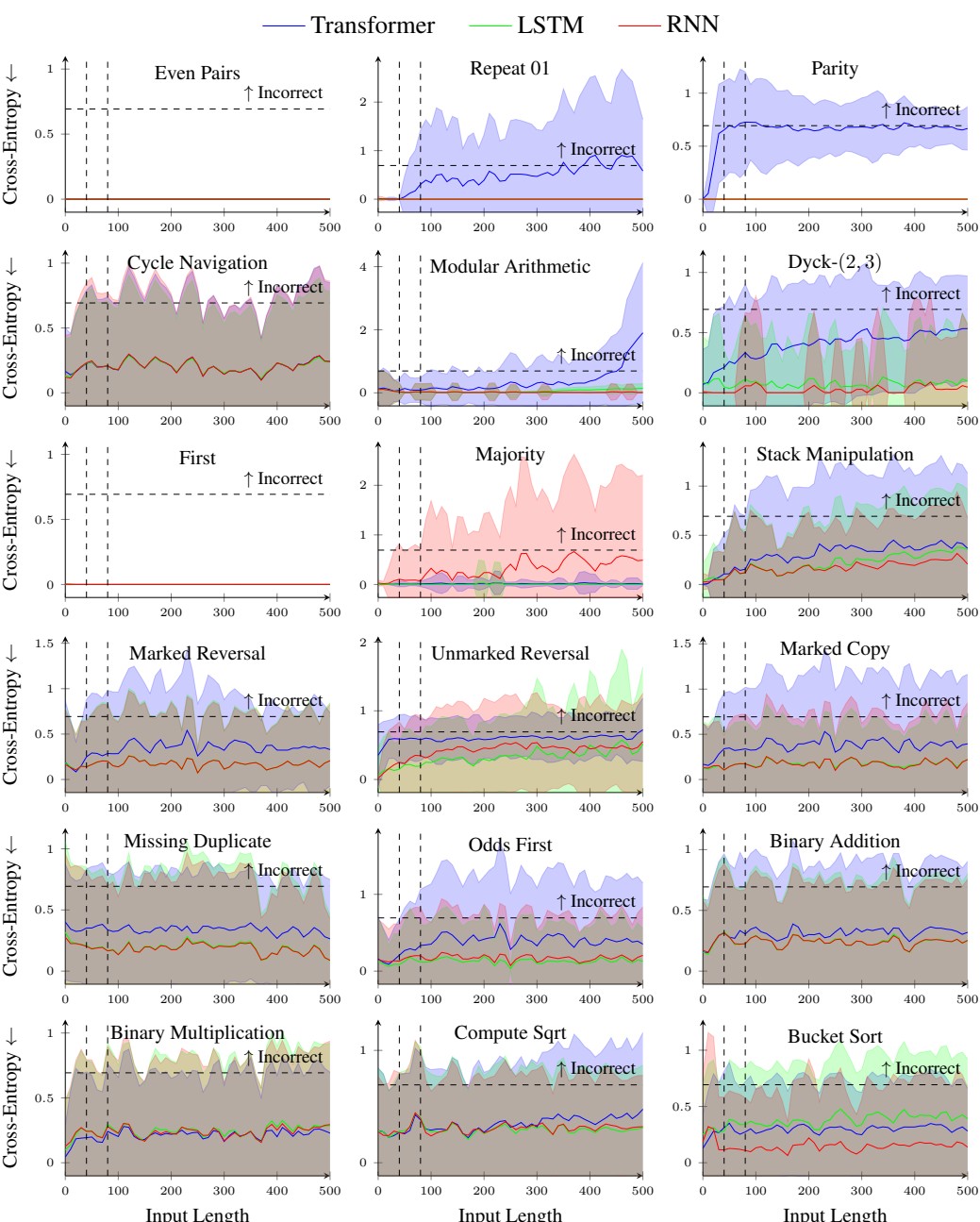

Figure 3: Recognition cross-entropy (lower is better) vs. input length for the models shown under "Expressivity" in Table 2.

## J   EDIT DISTANCE FROM LANGUAGES

In this section, we give a formal definition of string–language edit distance and provide an algorithm for computing it for regular languages.

**Definition 10.** *For any string $w$ and language $L$, the **edit distance** $\mathcal{D}(w, L)$ is defined as*

$$\mathcal{D}(w, L) \overset{\text{def}}{=} \min_{u \in L} \mathcal{D}(w, u), \tag{20}$$

*where $\mathcal{D}(w, u)$ is the Levenshtein distance between $w$ and $u$, or the minimal number of single-symbol edits (insertions, deletions, and replacements) required to transform $w$ into $u$.*

Suppose $L$ is a regular language over alphabet $\Sigma$ recognized by DFA $\mathcal{A}$. Then for any $w \in \Sigma^*$, $\mathcal{D}(w, L)$ can be computed as follows:

1. Construct a nondeterministic weighted finite automaton (WFA) $\mathcal{A}_{\mathcal{D}(w, \cdot)}$ over the **tropical semiring** $(\mathbb{R}_{\geq 0} \cup \{\infty\}, \min, +, \infty, 0)$ that assigns weight $\mathcal{D}(w, u)$ to every string $u$ (Figure 4);

2. Lift $\mathcal{A}$ to the tropical semiring by assigning weight 0 to all transitions and accepting states, resulting in a WDFA $\mathcal{A}_{\cdot \in L}$;

3. Intersect $\mathcal{A}_{\cdot \in L}$ and $\mathcal{A}_{\mathcal{D}(w, \cdot)}$ using the standard WFA intersection algorithm (Mohri, 2009), resulting in an automaton $\mathcal{A}_{\mathcal{D}(\cdot, L)}$ that encodes $\mathcal{D}(w, u)$ for every $u \in L$; and

4. Compute the shortest path in $\mathcal{A}_{\mathcal{D}(\cdot, L)}$ using the Floyd–Warshall algorithm (Floyd, 1962; Warshall, 1962).

The rest of this section describes these steps in more detail and argues their correctness.

### J.1   WEIGHTED FINITE AUTOMATA

We start by defining WFAs, which are used in multiple steps. This is a more general, nondeterministic version of Def. 4.

**Definition 11.** *A **weighted finite automaton (WFA)** over semiring $(\mathbb{K}, \oplus, \otimes, \mathbf{0}, \mathbf{1})$ is a tuple $\mathcal{A} = (Q, \Sigma, \delta, q_0, \rho)$ such that (1) $Q$ is a finite set of states, (2) $\Sigma$ is an alphabet, (3) $\delta \colon Q \times (\Sigma \cup \{\varepsilon\}) \times Q \to \mathbb{K}$ is the transition function, and (4) $q_0 \in Q$ is the start state;[8] and (5) $\rho \colon Q \to \mathbb{K}$ is the accept weight function. If $\delta(q, a, r) = w$, we say that $\mathcal{A}$ has a transition from $q$ to $r$ that scans $a$ with weight $w$, and we write $q \xrightarrow{a/w} r \in \delta$.*

This definition is nondeterministic in the sense that it permits multiple outgoing transitions on the same symbol from the same state. We define paths and path weights in a similar way to §3.1.2.

**Definition 12.** *A **path** $\pi$ in WFA $\mathcal{A}$ is a sequence of states and transitions*

$$\pi = r_0 \xrightarrow{a_1/w_1} r_1 \cdots r_{m-1} \xrightarrow{a_m/w_m} r_m \tag{21}$$

*such that*

1. *$r_0 = q_0$, and*

2. *for all $i = 0, \ldots, m-1$, $r_i \xrightarrow{a_{i+1}/w_{i+1}} r_{i+1} \in \delta$.*

*We say that $\pi$ **scans** the string $a_1 \cdots a_m$, and that the **path weight** of $\pi$ is*

$$\mathbf{w}(\pi) \overset{\text{def}}{=} \left( \bigotimes_{i=1}^{m} w_i \right) \otimes \rho(r_m). \tag{22}$$

---

[8]Some definitions use an initial weight function $\lambda \colon Q \to \mathbb{K}$ to indicate start states. For simplicity, we assume one start state with a weight of $\mathbf{1}$.

Note that in a nondeterministic WFA, multiple paths may scan the same string. We denote the set of all paths of $\mathcal{A}$ as $\Pi(\mathcal{A})$, and the set of paths that scan $w$ as $\Pi(\mathcal{A}, w)$.

The weight that a WFA assigns to a string is the sum of the weights of all paths that scan that string.

**Definition 13.** *The **stringsum** of string $w \in \Sigma^*$ under WFA $\mathcal{A}$ is*

$$\mathcal{A}(w) \stackrel{\text{def}}{=} \bigoplus_{\pi \in \Pi(\mathcal{A}, w)} \mathbf{w}(\pi). \tag{23}$$

We also make use of the sum of the weights of all paths in a WFA.

**Definition 14.** *The **allsum** of WFA $\mathcal{A}$ is*

$$Z(\mathcal{A}) \stackrel{\text{def}}{=} \bigoplus_{\pi \in \Pi(\mathcal{A})} \mathbf{w}(\pi) \tag{24a}$$

$$= \bigoplus_{w \in \Sigma^*} \mathcal{A}(w). \tag{24b}$$

### J.2 Algorithm details

First, we encode the input string $w = w_1 w_2 \cdots w_n$ into a chain-like WFA $\mathcal{A}_{\mathcal{D}(w, \cdot)}$ in the tropical semiring, as shown in Figure 4.

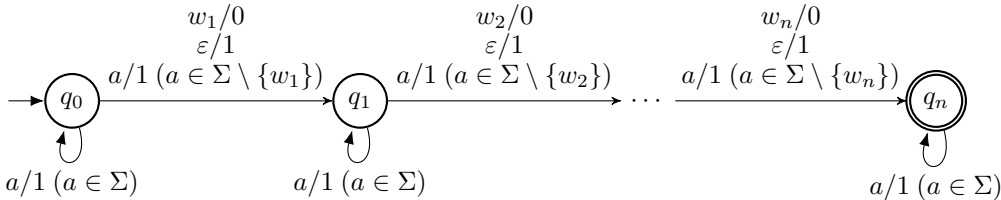

Figure 4: Diagram of $\mathcal{A}_{\mathcal{D}(w, \cdot)}$. The double-circled state has accept weight 0; the others have accept weight $\infty$.

Given an input string $u \in \Sigma^*$, every path in $\mathcal{A}_{\mathcal{D}(w, \cdot)}$ that scans $u$ encodes a way of transforming $w$ into $u$. Every time a transition deviates from scanning $w$ by inserting, replacing, or deleting a symbol, it incurs a cost of 1. Taking the minimum weight of any path that scans $u$ gives $\mathcal{D}(w, u)$.

**Lemma 1.** *For all $u \in \Sigma^*$, $\mathcal{A}_{\mathcal{D}(w, \cdot)}(u) = \mathcal{D}(w, u)$.*

*Proof.* For every $w_i$ in $w$, $\mathcal{A}_{\mathcal{D}(w, \cdot)}$ either matches a symbol in $u$ with $\mathcal{A}_{\mathcal{D}(w, \cdot)}$ with cost 0, simulates the deletion of $w_i$ with cost 1 so that it can continue scanning $u$ some other way, or simulates the replacement of $w_i$ with a symbol in $u$ with cost 1. Before and after symbols in $w$, $\mathcal{A}_{\mathcal{D}(w, \cdot)}$ can also simulate the insertion of any number of symbols, each with cost 1. So, a path is in $\Pi(\mathcal{A}_{\mathcal{D}(w, \cdot)}, u)$ iff it corresponds to a way of changing $w$ into $u$, and its weight is the number of edits it performs to turn $w$ into $u$. The stringsum gives the minimum number of edits.

$$\mathcal{A}_{\mathcal{D}(w, \cdot)}(u) = \bigoplus_{\pi \in \Pi(\mathcal{A}_{\mathcal{D}(w, \cdot)}, u)} \mathbf{w}(\pi) \tag{25a}$$

$$= \min_{\pi \in \Pi(\mathcal{A}_{\mathcal{D}(w, \cdot)}, u)} \mathbf{w}(\pi) \tag{25b}$$

$$= \mathcal{D}(w, u) \tag{25c}$$

$$\square$$

Next, we lift the weights of $\mathcal{A}$ into the tropical semiring, resulting in a WDFA $\mathcal{A}_{\cdot \in L}$. The weights of all transitions and the accept weights of all accepting states in $\mathcal{A}$ are set to 0 in $\mathcal{A}_{\cdot \in L}$. All other weights are set to $\infty$. So, $\mathcal{A}_{\cdot \in L}$ assigns weight 0 to all strings in $L$, and weight $\infty$ to all others.

**Lemma 2.** *For all $u \in \Sigma^*$,*

$$\mathcal{A}_{\cdot \in L}(u) = \begin{cases} 0 & \textit{if } u \in L \\ \infty & \textit{otherwise.} \end{cases} \tag{26}$$

*Proof.* By definition, $\mathcal{A}$ has a path that scans $u$ iff $u \in L$. By construction, $\mathcal{A}_{\cdot \in L}$ has a path that scans $u$ with weight 0 iff $\mathcal{A}$ has a path that scans $u$. Therefore, if $u \in L$, the stringsum $\mathcal{A}_{\cdot \in L}(u)$ is the minimum of one or more path weights of 0, so it is 0. Otherwise, the stringsum is a summation over an empty set of path weights, which is defined to be $\infty$ in the tropical semiring. $\square$

Next, we intersect $\mathcal{A}_{\mathcal{D}(w,\cdot)}$ and $\mathcal{A}_{\cdot \in L}$ using the standard intersection algorithm for WFAs, resulting in a WFA $\mathcal{A}_{\mathcal{D}(\cdot,L)}$ that assigns weight $\mathcal{D}(w,u)$ to $u$ if $u \in L$ and $\infty$ otherwise.

**Lemma 3.** *For all $u \in \Sigma^*$,*

$$\mathcal{A}_{\mathcal{D}(\cdot,L)}(u) = \begin{cases} \mathcal{D}(w,u) & \textit{if } u \in L \\ \infty & \textit{otherwise.} \end{cases} \tag{27}$$

*Proof.* By definition of intersection, the stringsum of the intersected automaton is

$$\mathcal{A}_{\mathcal{D}(\cdot,L)}(u) \overset{\text{def}}{=} \mathcal{A}_{\mathcal{D}(w,\cdot)}(u) \otimes \mathcal{A}_{\cdot \in L}(u). \tag{28}$$

Using Lemmas 1 and 2 and Eq. (28), we have

$$\mathcal{A}_{\mathcal{D}(\cdot,L)}(u) = \begin{cases} \mathcal{D}(w,u) + 0 & \text{if } u \in L \\ \mathcal{D}(w,u) + \infty & \text{otherwise} \end{cases} \tag{29a}$$

$$= \begin{cases} \mathcal{D}(w,u) & \text{if } u \in L \\ \infty & \text{otherwise.} \end{cases} \tag{29b}$$

$\square$

Finally, we compute the allsum of $\mathcal{A}_{\mathcal{D}(\cdot,L)}$, which gives us the minimum edit distance from $w$ to any string $u \in L$. Let $\mathcal{A}_{\mathcal{D}(\cdot,L)} = (Q, \Sigma, \delta, q_0, \rho)$. To compute the allsum, we first use the Floyd–Warshall all-pairs shortest path algorithm.[9] to compute the shortest path weight from $q_0$ to $r$, denoted $A[q_0, r]$, for every $r \in Q$. We then compute the allsum as

$$Z(\mathcal{A}_{\mathcal{D}(\cdot,L)}) = \bigoplus_{r \in Q} A[q_0, r] \otimes \rho(r) \tag{30a}$$

$$= \min_{r \in Q} A[q_0, r] + \rho(r). \tag{30b}$$

This gives us the edit distance $\mathcal{D}(w, L)$.

**Theorem 1.** $Z(\mathcal{A}_{\mathcal{D}(\cdot,L)}) = \mathcal{D}(w, L)$.

*Proof.* By definition, the allsum is

$$Z(\mathcal{A}_{\mathcal{D}(\cdot,L)}) \overset{\text{def}}{=} \bigoplus_{u \in \Sigma^*} \mathcal{A}_{\mathcal{D}(\cdot,L)}(u). \tag{31}$$

Using Lemma 3 and Def. 10, we have

$$Z(\mathcal{A}_{\mathcal{D}(\cdot,L)}) = \min_{u \in \Sigma^*} \mathcal{A}_{\mathcal{D}(\cdot,L)}(u) \tag{32a}$$

$$= \min_{u \in L} \mathcal{D}(w,u) \tag{32b}$$

$$= \mathcal{D}(w, L). \tag{32c}$$

$\square$

---

[9]This is a special case of Lehmann's algorithm (Algorithm 4) The only difference is that we do not need to compute the star operation in Algorithm 4, line 7, which is always 0 in the tropical semiring.

