# OpenReview forum: "Training Neural Networks as Recognizers of Formal Languages"
_ICLR.cc/2025/Conference — ICLR 2025 Poster_

### Official Review · Reviewer_eTo5 · 2024-10-31

**Soundness:** 4
**Presentation:** 4
**Contribution:** 3
**Rating:** 8
**Confidence:** 3

**Summary:**

This paper builds on Deletang et. al. 2023, and rigorously tests standard sequence neural architectures for their ability to _recognise_ formal languages that span the Chomsky Hierarchy. It also proposes a way to sample sequences from automata with a length constraint in order to generate training data for the experiment.

**Strengths:**

The paper references important work on the theoretical expressibility of Transformers, and other neural architectures, and, along with Deletang et. al. 2023, is useful information for understanding the learnability of these architectures in an empirical way.

It follows a similar set of experiments as the Deletang et. al. 2023 paper, while focusing on the recognition aspect of formal languages, rather than mixing recognition and transduction tasks. This of course comes with some drawbacks (e.g. uninformative training signal), but the authors spend some time addressing that issue with other reasonable auxiliary losses.

The length-constrained sampling from DFAs is an interesting aspect, even if it takes a bit of a detour from the original point of the paper

Understanding the limitations of the neural architectures we use is important, and can only be explored and understood in smaller scale experiments lik

**Weaknesses:**

I have only minor comments here as I find the paper well-executed:
- Table 2: decide on a threshold and make results above it visually dissimilar, this will give a better idea of where a model sits on these benchmarks and the hierarchy they’re measured to be in
- Missing a more conclusive figure regarding placement of models wrt to their level in the chomsky hierarchy. I know this is a tough ask, since the mapping of these models don’t fit as neatly into the Chomsky hierarchy as Deletang, et. al. 2023 suggest. Perhaps a better visual representation could be figured out.

**Questions:**

Can the length constrained algorithm for DFAs be modified for other automata? I suspect not due to the combinatorial way the paths compose for these other automata, and the infinite nature of the stack, etc.

---

> ### Author Response · Authors · 2024-11-15
> **Responses to comments and questions**
>
> Thank you very much for your positive comments and feedback. We greatly appreciate your suggestions for improving the paper further.
>
> > Table 2: decide on a threshold and make results above it visually dissimilar, this will give a better idea of where a model sits on these benchmarks and the hierarchy they’re measured to be in
>
> Thank you for this suggestion; we agree that making the positive results visually distinctive would be useful. We are hesitant to decide on a uniform threshold across languages like Deletang et al. (2023), because a score above 90\% accuracy does not necessarily imply success, due to the individual characteristics of each language. For example, as discussed in our results section, on Cycle Navigation, all models top out at about 93\% accuracy, but this is because they only learn the format of positive examples and fail to reject strings where the digit at the end is wrong.
>
> One idea would be to simply bold the best score among the three architectures in each row. Would that be helpful?
>
> > Missing a more conclusive figure regarding placement of models wrt to their level in the chomsky hierarchy. I know this is a tough ask, since the mapping of these models don’t fit as neatly into the Chomsky hierarchy as Deletang, et. al. 2023 suggest. Perhaps a better visual representation could be figured out.
>
> Thank you for this suggestion also. One idea would be to include a Venn diagram similar to Deletang et al. (2023), and to include each language that an architecture gets 100\% accuracy on as a dot in the Venn diagram. We will try to include this if we have space. As you pointed out, we don't expect the architectures to correspond neatly with classes in the Chomsky hierarchy. LSTMs have been shown to recognize non-regular counting languages, and transformers cut unevenly across the Chomsky hierarchy (see the survey by Strobl et al. (2024)). We will include more discussion of this in the results section of our next revision.
>
> > Can the length constrained algorithm for DFAs be modified for other automata? I suspect not due to the combinatorial way the paths compose for these other automata, and the infinite nature of the stack, etc.
>
> Yes, it can! It can be applied to any type of automaton that supports a semiring-general "allsum" algorithm that sums over all path weights. This includes pushdown automata (PDAs, corresponding to context-free languages) and embedded pushdown automata (EPDAs, corresponding to mildly context-sensitive languages, equivalent to tree-adjoining grammars, see [1]). Although PDAs can have infinitely many possible stack configurations, we can sum over all of their path weights using dynamic programming, based on a variant of Lang's algorithm [2]. It is similarly possible to do this for EPDAs. We plan to extend what we did for DFAs to PDAs and EPDAs in future work.
>
> [1] K. Vijay-Shanker. "A Study of Tree Adjoining grammars." 1987. Ph.D. thesis, University of Pennsylvania.
>
> [2] Bernard Lang. "Deterministic techniques for efficient non-deterministic parsers." 1974. In Proc. Colloquium on Automata, Languages, and Programming, pages 255–269.

---

> > ### Comment · Reviewer_eTo5 · 2024-11-19
> > **Thanks!**
> >
> > Thanks for the references, I'll take a look!

---

> > > ### Author Response · Authors · 2024-11-26
> > >
> > > We have updated our draft, and you may be particularly interested in these changes:
> > >
> > > 1. We have added a figure summarizing the expressivity results of our experiments to the introduction. Thanks again for this great suggestion!
> > > 2. We now bold the best scores in the main table of results.

---

### Official Review · Reviewer_gn5o · 2024-11-02

**Soundness:** 3
**Presentation:** 2
**Contribution:** 2
**Rating:** 5
**Confidence:** 4

**Summary:**

This paper argue that there exists a methodological disconnect between the complexity theory of the formal language and the empirical evaluation of neural networks' computational capabilities. The existing work related the capabilities of neural networks with Chomsky hierarchy, which is framed in terms of language recognition (classifying whether strings belong to a language). However their empirical studies evaluate models on string-to-string transformation tasks instead. This paper propose an experimental setup for training neural networks as recognizers, and a more efficient algorithm for length-controlled sampling from finite automata. The empirical results show that RNNs and LSTMs often perform better than transformers on the formal language recognition task.

**Strengths:**

1. It is an important direction to study neural networks' computational ability through formal language theory.
2. This paper introduces an efficient algorithm for length-controlled sampling from finite automata, which may have practical value for future research in formal language processing.
3. Experiments are conducted on a variety of formal languages with 3 neural models with different architectures (RNN, LSTM, transformer). The methodology appears to be well-documented and reproducible.

**Weaknesses:**

1. The paper overlooks significant existing work on training neural networks for formal language recognition tasks (e.g. [1, 2, 3]). This oversight diminishes the claimed novelty of the proposed experimental setup. The authors should acknowledge and position their work in relation to previous studies.
2. The claimed technical improvement in sampling from finite automata appears to be the main novel contribution, but its significance needs better contextualization.

References
[1] Bhattamishra, Satwik, Kabir Ahuja, and Navin Goyal. "On the ability and limitations of transformers to recognize formal languages." EMNLP 2020.
[2] William Merrill. 2019. Sequential Neural Networks as Automata.
[3] Satwik Bhattamishra, Kabir Ahuja, and Navin Goyal. 2020. On the Practical Ability of Recurrent Neural Networks to Recognize Hierarchical Languages.

**Questions:**

Please refer to the weaknesses part.

---

> ### Author Response · Authors · 2024-11-15
> **Responses to comments**
>
> Thank you very much for your review, for your positive remarks, and for pointing out areas where our work can be better contextualized with respect to past work.
>
> > The paper overlooks significant existing work on training neural networks for formal language recognition tasks (e.g. [1, 2, 3]). This oversight diminishes the claimed novelty of the proposed experimental setup. The authors should acknowledge and position their work in relation to previous studies.
>
> Thank you for these references. You are correct to point out that there are many prior empirical studies that attempt to train neural networks on formal languages; another example of a highly relevant paper is MLRegTest [4], which we will make more explicit comparisons to in the next revision. Broadly speaking, although there have been a number of past papers that use neural networks for formal language recognition, each paper uses a slightly different methodology, without much empirical justification for the particular choices made. Our work is an attempt to unify this methodology, to extend it to the broadest possible class of languages, to compare different training objectives used in past work, and to spur theoretically sound empirical work in the future. We believe that our work does improve on each of these prior papers in significant ways.
>
> Our work differs from [1] and [3] in the following ways:
>
> 1. They only use the next symbol prediction objective (NS) with MSE loss, whereas we experiment with multiple objectives (R, R+LM, R+NS, R+LM+NS). As far as we are aware, our work is the first to compare these different objectives for training recognizers. One of the interesting findings in our work is that using just binary cross-entropy (R) is often just as good as or better than NS, which overturns common wisdom going back to before Gers and Schmidhuber (2001) that using next symbol prediction is necessary. Reference [1] includes a discussion comparing NS to R and LM objectives in Appendix D.1, but we are the first to empirically compare them.
> 2. They do not do negative sampling. Next symbol prediction does implicitly include negative samples if it requires predicting EOS, but even then, negative examples are restricted to proper prefixes of positive examples. Our negative sampling approach is more general, and it can be used without needing an algorithm for computing the next symbol sets at each timestep. This is useful because for some languages, it might be feasible to write algorithms for positive sampling and membership testing, but much more complicated to compute next symbol sets. In this way, our work enables empirical testing on more languages than before. Negative examples in general are theoretically important for formal language learning (see [5]).
> 3. Reference [1] only tests regular and counter languages, and reference [3] only tests on Dyck languages, whereas we test languages from both inside and outside those classes.
> 4. They use simple language-specific algorithms for length-constrained sampling from regular languages, whereas we have proposed an efficient length-constrained sampling technique that works for arbitrary regular languages expressed as DFAs.
>
> Reference [2] only uses a language modeling objective (LM) and does not do negative sampling, and it is specific to the language $a^n b^n c$. As mentioned above, we also compare multiple training objectives and provide a general method for negative sampling. Moreover, in our work, we show that the LM objective is least often the best objective for training recognizers, whereas R is most often the best objective. We will emphasize this in the results section of our next revision.
>
> Reference [4] is more similar to our work than [1-3]. It does both positive and negative sampling, although it is restricted to regular languages expressed as DFAs, whereas our method applies to non-regular languages as well. Their negative sampling technique relies on algorithms that are specific to DFAs, and they only generate adversarial negative examples with edit distance 1. They propose an algorithm for length-constrained sampling from DFAs, but they do not provide a runtime analysis, and it does not sample from the true posterior distribution of the uniformly-weighted PDFA. They sample strings up to length 50, whereas we sample strings up to length 500.
>
> We will be sure to highlight these distinctions in the next revision as space allows.
>
> (Continued in next comment)

---

> > ### Author Response · Authors · 2024-11-15
> > **Responses to comments (2)**
> >
> > > The claimed technical improvement in sampling from finite automata appears to be the main novel contribution, but its significance needs better contextualization.
> >
> > Although we agree that our DFA sampling method is an important technical contribution, the main contribution of our paper is to point out the discrepancy between experiments and formal claims in some prior work, to provide a general experimental setup for empirically testing these claims, and to demonstrate its effectiveness on a suite of languages inspired by prior work. We are not aware of other work that points out this discrepancy or proposes a general language recognition setup like ours. This will drive theoretically sound work in the future; in fact, Deletang et al. (2023) claimed that they avoided string classification and focused on sequence-to-sequence transduction instead because of the difficulty of negative sampling, which is a problem we have now addressed.
> >
> > As for contextualizing the contribution of the DFA sampling algorithm, the closest work we are aware of is MLRegTest [4], as mentioned above. Their approach uses DFA intersection, and it does not sample from the posterior distribution, but reweights the DFA in a way that involves a topological sort. This seems to suggest that it runs in $O(n_{max}^2)$ time, but they do not provide a runtime analysis, so we're not sure if it's higher. If they were to sample from the posterior distribution, they would need to run weight pushing on the intersected DFA, and the runtime would be $O(n_{max}^4)$ as described in our paper. In contrast, our method samples from the posterior *and* runs in $O(n_{max}^2)$ time. It's also not clear if their method can be extended to other types of automata such as pushdown automata, but our method that uses the counting semiring can, which is something we hope to do in future work.
> >
> > [4] Van der Poel et al. "MLRegTest: A Benchmark for the Machine Learning of Regular Languages." 2023. URL: https://arxiv.org/abs/2304.07687
> >
> > [5] Gold, E. Mark. "Language identification in the limit." 1967. URL: https://www.sciencedirect.com/science/article/pii/S0019995867911655

---

> > > ### Author Response · Authors · 2024-11-26
> > >
> > > We have updated our draft. You may be particularly interested in our addition of a "Comparison to prior work" subsection that more explicitly lays out the differences from prior work that we have discussed. This includes a discussion of the contribution of our sampling method vs. that of MLRegTest. We have also added more references to prior work as appropriate throughout the paper.
> > >
> > > Please let us know if you have any other questions or concerns.

---

> > > > ### Comment · Reviewer_gn5o · 2024-11-27
> > > > **responses to authors**
> > > >
> > > > I would like to thank the author for their responses. My score has moved from 3 to 5 based on the current version of the paper.

---

### Official Review · Reviewer_t9kn · 2024-11-03

**Soundness:** 4
**Presentation:** 3
**Contribution:** 2
**Rating:** 6
**Confidence:** 3

**Summary:**

This paper takes a formal language theory approach to string classification and asks how well various types of neural networks recognize languages. The authors evaluate the network types on eighteen formal languages from three of the four languages classes of the Chomsky hierarchy (excluding recursively enumerable languages). Length generalization is tested systematically by using a short and a long validation set, and a test set that comprises much longer strings than in either validation set. They find that recurrent types of networks tend to be more accurate than transformers. The paper contains some analysis of certain languages on which one or more network types exhibits poor performance. The authors additionally describe an algorithm for efficiently sampling strings from a regular language that they describe as concurrent work.

--------

I will refer to the following papers in this review. [1] is relevant because the paper being reviewed here appears to be, at least in part, a response to [1]. [2] is relevant as an example of prior work with substantial overlap in experimental setting and possibly different results. [3] provides theoretical insights into the results reported in Section 4 of the paper reviewed here.

* [1] Neural networks and the chomsky hierarchy, Del{\'e}tang et al, 2023
* [2] MLRegTest: A Benchmark for the Machine Learning of Regular Languages, van der Poel et al, 2024
* [3] Why are Sensitive Functions Hard for Transformers?, Hahn & Rofin, 2024

**Strengths:**

This is a well-performed study of the ability of neural networks to recognize formal languages. The writing is of good quality, the methodology is clearly communicated, and it introduces an efficient algorithm for sampling strings from regular languages. Models are trained with validation sets of length [0, 40] (short) or [0, 80] (long) and tested on a set with strings of length [0, 500]. The authors of this study introduce several additional languages -- Dyck-(2, 3), binary strings that start with 1, and a distinction between marked and unmarked reversed strings -- that do not appear in [1]. The appendix contains copious information about the models trained.

**Weaknesses:**

* The general method for training neural networks as language recognizers is straightforward and is difficult to see as a substantial contribution (see similarities with e.g. [2]).
* The algorithm for efficiently sampling from a regular language is described, and the authors state that they use it for sampling training and evaluation instances from the regular languages (class `R` in Table 1). Since the algorithm is only used to generate data for 25% of the types of languages in this study, the amount of page real estate devoted to it seems disproportionate, particularly since algorithms for sampling from a formal language are not typically showcased in this venue.
* The insights about which languages are best-suited to which network types could be much deeper. With the literature on language recognition having advanced somewhat, reports that explain *the why* (e.g. [3]) are more compelling than those that describe *the what*. The results reported here are similar to [1] on the modular arithmetic language and marked string reversal and differ from [1] on cycle navigation. Some analysis is done here to characterize the kinds of errors transformers make on the task of recognizing strings from the cycle navigation language. The authors describe how with cycle navigation the recognition error of the instances with the highest cross entropy loss is always due to the model recognizing an instance with an incorrect final character (the position in the cycle). Since [1] reports 100% transduction accuracy for cycle navigation for RNN and LSTM, and 62% for a transformer, what explains the ceiling on recognition accuracy for all networks here? The analysis of Dyck-(2, 3) and Repeat 01 seems to primarily indicate that, with transformers,  some negative instances are near the decision boundary to positive instances. More insights about *why* this occurs with transformers and not the other networks would improve the paper.

Nit
* The standard deviation of recognition is a figure of merit for distinguishing neural network types, particularly in cases when no network achieves perfect recognition accuracy. This is included in the appendix. It would be helpful to report in Table 2 mean and stdev of recognition accuracy across multiple runs using the model from the appropriate table in the appendix.

**Questions:**

1. Can you clarify the contribution of this work in light of previous work on language recognition with neural networks? For instance, the authors of [2] have evaluated with larger datasets and have trained many more models. How does your contribution differ in terms of language classes tested and insights derived therefrom? Are there other aspects of your work that can be highlighted?
2. Have you considered investigating the scaling properties of the neural networks on the language recognition task? [1] demonstrated that none of the networks in your study are able to perform transduction on recursively enumerable languages without the support of an external memory module (e.g. Stack-RNN or Tape-RNN), but -- with respect to scaling -- only showed that increasing the amount of training data when the network is large enough to memorize the training data did not help generalization for some neural network types. Related: do you have any reason to believe that results you report here that differ in order from e.g. [1] -- such as recognition on cycle navigation -- are not due to the capacity of the network? Would larger networks recognize cycle navigation better?

---

> ### Author Response · Authors · 2024-11-15
> **Responses to comments and questions**
>
> Thank you very much for your thoughtful review and feedback.
>
> > The general method for training neural networks as language recognizers is straightforward and is difficult to see as a substantial contribution (see similarities with e.g. [2]).
>
> We agree that MLRegTest is an important point of comparison -- it's an excellent paper. MLRegTest focuses exclusively on regular languages, and its negative sampling method is specific to regular languages and cannot be readily applied to non-regular languages. In contrast, our method generalizes to any language (including non-regular languages), as long as that language has (1) an algorithm for positive sampling and (2) an algorithm for membership testing. We also provide a simple, general method for generating adversarial negative examples. Reference [2] specifically cited the difficulty of negative sampling as a reason that they did not do language recognition, which ultimately weakened their claims.
>
> The negative sampling method for MLRegTest converts the DFA for the language to a DFA that recognizes the complement of the language, then renormalizes and samples from the complement DFA to generate negative samples. They also generate adversarial examples by composing the DFA with an FST, and they only generate examples with an edit distance of 1. Our perturbation-based negative sampling is an improvement in that it is simpler, faster (since it does not require FST composition), is not restricted to an edit distance of 1, and is not restricted to regular languages.
>
> > The algorithm for efficiently sampling from a regular language is described, and the authors state that they use it for sampling training and evaluation instances from the regular languages (class R in Table 1). Since the algorithm is only used to generate data for 25\% of the types of languages in this study, the amount of page real estate devoted to it seems disproportionate, particularly since algorithms for sampling from a formal language are not typically showcased in this venue.
>
> A fair point. We will consider ways of making this section more succinct. Although it is currently used only for regular languages, we do believe our length-constrained sampling algorithm is an important contribution, and it requires a fair bit of technical explanation, which is why we devoted so much space to it in the main text. Additionally, this algorithm can be useful for sampling from non-regular languages as it can be extended for other types of automata, which is something we are planning to do in future work.
>
> (Continued in next comment)

---

> > ### Author Response · Authors · 2024-11-15
> > **Responses to comments and questions (2)**
> >
> > > The insights about which languages are best-suited to which network types could be much deeper. With the literature on language recognition having advanced somewhat, reports that explain the why (e.g. [3]) are more compelling than those that describe the what. The results reported here are similar to [1] on the modular arithmetic language and marked string reversal and differ from [1] on cycle navigation. Some analysis is done here to characterize the kinds of errors transformers make on the task of recognizing strings from the cycle navigation language. The authors describe how with cycle navigation the recognition error of the instances with the highest cross entropy loss is always due to the model recognizing an instance with an incorrect final character (the position in the cycle). Since [1] reports 100\% transduction accuracy for cycle navigation for RNN and LSTM, and 62\% for a transformer, what explains the ceiling on recognition accuracy for all networks here? The analysis of Dyck-(2, 3) and Repeat 01 seems to primarily indicate that, with transformers, some negative instances are near the decision boundary to positive instances. More insights about why this occurs with transformers and not the other networks would improve the paper.
> >
> > Regarding Cycle Navigation: This is a good question, and we will discuss this more in our next revision. In [1], the model is always given a properly formatted sequence of movements as input and only needs to predict the correct digit as output. In contrast, in our recognition task, the model must also learn the correct format of positive examples and must learn to reject examples that have the right format but the wrong digit at the end. This makes the recognition task more difficult, which may explain the accuracy ceiling.
> >
> > Regarding Dyck-(2, 3) and Repeat 01: Repeat 01 is a high-sensitivity Boolean function, and Dyck-(2, 3) is also highly sensitive to changing symbols in the input string, so the theoretical arguments in [3] apply to these two languages. On the other hand, some expressivity literature suggests that transformers should be able to learn regular languages [4] or Dyck-(2, 3) [5]. RNNs and LSTMs have long been related to finite automata, and it is not surprising that they do well on these two regular languages (see, e.g., [6]).
> >
> > Our next revision will include much more discussion of these results, with more comparison to the theoretical literature. Note that we already include some references to [3].
> >
> > > The standard deviation of recognition is a figure of merit for distinguishing neural network types, particularly in cases when no network achieves perfect recognition accuracy. This is included in the appendix. It would be helpful to report in Table 2 mean and stdev of recognition accuracy across multiple runs using the model from the appropriate table in the appendix.
> >
> > Thank you for this suggestion. We will include standard deviations in the next revision.
> >
> > > Can you clarify the contribution of this work in light of previous work on language recognition with neural networks? For instance, the authors of [2] have evaluated with larger datasets and have trained many more models. How does your contribution differ in terms of language classes tested and insights derived therefrom? Are there other aspects of your work that can be highlighted?
> >
> > Broadly speaking, although there have been a number of past papers that use neural networks for formal language recognition, each paper uses a slightly different methodology, without much empirical justification for the particular choices made. Our work is an attempt to unify this methodology, to extend it to the broadest possible class of languages, to compare different training objectives used in past work, and to spur theoretically sound empirical work in the future. Despite its popularity, [1] does not make theoretically sound claims. We demonstrate our method by rerunning the experiments of [1] as language recognition tasks proper, which includes languages from various levels of the Chomsky hierarchy up to context-sensitive, although our goal is not to claim that particular architectures correspond to particular classes in the Chomsky hierarchy.
> >
> > As mentioned above, the methods in [2] are specific to regular languages, whereas our method can be used for any language where sampling and membership testing are tractable. Our length-constrained sampling algorithm for regular languages also seems to be more scalable; they test up to length 50, whereas we test up to length 500.
> >
> > (Continued in next comment)

---

> > > ### Author Response · Authors · 2024-11-15
> > > **Responses to comments and questions (3)**
> > >
> > > Other relevant work on training recognizers includes [7] and [8]. Reference [7] does negative sampling for Dyck languages by editing positive examples, but does so in a language-specific way. Reference [8] also does negative sampling by editing positive examples up to 9 times, although it doesn't mention what distribution for the number of edits is used. In contrast, we sample numbers of edits from a geometric distribution and show the ground-truth edit distance distribution in Figure 1.
> > >
> > > Another aspect of our work that can be highlighted: Our work also includes two sets of experiments that carefully distinguish between tests of inductive bias and expressivity. We are not aware of work on language recognition that has made this direct comparison before. For example, the experiments of [1] exclusively test inductive bias, and they don't seem to make a distinction between inductive bias and expressivity. Our main finding is that although inductive bias could render the set of languages an architecture can recognize quite different from what it can express, we see a remarkable consistency between inductive bias and expressivity across languages in terms of the ranking of the architectures (we will update our presentation of the results in the next revision in a way that makes this more apparent). This helps vindicate the very large body of work on theoretical expressivity.
> > >
> > > Our method can also easily be scaled up to larger datasets and more models. Incidentally, in our next revision, we will include updated experimental results that aggregate across 10 runs per setting instead of 5.
> > >
> > > > Have you considered investigating the scaling properties of the neural networks on the language recognition task? [1] demonstrated that none of the networks in your study are able to perform transduction on recursively enumerable languages without the support of an external memory module (e.g. Stack-RNN or Tape-RNN), but -- with respect to scaling -- only showed that increasing the amount of training data when the network is large enough to memorize the training data did not help generalization for some neural network types.
> > >
> > > We purposely used a constant parameter budget for all of our experiments in order to keep the number of experiments manageable, but we agree that this is a question worth exploring. We had time to run some new experiments to answer this question. We trained models with 32k, 64k, and 128k parameters with (R) loss and with the long validation set, and looked at the max test accuracy of 10 runs. In general, we see that there is no difference in accuracy for most architectures on most languages, suggesting that the results are due to inherent properties of the architecture. There are only three notable exceptions where model size does correlate with accuracy:
> > >
> > > | Architecture on Language | 32k | 64k | 128k |
> > > | ------------------------ | --- | --- | ---- |
> > > | Tf on Modular Arithmetic | .86 | .88 | .92 |
> > > | LSTM on Marked Reversal | .91 | .95 | .96 |
> > > | LSTM on Bucket Sort | .81 | .83 | .93 |
> > >
> > > > Related: do you have any reason to believe that results you report here that differ in order from e.g. [1] -- such as recognition on cycle navigation -- are not due to the capacity of the network? Would larger networks recognize cycle navigation better?
> > >
> > > In the new experiments mentioned above, for Cycle Navigation, doubling the size of the network had no effect on accuracy. We have also tried increasing the size of the training data to 100k and increasing the frequency of perturbed negative examples, which also did not increase accuracy.
> > >
> > > [4] Liu et al. "Transformers Learn Shortcuts to Automata." 2023. URL: https://openreview.net/forum?id=De4FYqjFueZ
> > >
> > > [5] Yao et al. "Self-Attention Networks Can Process Bounded Hierarchical Languages." 2021. URL: https://aclanthology.org/2021.acl-long.292/
> > >
> > > [6] Hewitt et al. "RNNs can generate bounded hierarchical languages with optimal memory." 2020. URL: https://aclanthology.org/2020.emnlp-main.156/
> > >
> > > [7] Bhattamishra et al. "Separations in the Representational Capabilities of Transformers and Recurrent Architectures." 2024. URL: https://arxiv.org/abs/2406.09347
> > >
> > > [8] Weiss et al. "Extracting Automata from Recurrent Neural Networks Using Queries and Counterexamples." 2018. URL: https://proceedings.mlr.press/v80/weiss18a.html

---

> ### Comment · Reviewer_t9kn · 2024-11-21
>
> Thank you very much for the clarification wrt MLRegTest. I should have caught that a key difference between your work and MLRegTest is the scope of the languages. (It's right there in the title of their paper, after all, but some of the terms they use is unfamiliar to me and I see now they are different types of regular languages.)
>
> With respect to the amount of the paper devoted to the sampling algorithm, I understand that its contribution is substantial. My concern is that most or all of the description of the algorithm is already included in the concurrent work cited on line 196. If that's the case, the paper under review here could benefit from -- as you indicate -- a more succinct treatment of the algorithm and more experimental results and analysis. I do acknowledge the need for some presentation of the algorithm, but the results and analysis sections could be burgeoned to great effect.
>
> Based on author responses, I have increased my rating from 5 to 6. If this paper is accepted, I would like to see more results and analysis thereof in the camera-ready version, and a more succinct treatment of the sampling algorithm.

---

> > ### Author Response · Authors · 2024-11-26
> >
> > Thanks! We have updated our draft, and you may be particularly interested in these changes:
> >
> > 1. We have shortened the sampling section from about 2.25 pages to about 1.9 pages. We whittled it down to definitions, choices that are unique to this paper (such as the way we set the PDFA weights and the lifting function), and a brief summary of the algorithms and their time complexity.
> > 2. We have expanded our results section with more discussion and experiments, including analyses of performance vs. length, more comparison to the theoretical literature, and more discussion of accuracy ceilings. We also have a lot more references for theoretical results about specific languages in the appendix now. When we have more time, before the camera-ready, we will also include the result on smaller and larger models.
> > 3. We have added a "Comparison to prior work" subsection that more explicitly lays out the differences from prior work that we have discussed. We have also added more references to prior work as appropriate throughout the paper.
> > 4. We have included standard deviations in the main results table.

---

### Official Review · Reviewer_v4NN · 2024-11-04

**Soundness:** 3
**Presentation:** 3
**Contribution:** 3
**Rating:** 6
**Confidence:** 3

**Summary:**

The paper explores formal language recognition tasks with different architectures. The authors argue that the primary motivation underlying formal language theory is language recognition and not next token prediction as used by existing literature. As such, they propose a novel data generation algorithm that can generate positive and hard negative examples from a diverse set of formal languages. Finally, through extensive experiments, they show that RNNs and LSTMs generally outperform transformer architectures.

**Strengths:**

The strength of the paper lies in its motivation to quantify gaps between different architectures by formal language recognition tasks. The authors start with a discussion on the basis of formal language theory, and how existing works differ from this basis in their experimental settings. Then, they propose a novel data generation algorithm that can sample positive and hard negative samples from each formal language. The algorithm has been discussed in great detail, and the running time has been contrasted with naive data generation algorithms.

The authors then conduct extensive comparisons of RNNs, LSTMs, and transformers across a wide suite of formal languages. Interestingly, RNNs and LSTMs generally perform better. Other interesting observations include (a) transformers can't learn Dyck-(2,3), as opposed to the results of Yao et al. (2021), and (b) the order of ranking between the different architectures is different from Deletang et al. (2023). The work emphasizes the need for empirical validations that align with theoretical results for a clearer understanding of neural network capabilities.

**Weaknesses:**

As such, the work doesn't have many weaknesses. I have a couple of questions regarding the setup.

a) **Behavior with input length**
- The authors report aggregated scores over strings of all lengths. It would be interesting to include a discussion on input length v/s performance for different models. Are there tasks where the transformer's performance shows a more exponential/drastic drop with input length?

b) **Tasks where additional loss terms hurt:**
- Are there tasks where the additional language model loss and the next symbol loss hurt performance? For example, for parity, language model loss can intuitively hurt performance as the token at each step could be randomly selected for each string.
- Furthermore, how do the optimal $\lambda$ hyper-parameters behave for different tasks for different models? Are there cases, where the additional losses help one architecture more than others?

c) **How is 64k parameters and 5 layers decided for the experiments?** What happens for smaller and wider models?

**Questions:**

Please check my questions in the above section.

---

> ### Author Response · Authors · 2024-11-15
> **Responses to questions**
>
> Thank you very much for your review and positive feedback.
>
> > The authors report aggregated scores over strings of all lengths. It would be interesting to include a discussion on input length v/s performance for different models. Are there tasks where the transformer's performance shows a more exponential/drastic drop with input length?
>
> In early experiments on Marked Reversal, we did not see a trend in accuracy as a function of length, which is why we chose to analyze edit distance instead. We did not examine this for other languages, but we agree that doing so would be interesting, since a common failure mode in the language modeling/transduction setup of Deletang et al. (2023) was for accuracy to decrease dramatically with length. We will analyze this for the other languages and report back with our findings.
>
> > Are there tasks where the additional language model loss and the next symbol loss hurt performance? For example, for parity, language model loss can intuitively hurt performance as the token at each step could be randomly selected for each string.
>
> Yes. For Parity, the language modeling term hurts the mean test accuracy of the LSTM but has almost no effect for the transformer (see Table 6). For Even Pairs, next symbol prediction hurts the mean test accuracy of the LSTM, but benefits the RNN, and has little effect on the transformer. We didn't notice a consistent trend across languages or architectures. In general, recognition alone (R) is the most frequent best loss function, followed by next symbol prediction (R+NS). Loss functions that include language modeling (R+LM, R+LM+NS) are least frequently the best. We will add more discussion of the effect of different loss terms to our results section in the next revision.
>
> We'd like to note that language modeling doesn't necessarily hurt performance due to randomness in the language *per se*; if there is randomness, the language modeling head just needs to learn the optimal *distribution* over next symbols.
>
> > Furthermore, how do the optimal $\lambda$ hyper-parameters behave for different tasks for different models? Are there cases, where the additional losses help one architecture more than others?
>
> We will analyze the $\lambda$ values and get back to you.
>
> Yes, there are cases where additional loss terms help one architecture more than others. For instance, on the language First with the short validation set, the LM loss increases the mean test accuracy for the RNN much more than for the transformer, although the transformer has higher accuracy. It has little effect on the LSTM.
>
> > How is 64k parameters and 5 layers decided for the experiments?
>
> This was admittedly a somewhat arbitrary choice. We deliberately kept the number of hyperparameters we swept over very small in order to keep the number of experiments manageable. Using 64k parameters is enough for the RNN to have 79 hidden units, for the LSTM to have 40 hidden units, and for the transformer to have a $d_{model}$ of 32 when the alphabet is binary. These model sizes are comparable with prior work that evaluates NNs on formal languages, where it is typical to use smaller models than one would use for, say, natural language. Using 5 layers seemed like a reasonable compromise between too few and too many layers. A lot of prior work on formal languages uses only 1 layer, so we wanted to use more.
>
> > What happens for smaller and wider models?
>
> We had time to run some experiments to answer this question. We trained models with 32k, 64k, and 128k parameters with (R) loss and with the long validation set, and looked at the max test accuracy of 10 runs. In general, we see that there is no difference in accuracy for most architectures on most languages, suggesting that the results are due to inherent properties of the architecture. There are only three notable exceptions where model size does correlate with accuracy:
>
> | Architecture on Language | 32k | 64k | 128k |
> | ------------------------ | --- | --- | ---- |
> | Tf on Modular Arithmetic | .86 | .88 | .92 |
> | LSTM on Marked Reversal | .91 | .95 | .96 |
> | LSTM on Bucket Sort | .81 | .83 | .93 |

---

> ### Author Response · Authors · 2024-11-21
> **Performance vs. input length**
>
> We had time to generate plots of recognition cross-entropy vs. length for the models with the best test accuracy scores after being trained on the long validation set. In general, we found that cross-entropy did not increase significantly on longer strings for all languages and architectures. This is in contrast to Deletang et al. (2023), who found that many architectures fail catastrophically on longer strings in their language modeling/sequence-to-sequence setup. This difference might partly be because our experimental setup tests expressivity instead of inductive bias, so it is less prone to overfitting on the training length distribution.
>
> Would you be interested in seeing these plots in the next revision?

---

> ### Author Response · Authors · 2024-11-26
> **Response**
>
> We have uploaded a revised draft of our paper. In particular, you may be interested in the following changes:
>
> 1. We have added plots of cross-entropy vs. length to the appendix, which we discuss in the main text. For some languages, such as Repeat 01 and Dyck-(2, 3), we actually do see a slight upward trend in cross-entropy for the transformer, but it is quite gradual, unlike what is shown in Deletang et al. (2023).
> 2. We have added a discussion of cases where certain loss terms help and hurt in the Results section, pointing out that LM is detrimental for the LSTM on Parity.
>
> When we have time, before the camera-ready, we will also add the results on smaller and larger models.
>
> Regarding the $\lambda$ values: below is a table of the $\lambda$ values of the best models trained on the long validation set. A value of 0.000 means the corresponding loss term wasn't included. They tend to have pretty high variance for all architectures. It does seem like the transformer favors a range of smaller LM and NS coefficients than the RNN/LSTM. Other than that, we haven't noticed any clear trends.
>
> | Language | Tf | RNN | LSTM |
> | ----------------- | ------- | ------- | ------- |
> | even-pairs | LM:0.000, NS:0.000 | LM:0.000, NS:4.298 | LM:0.000, NS:0.000 |
> | repeat-01 | LM:0.000, NS:0.000 | LM:0.000, NS:0.000 | LM:0.000, NS:0.000 |
> | parity | LM:0.000, NS:2.860 | LM:9.756, NS:0.000 | LM:0.000, NS:0.000 |
> | cycle-navigation | LM:0.000, NS:0.000 | LM:0.000, NS:0.000 | LM:0.000, NS:0.000 |
> | modular-arithmetic-simple | LM:0.000, NS:0.021 | LM:0.000, NS:0.000 | LM:0.030,NS:0.000 |
> | dyck-2-3 | LM:0.000, NS:0.038 | LM:1.189, NS:0.851 | LM:0.000, NS:0.011 |
> | first | LM:0.000, NS:0.000 | LM:0.000, NS:0.000 | LM:0.000, NS:0.000 |
> | majority | LM:0.148, NS:0.000 | LM:0.784, NS:0.189 | LM:7.859, NS:9.095 |
> | stack-manipulation | LM:0.081, NS:1.793 | LM:0.000, NS:0.000 | LM:0.022, NS:0.000 |
> | marked-reversal | LM:0.474, NS:0.000 | LM:0.182, NS:0.000 | LM:0.032, NS:0.000 |
> | marked-copy | LM:0.000, NS:0.395 | LM:0.000, NS:0.055 | LM:0.000, NS:0.000 |
> | missing-duplicate-string | LM:0.025, NS:1.128 | LM:0.060, NS:3.989 | LM:5.458, NS:0.000 |
> | odds-first | LM:0.266, NS:0.451 | LM:0.000, NS:0.000 | LM:2.148, NS:4.674 |
> | binary-addition | LM:0.624, NS:0.000 | LM:0.000, NS:6.422 | LM:0.000, NS:0.710 |
> | binary-multiplication | LM:0.000, NS:1.682 | LM:0.000, NS:0.752 | LM:0.176, NS:0.389 |
> | compute-sqrt | LM:0.000, NS:0.000 | LM:3.848, NS:0.040 | LM:0.000, NS:0.000 |
> | bucket-sort | LM:0.000, NS:0.482 | LM:0.000, NS:0.537 | LM:0.032, NS:0.024 |

---

> > ### Comment · Reviewer_v4NN · 2024-12-02
> >
> > Apologies for the delay in my response. I would like to thank the authors for their extensive experiments and detailed responses. I also read through the authors' responses to reviewers t9kn and gn5o, noting that the primary concern raised was the novelty of the work in comparison to prior studies. In hindsight, I realize I should have raised a similar question before the review deadline.
> >
> > I agree with reviewer t9kn that the primary focus here should be comparison to MLRagTest, and how this paper advances their claims by incorporating experiments beyond regular languages. My score with the current draft is closer to a score of 7, so I will maintain my current score. But I will vouch for this paper during AC-reviewer discussion period.

---

> > > ### Author Response · Authors · 2024-12-02
> > >
> > > Thank you very much for your response and for carefully reviewing our discussions with the other reviewers. As you probably saw, we do now include a dedicated section comparing against MLRegTest and some other work that uses elements of our methodology for training recognizers. We agree with the reviewers' concerns about laying out differences from prior work, although we would like to push back a little on the notion that MLRegTest should be our primary focus. We overlap in only a few areas: (1) length-constrained sampling from DFAs (we provide a better algorithm), (2) the fact that we test multiple language classes, (3) the fact that we train recognizers, and (4) we both attempt to generate adversarial examples (we have a more general method). But in addition to our methodological improvements, we answer some questions they don't consider, namely the effects of using different training objectives, and differences between testing inductive bias and expressivity.

---

> > > > ### Comment · Reviewer_v4NN · 2024-12-03
> > > >
> > > > Thank you for the response. Yes, I re-read the responses and I agree with the authors that the paper substantially differs from MLRegTest.

---

### Meta-Review · Area_Chair_YyWg · 2024-12-23

**Metareview:**

This submission addresses a fundamental concern in the literature on neural network expressivity: the discrepancy between theoretical claims of what neural architectures can recognize and how these capabilities are tested empirically. The authors propose training neural networks directly as recognizers of formal languages (i.e., binary classification tasks) rather than relying on proxy tasks like language modeling or sequence-to-sequence transduction. They introduce a general and flexible approach for generating both positive and negative samples that can be applied to a wide range of formal languages across different levels of the Chomsky hierarchy. A core technical contribution is an efficient algorithm for length-controlled sampling from regular languages represented as DFAs, which can be extended to more complex automata.

The experiments compare three common architectures—RNNs, LSTMs, and transformers—on a broad set of 18 languages. The authors report that recurrent architectures often achieve better performance than transformers, contradicting some earlier claims that transformers can learn certain complex formal languages with ease. The paper also systematically varies training objectives, comparing purely recognizable (R) losses to mixed objectives that include language modeling (LM) and next-symbol prediction (NS). While no single objective is best across all scenarios, the authors find that simpler binary classification objectives can be quite effective. The authors also carefully distinguish between testing inductive bias vs. testing expressivity.

Strengths:
	•	The authors highlight a significant methodological gap between theoretical claims and empirical evaluations in the literature.
	•	They propose a unified methodology for testing neural networks on actual recognition tasks that are aligned with formal language definitions.
	•	An efficient length-constrained sampling algorithm for DFAs is introduced and analyzed, improving upon existing methods.
	•	Empirical results are provided for a variety of languages (regular to context-sensitive), showing unexpected performance differences, such as RNNs/LSTMs often outperforming transformers.
	•	Extensive responses to reviewers address issues related to novelty, placement relative to prior work (MLRegTest and others), and provide additional results (including performance vs. input length and model scaling).

**Additional Comments On Reviewer Discussion:**

Weaknesses and Concerns Addressed:
	•	Some reviewers initially felt the paper did not fully situate its approach relative to prior work that also trains recognizers, such as MLRegTest and others. In subsequent revisions and responses, the authors clarified how their method differs (e.g., general applicability beyond regular languages, more flexible negative sampling, comparisons of multiple loss functions, and length scaling).
	•	Concerns that the sampling algorithm section was too long have been mitigated by making that section more concise and dedicating more space to experimental results and analysis.
	•	The authors have included additional references and a new “Comparison to prior work” subsection to strengthen the contextualization of their contribution.

Reviewers’ Positions:
	•	One reviewer (R4) was supportive from the start, emphasizing that the work provides valuable methodological insights and references important theoretical work.
	•	Another reviewer (R2) and yet another (R3) initially rated the paper as marginally above the threshold but after the authors’ clarifications, both expressed more confidence and support. They appreciated the authors’ additional experiments, detailed responses, and clarifications of how this work complements and extends previous results.
	•	A third reviewer (R1) initially felt the novelty was insufficiently emphasized. However, after the discussion and revision, they raised their score. All reviewers now generally view the paper positively.

Final Recommendation:
After considering the authors’ thorough revisions, clarifications, and the additional experiments run in response to reviewer feedback, the paper now seems well-positioned as a valuable contribution to the literature. It advances the methodology for empirical studies of formal language recognition by neural networks, makes clear how it differs from and improves upon prior studies, and provides rich empirical data and insights. The consensus among the reviewers after discussion is more positive, and the paper’s contributions are now well-supported.

---

### Decision · Program_Chairs · 2025-01-22

Accept (Poster)